# Characterization of Freshly Isolated Human Peripheral Blood B Cells, Monocytes, CD4+ and CD8+ T Cells, and Skin Mast Cells by Quantitative Transcriptomics

**DOI:** 10.3390/ijms252313050

**Published:** 2024-12-04

**Authors:** Srinivas Akula, Abigail Alvarado-Vazquez, Erika Haide Mendez Enriquez, Gürkan Bal, Kristin Franke, Sara Wernersson, Jenny Hallgren, Gunnar Pejler, Magda Babina, Lars Hellman

**Affiliations:** 1Department of Cell and Molecular Biology, Uppsala University, The Biomedical Center, Box 596, SE-751 24 Uppsala, Sweden; srinivas.akula@icm.uu.se; 2Department of Animal Biosciences, Swedish University of Agricultural Sciences, Box 7023, SE-75007 Uppsala, Sweden; sara.wernersson@slu.se; 3Department of Medical Biochemistry and Microbiology, The Biomedical Center, Box 582, SE-75123 Uppsala, Sweden; abigail.vazquez@imbim.uu.se (A.A.-V.); erika.enriquez@imbim.uu.se (E.H.M.E.); jenny.hallgren@imbim.uu.se (J.H.); gunnar.pejler@imbim.uu.se (G.P.); 4Institute of Allergology, Charité—Universitätsmedizin Berlin, Corporate Member of Freie Universität Berlin and Humboldt-Universität zu Berlin, Hindenburgdamm 30, 12203 Berlin, Germany; guerkan.bal@charite.de (G.B.); kristin.franke@charite.de (K.F.); magda.babina@charite.de (M.B.); 5Fraunhofer Institute for Translational Medicine and Pharmacology ITMP, Immunology and Allergology IA, Hindenburgdamm 30, 12203 Berlin, Germany

**Keywords:** mast cells, B lymphocytes, T lymphocytes, monocytes, transcriptome, granule proteases, protease inhibitors, Fc receptors, CD molecules, MHC Class I and II, transcription factors, signaling molecules, integrins, selectins, complement components, pattern recognition receptors, TLRs

## Abstract

Quantitative transcriptomics offers a new way to obtain a detailed picture of freshly isolated cells. By direct isolation, the cells are unaffected by in vitro culture, and the isolation at cold temperatures maintains the cells relatively unaltered in phenotype by avoiding activation through receptor cross-linking or plastic adherence. Simultaneous analysis of several cell types provides the opportunity to obtain detailed pictures of transcriptomic differences between them. Here, we present such an analysis focusing on four human blood cell populations and compare those to isolated human skin mast cells. Pure CD19^+^ peripheral blood B cells, CD14^+^ monocytes, and CD4^+^ and CD8^+^ T cells were obtained by fluorescence-activated cell sorting, and KIT+ human connective tissue mast cells (MCs) were purified by MACS sorting from healthy skin. Detailed information concerning expression levels of the different granule proteases, protease inhibitors, Fc receptors, other receptors, transcription factors, cell signaling components, cytoskeletal proteins, and many other protein families relevant to the functions of these cells were obtained and comprehensively discussed. The MC granule proteases were found exclusively in the MC samples, and the T-cell granzymes in the T cells, of which several were present in both CD4^+^ and CD8^+^ T cells. High levels of CD4 were also observed in MCs and monocytes. We found a large variation between the different cell populations in the expression of Fc receptors, as well as for lipid mediators, proteoglycan synthesis enzymes, cytokines, cytokine receptors, and transcription factors. This detailed quantitative comparative analysis of more than 780 proteins of importance for the function of these populations can now serve as a good reference material for research into how these entities shape the role of these cells in immunity and tissue homeostasis.

## 1. Introduction

The human body contains more than 200 different cell types with different functions and phenotypes. The function and phenotype of these cells are in turn determined by the genes they express. One way to obtain a detailed map of the phenotype of a cell is to study its transcriptome. This information gives a detailed description of genes that are active and also a good estimate of the levels of these proteins in the cells. Most previous studies have been presented in the form of heat maps, where almost all quantitative information is being lost, or by single-cell analysis, where the result is great for tracing lineage commitment. However, due to low levels of RNA in the individual cells, these measurements have low resolution concerning relative expression levels. In order to better reflect the actual expression levels, which are important for the evaluation of the biological significance of the expression of these genes, here, we present a detailed description of the actual expression levels of a large panel of differentially expressed genes among five human immune cell populations. We focus on the transcriptionally highly active cell populations of the human blood, such as CD19^+^ B cells, CD14^+^ monocytes, and both CD4^+^ and CD8^+^ T cells, and compared those to tissue-resident human skin mast cells (MCs). Skin MCs are the connective tissue type of MCs originating from an early wave of progenitor cells from the yolk sac [1,2]. Terminally differentiated immune blood cells such as neutrophils, eosinophils, and basophils were not included in our analysis, since many of their specific genes have already been turned off, and therefore an analysis of their transcriptome does not give an accurate picture of their phenotype.

The RNAs from the five purified cell fractions were analyzed using the Ampliseq methodology to obtain quantitative information concerning the expression levels from essentially all of the approximately 21,000 human genes. Two individual samples from each cell type originating from two different donors were included in this study. The data are presented as actual reads and not as heat maps to give an accurate picture of transcript levels and the differences between genes in expression levels. More than 780 differentially expressed transcripts were identified among these immune cell populations and grouped by molecules of similar characteristics, such as granule proteases, Fc receptors, cytokines, cytokine receptors, transcription factors, and signaling molecules, with the aim to highlight the differences in expression between the different cell types and the consequences it may have on their biology.

Our quantitative gene expression map may be used together with the Human Protein Atlas (https://www.proteinatlas.org) as a tool for a deeper analysis of the complex phenotypes of these immune cells and how this affects their biological function during normal tissue homeostasis and under inflammatory conditions.

## 2. Results and Discussion

### 2.1. Purification of Human CD19^+^ B Cells, CD14^+^ Monocytes, CD4^+^ and CD8^+^ T Cells, and Skin MCs

Human PBMCs from buffy coats were used to FACS-purify transcriptionally active CD19^+^ B cells, CD14^+^ monocytes, and CD4^+^ and CD8^+^ T cells. Cell purity was then determined by post-purity FACS check and ranged from 87 to almost 95% (Figure 1).

Samples of human foreskin and breast collected after surgery were digested with dispase and collagenase to obtain single-cell suspensions. MCs were purified by magnetic cell sorting using a nonactivating anti-c-kit antibody, which resulted in approximately 98% pure MCs. The total RNAs from the isolated cells were used for Ampliseq transcriptome analysis.

Two samples for each of the purified cell populations were analyzed. For MCs, the expression of selected genes after 3 weeks of in vitro culture was included in the analysis, by adopting previously published data and data found in Appendix A [3]. The Excel file for the entire 15 samples, including all 20,803 listed genes, is available in Appendix A. This file includes the four samples of cultured human skin MCs analyzed in a previous comparative analysis of similarities and differences in the phenotypes between freshly isolated and cultured human skin MCs [3].

To facilitate interpretation of the data, genes with reads between 1 and 9 are marked in light gray, genes with reads between 10 and 99 in light blue, and genes with reads > 100 in red.

### 2.2. The Major Granule Proteases Are Mainly Expressed in MCs and T Cells

The different granule proteases are of particular interest for several of these five studied cell populations as they are highly expressed and also restricted to a particular cell type. For example, several granule proteases are MC-specific and can account for as much as 35% of the total protein content of MCs [4]. All the classical MC granule proteases, including cathepsin G (**CTSG**), the beta tryptase (**TPSB2**), the MC-specific carboxypeptidase A3 (**CPA3**), and the MC chymase (**CMA1**), were almost exclusively expressed by the MCs, and all of them at very high levels. Only a very low level of one of them, the chymase **CMA1**, was detected in one of the two monocyte samples (Table 1). The number of reads of **CTSG** was 9445 and 6194 in MCs, whereas 7096 and 6177 reads were identified for the tryptase **TPSB2**, 4403 and 3086 for **CPA3,** and 1713 and 1359 for chymase **CMA1** (Table 1). Two additional tryptases, the delta tryptase **TPSD1** and the gamma tryptase **TPSG1**, and one additional carboxypeptidase, **CPM**, were also expressed exclusively in MCs, but at much lower levels, with 418 and 67 reads, 86 and 49 reads, and 140 and 81 reads, respectively (Table 1). Two of the proteases expressed by MCs have also been previously detected in basophils, a cell type not included in this analysis because they are similar to neutrophils and eosinophils terminally differentiated when they leave the bone marrow and the majority of lineage specific genes have been turned of and can therefore not be detected in the transcrioptome. Both tryptase (**TPSB2**) and carboxypeptidase A3 (**CPA3**) have been detected by immunohistochemistry in human basophils [5,6]. Mouse basophils express two other basophil-specific proteases, **mMCP-8** and **mMCP-11,** as well as **CPA3** [7,8,9,10]. A number of different functions have been described for these very abundant MC granule proteases, including the generation of a blood pressure regulating peptide, angiotensin II, from angiotensin I; activation of matrix metalloproteases; selective cleavage of TH2-promoting cytokines; cleavage and inactivation of various venoms; and also cleavage of anti-coagulant proteins from leach, ticks and mosquitos [6,11,12,13,14,15].

The results from the analysis of the T-cell- and NK-cell-expressed granzymes were also illuminating. All five of the human granzymes were expressed exclusively in T cells (Table 1). Interestingly, three of the five human granzymes, granzyme A (**GZMA**), K (**GZMK**), and M (**GZMM**), were expressed in both CD4^+^ and CD8^+^ T cells, whereas granzymes B (**GZMB**) and H (**GZMH**) were only detected in the CD8^+^ T cells (Table 1). Granzyme A, K, and M transcript levels were approximately 50–80% lower in the CD4^+^ T-cells compared to CD8^+^ cells (Table 1). A similar situation was seen for two additional granule proteins of T cells, perforin (**PRF1**) and granulysin (**GNLY**). Both of these were only expressed in T cells, and here, the levels in CD4^+^ T cells were even lower compared to CD8^+^ cells, differing by a factor of 4–15 in expression levels (Table 1). The expression levels of the granzymes in T cells were much lower than the granule proteases in MCs. The expression levels ranged between 70 and 280 reads in the CD8^+^ T cells compared to 1359 to 9445 reads for the MC proteases in MCs, which was a 10–30-fold difference in expression levels (Table 1). It is important to note that we looked at T cells in the circulation, which most likely almost exclusively were resting T cells and only very few activated cells. The levels of these **granzymes** and two other granule-stored proteins, **perforin** and **granulysin**, probably increase after activation of these cells, similar to what we have seen for human monocytes after activation with LPS [16]. During our analysis of the human monocytes, we could see that some of the inflammatory cytokines and chemokines increased from almost zero to becoming the dominating transcripts after only a few hours in contact with LPS [16]. For example, IL-6 increased from 0.1 to 7500 reads, which corresponds to an increase of 75,000 times after 4 h of incubation in the presence of LPS [16]. However, to what level these granzymes increase upon the activation of T cells needs to be analyzed in more detail by obtaining quantitative data for both transcriptional activation and protein accumulation in activated cells. Also of interest are the levels these granzymes are expressed in different NK-cell populations, which are not included in this study. There are numerous conflicting reports in the literature that need to be sorted out before we have the correct view of their expression of the granzymes in various NK-cell populations (work in progress).

Concerning the function of these granzymes, granzyme B is the granzyme in which the function is most clearly defined. It has been shown to be a key component together with the pore-forming perforin in the induction of apoptosis in virus-infected cells. Granzyme A was also considered to be an apoptosis-inducing granzyme several years ago. However, more recent data have questioned this and also its potential cytokine-inducing effect, which is why we are still left without a good explanation as to the biological function of both granzymes A and K [17].

A number of labs have also reported the expression of granzymes in MCs, primarily granzyme B. However, we could not detect any expression in these human skin MCs. However, we did detect granzyme B in one of the cultured skin MCs, with 11 reads, which we analyzed in a previous study, indicating that granzyme expression is not naturally occurring in tissue MCs, at least at a level detectable in this analysis [3]. This indicates that granzyme expression is not naturally occurring in tissue MCs, at least not at a level detectable in our present analysis. Low levels of granzyme B have also been detected in freshly isolated peritoneal MCs from mouse models, although at more than 200-fold lower levels than those of the classical MC granule proteases, indicating that granzymes play a rather minor role in MC biology [18].

No expression of any of the neutrophil granule proteases, except cathepsin G, which is highly expressed in MCs, was seen in any of the five studied cell populations (Table 1). Proteinase 3 (**PRTN3**), N-elastase (**ELANE**), neutrophil serine protease 4, and NSP-4 (**PRSS57**) were all negative in these five cell populations (Table 1).

Low levels, in the range of 13 to 40 reads, were seen in four of these cell fractions for cathepsin C (**CTSC**) and only very low levels, with 5 and 6 reads, in the fifth cell population, the B cells. Cathepsin C is the protease that removes the N-terminal activation peptide of the granule proteases and therefore is an activating protease, indicating that low levels are sufficient for the proper processing of these very abundant granule proteases in both T cells and MCs before the granule storage of these proteases in their active form (Table 1).

### 2.3. Transcript Levels for Lysosomal Proteases, Matrix Proteases, and a Few Other Proteases

When we analyzed lysosomal proteases, we observed that many of them were expressed by the majority of these cells. This was expected since all cells need lysosomes for degrading damaged proteins and other macromolecules. However, there were marked differences in their expression levels. Cathepsin D (**CTSD**) was very highly expressed in both monocytes and MCs, with 1792 and 1517 reads in monocytes; 2356 and 1723 reads in MCs; and much lower, in the range of 55 to 511 reads, in the other cell populations (Table 2). Cathepsin B and L1(**CTSB** and **CTSL1**) were higher in MCs than in the other cells, within the range of 45 to 315 reads, whereas cathepsins S (**CTSS**) was much higher in monocytes, with 538 and 1011 reads (Table 2). Cathepsin W (**CTSW**) was basically only expressed by T cells, as previously shown, and also much higher in CD8^+^ than in CD4^+^ T cells, with 627 and 1151 reads in CD8^+^ cells and 104 and 105 reads in CD4^+^ cells (Table 2) [19]. This protease seems to be associated with the cell membrane or the endoplasmic reticulum, but the actual target and function have not yet been identified [19]. Serine carboxypeptidase 1 (**SCPEP1**) was primarily found in monocytes, with 206 and 303 reads, but low levels were also found in other cells (Table 2). This is an extracellular protease predicted to be involved in negative regulation of blood pressure [20]. A number of proteases belonging to the family of disintegrins and metalloprotease family (**ADAM**) were also identified in this screening (Table 2). As can be seen from the table, **ADAM15** was primarily expressed by monocytes, with 169 and 207 reads; **ADAM19** and **28** were higher in B cells, with 37 and 195 reads; and **ADAMTS7** was primarily identified in MCs, with 58 and 80 reads (Table 2).

**PSMB10** is a protease component of the proteasome and is involved in MHC Class I presentation [21]. This gene was highly active in monocytes, with 796 and 1314 reads, and less so in B cells, T cells, and MCs but present in all five cell types (Table 2). **NAPSB** is a Napsin B aspartic protease pseudogene, and here, it was expressed only in B cells and monocytes and at relatively high levels, between 103 and 459 reads [22]. **PRSS12**, also named neurotrypsin, which is a protease expressed by motor neurons, was found to be expressed at low levels by MCs only, with 38 and 59 reads (Table 2). **MMP17**, one of the matrix metalloproteases, was primarily expressed in B cells and monocytes but at relatively low levels, with between 21 and 79 reads (Table 2).

The urokinase plasmin activator (**PLAU**) was only expressed by MCs, with 38 and 102 reads, and its receptor (**PLAUR**) was highly expressed by both monocytes and MCs, with between 415 and 1371 reads (Table 2). Dipeptidyl peptidase-4 (**DPP4**) is a cell-surface protease expressed by most cell types and thought to be involved in inflammation [23]. This protease was primarily found in CD4^+^ and CD8^+^ T cells, with between 71 and 129 reads, and at lower levels in MCs but not in monocytes and B cells (Table 2).

Caspase 3 (**CASP3**), one of the key components in apoptosis induction, was expressed in all cells but at very low levels and markedly higher in MCs, with 74 and 88 reads (Table 2). The levels in the cultured MCs were much lower, similar to the other cell types, with 3–10 reads ([3] and Appendix A). The **CAPN10** gene encodes a member of a well-conserved family of calcium-dependent cysteine proteases, calpain-10. Here, it was expressed at low levels in all five cell populations, with a range of 5 to 47 reads. The gene **BACE2**, encoding beta-secretase 2, which is also known as memapsin-1, an aspartyl protease, was expressed in MCs only, with 49 and 103 reads, and at very low levels in B cells, with 10 and 12 reads [24].

### 2.4. Transcript Levels for a Panel of Protease Inhibitors

There was a relatively good correlation between the expression profiles of stored proteases and the expression levels of their inhibitors. For example, cystatin 3 (**CST3**), which is a potent inhibitor of lysosomal proteases, was highly expressed by monocytes, with 4677 and 6287 reads, as well as by MCs, but it was detected at very low levels in B and T cells (Table 3). In contrast, cystatin 7 (**CST7**) was only expressed in T cells and MCs and at much lower levels, with reads in the range of 23 to 290 (Table 3).

Tissue metalloprotease inhibitor 1 (**TIMP1)** was broadly expressed, with reads in the range of 105 to 407, except for B cells (Table 3). **TIMP2** was broadly expressed at low levels, whereas its level was high primarily in monocytes, with 491 and 488 reads, and **TIMP3** was only expressed in MCs but at a very high level, with 1270 and 995 reads (Table 3).

Serpin B1, **SERPINB1**, was broadly expressed at low levels but was higher in MCs, with 156 and 333 reads. Serpins **H1** and **E1** and **LXN** were expressed only in MCs, with between 27 and 352 reads (Table 3). The inter-alpha-trypsin inhibitor heavy-chain H4 (**ITIH4**) was detected only at low levels, primarily in monocytes, with 48 and 78 reads, and the Kunitz-type protease inhibitor 1 (**SPINT1**) was also detected primarily in monocytes, with 75 and 100 reads, (Table 3). Interestingly, here, four of these inhibitors were almost exclusively expressed by MCs, including **TIMP3**, **SERPINH1**, **SERPINE1**, and **LXN**, with the most extreme being **TIMP3** (Table 3). MCs store massive amounts of active proteases, both granule and lysosomal proteases, and may therefore also have high levels of protease inhibitors to protect from potential granule leakage of these proteases into the cytoplasm.

### 2.5. Transcript Levels for Eosinophil-, Neutrophil-, and Macrophage-Related Proteins

To analyze how specific gene expression was for “lineage-specific” protein marker genes, we analyzed the major granule proteins of neutrophils and eosinophils. None of the five cell types expressed the eosinophil cationic protein (**ECP**) (named **RNASE3**), eosinophil peroxidase (**EPX**), or the Charcot–Leyden crystal protein (**CLC**) (Table 4). In contrast, a protein closely related to **ECP**, **EDN** (named **RNASE2**), was expressed at very low levels in monocytes, with 14 and 19 reads (Table 4). Another related RNase (**RNASE6**) showed low levels of expression in B cells and monocytes (Table 4).

Low levels of one of the neutrophil proteins were found in the monocytes myeloperoxidase (**MPO**), with 13 and 9 reads, but the iron-binding protein lactoferrin (**LTF**) was not detected in any of the five cell types (Table 3). However, we found relatively high levels of two neutrophil cytosolic factors: **NCF1** (in both B cells and monocytes) and **NCF2** (in monocytes) (Table 4). These proteins are components of the reactive oxidant-generating system of neutrophils. **NCF1** and **NCF2** are components of the multi-protein enzymatic complex known as NADPH oxidase, and mutations in them cause chronic granulomatosis disease with defects in the defense against bacterial infections [25,26].

Very high levels of the antibacterial protein lysozyme, **LYZ**, were seen in monocytes. Lysozyme is an enzyme that cleaves a bond between two sugar units in bacterial cell walls, known as the peptidoglycan. Lysozyme was actually the most highly expressed protein in monocytes, with 18,602 and 23,009 reads (Table 4). Very low levels of this protein were found in all the other cell types (Table 4). The low levels in these cell populations may even originate from a few contaminating monocytes. **MPEG1** is an antibacterial protein forming pores in bacterial cells, thus enabling access to periplasmic space. This protein was expressed only by B cells and monocytes but at relatively low levels, with between 25 and 50 reads (Table 4) [27]. **COCH** is the gene for cochlin, a protein present in the inner ear, but seems to also contribute to innate immunity after cleavage by aggrecanases [28]. It was expressed only in B cells among these five cell populations but at a low level, with between 59 and 63 reads (Table 4). **MNDA**, the myeloid cell nuclear differentiation antigen, has been detected in the nuclei of cells in the granulocyte–monocyte lineage [29]. This gene was primarily expressed in monocytes, with 152 and 26 reads, but it was also at lower levels in B cells (Table 4).

### 2.6. Transcript Levels for a Panel of Cell-Surface Receptors Used as Markers of Immune Cell Populations

The expression levels of the cluster of differentiation proteins (**CDs**) are highly informative concerning the function and regulation of the five cell types, as they were identified on the basis of being lineage-specific (Table 5).

**CD2** acts as a costimulatory molecule on T cells and NK cells. Here, it was only detected in T cells, and at relatively low levels, between 26 and 128 reads. A CD2-related molecule, **SLAMF6**, was detected only in B and T cells at a low level, between 18 and 47 reads (Table 5). **CD5**, **CD6**, and **CD7** were also expressed essentially only in T cells, and similar expression levels were observed in both CD4^+^ and CD8^+^ T cells (Table 5). However, none of the three were entirely T-cell-specific, as very low levels of **CD5** and **CD6** were also seen in B cells. Very low levels of **CD7** were instead observed in monocytes and MCs, within the range of 4 to 29 reads (Table 5). **CD5** was present in 20–30 times lower levels in B cells compared to T cells (Table 5). **CD5** is expressed on B1-B cells in mice, and these cells normally reside in the peritoneum, which indicates that the majority of circulating B cells belong to the B2 type. However, the expression of **CD5** on human B1 cells is less restrictive and can therefore not be used as a specific marker for B1 cells in humans [30]. **CD6**, which has been found to be important for the continued activation of T cells, was almost T-cell-specific, with only 10 and 6 reads in B cells, compared to 468, 371, 261, and 407 in the T-cell samples (Table 5).

All four **CD3** components, **CD3D**, **CD3E**, **CD3G**, and **CD247 (CD3Z)**, which are key components of the T-cell receptor complex, were found only in T cells, as expected (Table 5). The expression levels of these components were almost identical between CD4^+^ and CD8^+^ T cells, except for **CD3E**, which showed much higher levels (Table 5). **CD3E** forms heterodimers with both **CD3G** and **CD3D** and should thereby be present at a concentration twice that of these other two CD3 members, making the higher expression of **CD3E** logical. **CD8A** and **CD8B** were also as expected; they were only expressed by CD8^+^ T cells, with between 332 and 1439 reads (Table 5).

As expected, **CD4** was highly expressed by CD4^+^ T cells, with 501 and 526 reads, but it was also high in both monocytes and MCs. Interestingly, in both monocytes and MCs, the expression level of this marker was as high as almost 50% of the levels in CD4^+^ T cells (Table 5). Notably, the expression of **CD4** was not downregulated upon the culturing of MCs but instead possibly even slightly increased ([3] and Appendix A).

Concerning the T-cell regulatory components **CD28**, **CTLA4,** and **PD1**, we found that all three of them were T-cell-specific (Table 5). The expression of **CD28** was approximately two times higher in CD4^+^ compared to CD8^+^ T cells, with 158 and 198 reads in CD4^+^ T cells and 66 and 94 reads in CD8^+^ T cells (Table 5). The difference in the expression of **CTLA4** was even greater, as it was 5–10 times lower in CD8^+^ than in CD4^+^ T cells, and the absolute levels of **CTLA4** were much lower than for **CD28**, between 19 and 23 reads, which is expected as it is a negative regulator of T-cell activation and is upregulated after T-cell activation (Table 5). When it comes to the **B7:1** and **B7:2** molecules, the ligands of **CD28** and **CLTA4**, which are crucial for T-cell activation, we found that **B7:1** (**CD80**) was primarily expressed in B cells, with 14 and 21 reads, and very low levels in MCs, with 3 and 9 reads (Table 5). In contrast, **B7:2** (**CD86**) was expressed in B cells and monocytes (Table 5). The expression levels were relatively low for both **B7:1** and **B7:2**, in the range of 15–30 reads (Table 5). A third member of this family, **B7-H3** (**CD276**), was only found in MCs, and this expression was maintained after culturing for 3 weeks, with 17 and 32 reads directly after purification and 34, 41, 32, and 44 reads after 3 weeks in culture (Table 5) ([3] and Appendix A). Similar to **PDL1**, this molecule may act as a negative regulator of adaptive immunity, a checkpoint inhibitor [31]. A **CD28**-related molecule that is present in activated T cells and thought to be primarily involved in Th2 T-cell activation, **ICOS**, was essentially only expressed by T cells and was higher in CD4^+^ T cells than CD8^+^ T cells (Table 5). Very variable levels were found in the CD8^+^ T cells, with one individual having 28 reads and the other only 3, whereas in CD4^+^ T cells, there were 51 and 50 reads in the two individuals (Table 5). The binding partner to **ICOS**, **ICOSLG**, was found to be highly expressed by B cells, with 372 and 533 reads, slightly lower in monocytes, and at low levels in both T cells and MCs (Table 5). The regulator of T-cell activation, **PD1**, was expressed only by T cells and at very low levels (Table 5). Interestingly, the binding partner **PDL1** was essentially only expressed by MCs, except for very low levels in both CD4^+^ and CD8^+^ T cells (Table 5). **PDL1** levels dropped after culturing the MCs, almost 10-fold from 106 and 158 reads to 15, 10, 17, and 17 reads in four individual cultures ([3] and Appendix A).

Except for very low levels in MCs, no expression of **PDL2** (**PDCD1LG2**) was observed in the five cell populations (Table 5). **CD27** is another molecule that may act as an immune checkpoint receptor by binding to **CD70**. **CD27** may also act in the maintenance of T-cell memory and regulate B-cell activation and immunoglobulin synthesis [32,33]. It was found to be expressed by B cells, with 45 and 35 reads, and at a higher level by T cells, with between 106 and 541 reads, but not by monocytes and MCs (Table 5).

When it comes to the **CD40-CD40L** pair of receptors so important for B-cell–T-cell interaction, we found the expression of **CD40** essentially only in the B cells, with 120 and 164 reads. **CD40L** was instead only detected in T cells (Table 5). The levels of the **CD40L** were 3–4 times higher in CD4^+^ compared to CD8^+^ T cells, with 68 and 84 reads in CD4^+^ T cells, which makes sense as it is primarily the CD4^+^ T cells that are considered to interact with B cells (Table 5).

A number of other CD molecules were preferentially expressed in non-T-cell populations. One of them, **CD9**, a member of the tetraspanin 4 family, was very highly expressed in MCs, with 931 and 1078 reads. The cross-linking of **CD9** on MCs has been shown to trigger granule release and act as an IL-16 receptor in the absence of **CD4** [34,35]. In the other cells, the expression of **CD9** was 100–1000-fold lower (Table 5). This suggests that the expression level of **CD9** is much higher in MCs compared to the other immune cells, and this is in spite of the published work on the role of CD9 in the activation of several other hematopoietic cell types [36]. The expression of **CD9** by these human MCs is among the highest of all tetraspanin members in human MCs, which indicates a major difference between mouse and human MCs, as deduced from the mouse data presented by the ImmGen consortium [37].

**CD14** was expressed almost exclusively by monocytes, with 1304 and 1388 reads in the two individuals (Table 5). A very low level was also seen in MCs, with between 22 and 26 reads. This expression was also stable after in vitro culture, with 9, 36, 21, and 27 reads in the four cultures ([3] and Appendix A). However, as can later be seen from the expression levels of **TLR4** and also **CD19** in Table 5, with 11 and 16 reads on B cells, even a very low level of expression can be of major biological significance. Low transcript levels have also previously been observed for **TLR**s with two different sequence methods, RNA seq and Ampliseg, but also, much higher levels have been presented in other studies, indicating that the question of the expression levels of **TLR**s and other pattern recognition receptors is still not fully resolved [38,39].

**CD36** was also almost monocyte-specific, with 117 and 138 reads and only very low levels in MCs. **CD36** acts as a scavenger receptor of importance for phagocytosis in monocytes and macrophages but also for the expansion of hematopoietic stem cells during infection [40,41].

Then, we looked at a number of classical B-cell markers, including **CD19**, **CD20**, and **CD21**. Most of them were only expressed by B cells. **CD19,** the surface marker used for B-cell isolation, was expressed only in B cells, as expected, but at a very low level, with between 11 and 16 reads (Table 5). However, the expression level was clearly sufficient for the antibody-mediated isolation of these cells. **CD20** (**MS4A1**) was also only expressed by B cells but at a much higher level, with 214 and 359 reads (Table 5). The knockout of **CD20** in mice results in decreased humoral immunity in both T-cell-dependent and T-cell-independent responses [42]. **CD21**, the complement receptor 2 (**CR2**), was primarily expressed in B cells, with 40 and 40 reads in the two individuals (Table 5). In contrast, very low levels of **CR2** were seen in both CD4^+^ and CD8^+^ T-cells, and no reads were found in monocytes and MCs (Table 5). **CD22**, also named **Siglec-2**, a member of a sialic acid-binding family of proteins, was primarily expressed by B cells but also by MCs, and it was actually relatively highly upregulated in MCs when put in culture. We found 737 and 655 reads for B cells and 146 and 110 reads for freshly isolated MCs, which then increased 4–5-fold after 3 weeks of in vitro culture ([3] and Appendix A). **CD22** functions as an inhibitory receptor for B-cell receptor signaling and is also involved in B-cell trafficking in Peyer’s patches in mice [43].

**CD53** is a member of the transmembrane 4 superfamily though to be involved in enhancing signaling by **CD2** in T and NK cells. However, we only detected the expression of **CD53** in B cells, and also at a relatively low level, with 18 and 36 reads (Table 5). The marker of immature hematopoietic cells, **CD34**, was not detected at significant levels in any of the five cell populations (Table 5).

The majority of another set of B-cell-related genes were exclusively, or almost exclusively, expressed by B cells. **CD72** is a regulatory B-cell receptor interacting with **CD5** that seems to function as a negative regulator of B-cell activation [44]. The expression level of this receptor was low, with 42 and 53 reads, and almost absent in the other cell types (Table 5). **CD79A** and **CD79B** are the two signaling components of the membrane-bound immunoglobulin, the Igα/Igβ heterodimer. Interestingly, we observed a very large difference in expression levels of these two components. **CD79A** had 3902 and 4312 reads in B cells and **CD79B**, at a level of only 10% of the levels of **CD79A** (Table 5). Low levels of these coreceptors were found in all the other cell types, with a maximum of 33 reads in the CD8^+^ T cells for **CD79A** (Table 5). The question here is whether these two signaling components of the B-cell receptor also have a function in T cells.

In these mature B cells, we did not detect any transcripts for one of the pre-B-cell surrogate light chains, the variable region **VPREB1**, also named **CD179a** (Table 5). The corresponding constant domain (lambda 5), encoded by the ***IGLL1*** gene, was not detected in the Ampliseq library, so we cannot say anything about its expression levels, although we expect also this gene to be silent. These two molecules form the surrogate light chain on pre-B cells before the rearrangement of the immunoglobulin light chains, kappa or lambda. Instead of these pre-B cell receptor light-chain transcripts, we found relatively high levels, with 207 and 686 reads for the lambda-like light chain 5, a transcript encoding a lambda constant domain and a joining region connected to a N-terminal region not related to a classical variable region, **IGLL5** (NCBI). This gene encodes a protein with still unknown function. We also detected transcripts for the ***VPREB3*** gene, which encodes a protein that is not part of the pre-B-cell receptor but seems to have a function in the transport of the IgM heavy chain [45]. This gene was still active in these mature B cells with expression levels of 135 and 162 reads (Table 5). The gene encoding the immunoglobulin J chain of pentameric IgM and dimeric IgA (**IGJ**) was expressed exclusively in B cells, with 206 and 270 reads (Table 4). The ***MZB1*** gene encodes a marginal zone and B1-B-cell-specific transcript and the 51 and 66 reads found for this gene in the **CD19**-positive cells indicate the presence of a small number of B1-B cells among these peripheral B cells [46]. FACS analyses using **CD20**, **CD27**, and **CD43** positivity as B1 selective markers have exhibited highly variable numbers of B1 cells in human peripheral blood, ranging from approximately 1 to 15% of the circulating B cells [47].

Several of the five cell types expressed the following CD molecules: **CD63**, **CD68**, **CD69**, **CD81**, **CD82**, **CD83**, **CD96**, **CD97**. By contrast, a few CD molecules, including **CD163**, **CD180**, **CD200**, and **CD248**, were expressed at low levels in only one of the five populations. **CD63** is a member of the transmembrane 4 superfamily, together with **CD9**, **CD37**, **CD53**, **CD81**, **CD82**, and **CD151** [35,48]. **CD63** is associated with intracellular vesicles and is used as a marker in the basophil activation test (**BAT**) [49]. **CD63** was found to be expressed in all five cell types but at much higher levels in monocytes and MCs (Table 5). It was also upregulated in MCs upon in vitro culture, from 265 to 305 reads in freshly isolated cells to 1274, 1264, 1255, and 1992 reads in the four cultures ([3] and Appendix A). **CD68** has been used as a marker for macrophage and monocyte populations, and we found high levels of transcripts for this heavily glycosylated protein in monocytes but also in MCs (Table 5). **CD69** is a C-type lectin and an activation marker for several hematopoietic cells [50,51]. It was highly expressed in four out of the five cell populations in this analysis, i.e., in B cells, CD4^+^ and CD8^+^ T cells, and MCs but not in monocytes (Table 5). It seems remarkable that an activation marker also shows such a high level in nonactivated cells. **CD81** and **CD82** are members of the transmembrane 4 superfamily. **CD81** was found to be highly expressed in all cells except monocytes where the levels were considerably lower (Table 5). **CD81** may have a role in T-cell activation by associating with CD4 and CD8 to provide costimulatory signals with **CD3**, but as shown here, **CD81** was more highly expressed by MCs, and it was also highly expressed by B cells, which suggests additional functions in these cells as well [52]. **CD82** was also expressed by all five cell populations but was much lower in monocytes and T cells. The exact function of this molecule is not known. **CD83** seems to have diverse functions in several hematopoietic cells, but in our analysis, it was primarily expressed by B cells and MCs, with very high levels in MCs, with 1555 and 969 reads. **CD96** is a member of the immunoglobulin superfamily that appears to have a role in adhesive interaction by activated T and NK cells [53]. Relatively high levels, with between 58 and 251 reads, were observed in T cells. **CD97** belongs to the adhesion G protein-coupled receptor (**GPCR**) family and binds to the complement control protein decay accelerating factor (**DAF**), also named **CD55**. This cell surface protein has been found to have a critical role in host defense by regulating granulocyte homeostasis, but other functions have also been described [54]. Here, **CD97** was expressed by all five cell types but was higher in T cells and MCs. **CD163** was only expressed by monocytes in our study at relatively low levels, with 37 and 48 reads. **CD163** is a high-affinity scavenger receptor for hemoglobin–haptoglobin complexes and is also a sensor for both Gram-positive and Gram-negative bacteria on monocytes–macrophages [55,56]. **CD180** was primarily expressed by B cells in this study, with 40 and 44 reads (Table 5). **CD180** is a leucine-rich repeat (**LRR**)-containing surface protein that interacts with **MD-1**, which acts together with Toll-like receptor 4 (**TLR-4**) in the sensing of bacterial lipopolysaccharides (**LPSs**) from Gram-negative bacteria [57,58]. **MD-1,** also named **LY86**, was expressed by B cells and monocytes only, with between 62 and 143 reads. In this analysis, **CD200** was B-cell-specific, but at a low level, with 44 and 63 reads. **CD200** may have an immunosuppressive function, which still seems poorly defined [59]. It is claimed to be expressed in other immune cells, which we did not see in this analysis, except for very low levels in CD4^+^ T cells and MCs (Table 5) [59]. **CD248** showed an interesting pattern as we almost only found transcripts for this protein in CD8^+^ T cells and very low levels in the other cells (Table 5). **CD248** belongs to a novel family of C-type lectins with yet an unknown function [60].

As the last set of genes in this section, we have members of the **CD300** family. Members of this family appear to have either inhibiting or activating functions on various immune cells. All of them were expressed in monocytes, and several almost exclusively in monocytes, including **CD300E**, **CD300LB**, and **CD300LF** (Table 5). **CD300A** has been shown to downregulate the cytolytic activity of NK cells and inhibit MC degranulation [61,62]. In contrast to the above-listed members, **CD300A** was more broadly expressed and found in monocytes, T cells, and MCs but not B cells (Table 5). **CD300C**, which is a paralog of **CD300A**, with a less defined role in immunity, was only expressed in monocytes and MCs (Table 5). Both **CD300E** and **CD300LB** seem to act as activating receptors on myeloid cells, whereas **CD300LF** has the opposite role, being an inhibitory receptor [63].

Most of the CD markers presented in this table adhere to the existing view of their expression profile and role in immunity, but we can now add quantitative values and also clarify several inconsistencies concerning their expression in freshly isolated normal nonactivated cells.

### 2.7. Transcript Levels for the Major Histocompatibility-Related Genes (MHCs)

Members of the MHC Class I and II genes and related genes are involved in antigen presentation. We found that the different Class II genes are primarily expressed by B cells and monocytes at very high expression levels, with between 956 and 3761 reads for the **DR** and **DP** genes in monocytes, and between 2055 and 6459 reads for B cells. This indicates a 2–5 times higher expression level in B cells compared to monocytes and an almost 100-fold higher level in B cells compared to T cells and MCs (Table 6). Of the **DQ** genes, only low levels of **DQA1** were detected in B cells, with a high variation (4 and 458 reads) between the two individuals. No transcripts for the two other **DQ** genes, **DQA2** and **DQB2**, were detected in the five cell types. Of the proteins needed for the transcriptional activation of Class II genes and for antigen presentation (**CIITA**, **HLA-DMA**, **HLA-DOA**, **HLA-DOB**, and **CD74**), the transcriptional coactivator **CIITA** levels matched the expression levels for Class II genes, with a 2–3 fold higher level of **CIITA** in B cells compared to monocytes, and almost 100-fold higher in B cells compared to the levels in T cells and MCs. **HLA-DMA** plays a role in the peptide loading of Class II molecules and was here expressed at almost equal levels in B cells and monocytes and approximately 20 times lower levels in T cells and MCs. **HLA-DOA** and **DOB** form a heterodimer and have been found to participate in antigen loading on Class II genes primarily in B cells; here, we also found them expressed almost exclusively in B cells, except for low levels of **DOA**, also in CD8^+^ T cells. **CD74** is the invariant chain that blocks peptide binding to Class II molecules before entering the endosomal compartment for antigen loading. We found that **CD74** was expressed at approximately 5–6 times higher levels in B cells compared to monocytes and 10 times higher than in T cells and MCs.

Of the MHC Class I-related **CD1** molecules, only **CD1D** was expressed at significant levels in these cells. The expression was much higher in monocytes than in B cells, with 441 and 466 reads in monocytes and 15 and 40 reads in B cells. No expression of any of the four chains was observed in T cells or MCs (Table 6). Only very low levels of **CD1A** and **CD1C**, between 1 and 7 reads, were observed in B cells and monocytes, and no expression of **CD1B** was found in any of the five cell types.

In contrast to **CD1** molecules, the classical Class I genes were expressed at very high levels in all five cell types. For example, the expression of β2-microglobulin (**B2M**), the binding partner to the alpha chain of Class I genes, varied between 2218 and 11,150 reads in these five cell types, with the highest level found in the T cells (Table 6). Of the MHC Class I alpha-chain genes, a high level of expression of **HLA-A** was detected in all five cell types. However, 2–15 times lower levels of Class I **A**, **B**, and **C** genes were observed in MCs compared to the four blood cell populations, and there was also a major variation in expression levels between individuals for the three Class I genes (Table 6). One individual almost totally lacked expression of **HLA-B** and **C**, which may have major implications for defense against viral infections (Table 6). However, the two individuals that were the origin of the four blood cell populations were ordinary anonymous blood donors, indicating that they had no major health issues. This may indicate that a highly limited Class I repertoire is consistent with a normal life. The levels of **HLA-B** in one person were 100–200 times lower than in the second donor, and for the **HLA-C,** the difference was even greater, with one donor having between 514 and 9613 reads, whereas the other had between 2 and 17 reads. However, we could see that the **HLA-A** transcript level was considerably higher in the donor almost lacking **HLA-B** and **C**, possibly as a compensatory mechanism (Table 6).

In contrast, the invariant Class I molecule **HLA-E** was highly expressed in all five cell populations. **HLA-E** is a non-classical MHC Class I molecule with limited polymorphism of importance for inhibiting NK-cell lysis by binding to the inhibitory receptor **CD94/NKG2C** [64].

### 2.8. Transcript Levels for Fc Receptors

Receptors with specificity to the constant domains of the various immunoglobulins, the Fc receptors, differ markedly between different immune cell populations. Most of them act as receptors for the uptake, i.e., phagocytosis, of immune complexes, or as for the high-affinity IgE- receptor, in the activation and degranulation of the cell. In our data, two of the three subunits of the high-affinity IgE receptor FcεRI, the alpha and beta chains (**FCERIA** and **MS4A2**), were almost exclusively expressed by MCs, with 214 and 127 reads for the alpha chain (**FCERIA**) and 817 and 441 reads for the beta chain (**MS4A2**) (Table 7). A very low level of the alpha chain was also seen in monocytes, with eight and five reads, almost 20 times lower than in MCs, but no expression of the beta chain was observed, which is in line with previous findings indicating that monocytes and dendritic cells can express a variant of the high-affinity IgE receptor consisting only of the alpha and gamma chains and lacking the beta chain (Table 7) [65]. The IgE receptor gamma chain **FCERIG** is also a signaling component of two of the IgG-Fc receptors, the high-affinity receptor **FCGR1** and the low-affinity receptor **FCGR3 [66]**. The gamma chain was expressed at high levels in monocytes and MCs, with between 741 and 910 reads, and at low levels in T cells, with between 7 and 57 reads, but not in B cells (Table 7). In contrast, the low-affinity IgE receptor **FCER2**, also named **CD23**, which is a C-type lectin, was found almost exclusively in B cells, with between 598 and 643 reads, and at a 100-fold lower level in monocytes (Table 7).

When it comes to the different receptors for IgG, we saw that many of them were expressed by monocytes, including the high-affinity IgG receptor **FCGRIA**; the three variants of the intermediate-to-low-affinity receptor, **FCGR2A**, **FCGR2B**, and **FCGR2C**; and one of the other low-to-medium affinity receptors **FCGR3A**, at very low levels (Table 7). Here, the high-affinity IgG receptor **FCGR1A** was actually only found in monocytes and at a relatively low level, with 13 and 15 reads. The activating gamma 2 receptor **FCGR2A** was expressed by both monocytes, with 61 and 66 reads, and MCs, with 142 and 199 reads. This is the only IgG receptor we detected in MCs in contrast to the literature, where other receptors have been detected. The reason for this discrepancy may be due to the different model systems used or the states of activation, as we have seen that the transcript levels for these receptors can change quite markedly upon in vitro culture or by analysis of cell lines. For example, three weeks of culturing freshly isolated human skin MCs resulted in a 5-fold increase in **FCERIA** and a 90% reduction in **FCGR2A** levels [3]. The inhibitory receptor **FCGR2B** was expressed in both B cells and monocytes, and for monocytes, we saw that the expression levels for both this receptor and the related activating receptor **FCGR2C** differed markedly between individuals. We saw an approximately 5 times difference in expression levels for both of these receptors between the two individuals in this study, with 38 reads for both in one individual and 188 and 190 reads for the second individual. Interestingly, no differences in expression levels for these two receptors were seen in B cells, indicating differences in the regulation of these two receptors between these cell types.

In contrast to the other receptors, **FCGR3A** was here almost exclusively expressed by CD8^+^ T cells, with 58 and 65 reads. Only low levels, with six and eight reads, were observed in monocytes, and extremely low levels were observed in CD4-positive cells, with one and one read (Table 7). The closely related gene ***FCGR3B*** was not detected in any of the five cell types. This receptor seems to be mainly expressed by neutrophils, in which it is one of the most abundant Fc receptors [67].

The large subunit of **FCRn**, the neonatal IgG receptor, is encoded by the ***FCGRT*** gene. This receptor is closely related to MHC Class I alpha chains and was here expressed by all five cell types. However, monocytes showed approximately 10 times higher levels than the other four cell types, with 135 and 235 reads in monocytes. FCRn is involved in the recycling of IgG and albumin by endothelial cells [68].

***FAIM3*** encodes the IgM-specific receptor that has been found to be primarily expressed by B and T cells, which fits very nicely with the data here [69,70]. Both B and T cells expressed relatively high levels of this receptor, within a range of 199 to 468 reads (Table 7).

During the analysis of the total human genome, a number of genes related in structure to the classical IgG and IgE receptors were identified, the Fc receptor-like molecules. Six such genes were identified, and the majority of them were expressed almost exclusively in B cells, including ***FCRLA***, ***FCRL1***, ***FCRL2***, and ***FCRL5*** (Table 7). One of them, ***FCRL3***, was expressed in both B cells and CD8^+^ T cells, with between 16 and 66 reads and at very low levels in CD4^+^ T cells. The remaining receptor number 6, ***FCRL6***, was found to be expressed exclusively by CD8^+^ T cells, with 72 and 126 reads. ***FCRL6*** has previously been shown to be expressed by CD8^+^ T cells and NK cells and to be involved in inhibiting or controlling the activation of these cells by binding to MHC Class II DR [71]. Transcripts for one of the **FCRL** genes, ***FCRLB***, were almost totally absent in all five cell types. The **FCRL** receptors are most likely ancestors of the classical IgG and IgE receptors and are found earlier in evolution before the appearance of the classical IgG and IgE receptors [66]. Relatively little is still known about their function but some results indicate the role of **FCRL1** as a coactivating receptor for B cells and that **FCRL3** and **4** bind IgA and **FCRL5** to IgG [72]. All of the classical Fc receptors and Fc-receptor-like receptors in the human genome are encoded from a single locus on chromosome 1, indicating that they have appeared by successive gene duplications of one or a few ancestral genes [66].

The ***MILR1*** gene encodes a receptor that has been shown to act as a negative regulator of MC activation. Here, it was expressed at relatively low levels in B cells, monocytes, and MCs, in the range of 30–60 reads (Table 7) [73]. **CD200R1** is an inhibitory receptor with two immunoglobulin domains that has been shown to be primarily expressed by myeloid cells; it is involved in downregulating the expression of inflammatory cytokines and is also able to inhibit MC degranulation [59,74]. Here, this gene was expressed almost exclusively by MCs but at a very low level, with 24 and 6 reads (Table 7). These two latter genes are not classical Fc receptors but were included in this list as they have interesting immune regulatory functions.

Another class of receptors, the **MRGPRX** family of receptors, has received a lot of attention due to the fact that one of its members has activating functions on MCs similar to the high-affinity IgE receptor. **MRGPRX2** was the only one of the four family members that was expressed in any of these five cell types and was found in MCs, with 253 and 365 reads. **MRGPRX2** is an activating receptor that has been shown to be expressed almost exclusively on one type of MCs, the connective tissue-type MCs [75]. **MRGPRX2** is also not an Fc receptor but a receptor for substance P and other positively charged low molecular weight compounds but was included in this table due to its MC-activating properties similar to the high-affinity IgE receptor [75].

### 2.9. Transcript Levels for Leukocyte Immunoglobulin-like Receptors (LILRs) and Killer Cell Lectin-like Receptors (KLRs)

Leukocyte immunoglobulin-like receptors, **LILR**s, is a family of immunoreceptors that, according to the literature, are primarily expressed by myeloid antigen-presenting cells such as monocytes and dendritic cells, but also by granulocytes, NK cells, T and B lymphocytes, and hematopoietic stem cells, as well as by non-immune cells, such as endothelial cells and neurons [76]. They are distantly related in structure to Fc receptors and are not encoded on chromosome 1, where the classical Fc receptors are found but on chromosome 19 [66]. The **A family** members of the **LILR**s contain short cytoplasmic domains and may primarily be activating receptors, whereas all of the **B family** members are inhibitory [76]. The inhibitory **LILR**s constitute a safeguard system that mitigates the inflammatory response, allowing for a prompt return to immune homeostasis [77]. The majority of the **LILRs** listed here were almost exclusively expressed by monocytes (Table 8). One of them, **LILB3**, reached as high as 500 reads in one of the samples. Only one of the eight members we have listed here was also expressed by another cell population, and that is **LILRB1**, which was also expressed by B cells, but at a relatively low level, with 34 and 37 reads. A few additional **LILRs** were also expressed at very low levels in MCs, generally in the range of a few reads (Table 8). **LILRA1** is an activating receptor that interacts with MHC Class I molecules [76]. **LILRA2** has been found to sense microbially cleaved immunoglobulins to activate myeloid cells [78]. Both **LILRA3** and **LILRA6** may act as receptors for MHC Class I molecules similar to **LILRB1**, **LILRB2**, **LILRB3,** and **LILRB4**, but in contrast to the previous ones, they act in an inhibitory fashion in various immune cells [76]. **LILRA1** and **LILRA2** were both only detected at low levels in monocytes. Here, both **LILRA3** and **LILR6** were also expressed at relatively low levels in monocytes, with 78 and 34 reads and 141 and 128 reads, respectively. The osteoclast-associated receptor (**OSCAR)** was also a member of the leukocyte receptor complex, together with **LILRs,** and is thought to be involved in the regulation of both innate and adaptive immunity [79]. Among these five cell types, **OSCAR** was expressed exclusively by monocytes, with 327 and 284 reads (Table 8).

In contrast to **LILRs**, the killer lectin-like receptors belong to the C-type lectin family, and they were not expressed by monocytes. **KLRB1**, **KLRC3**, **KLRC4**, **KLRG1,** and **KLRK1** were instead all expressed by T cells (Table 8). Among these five cell populations, here, three of them were expressed in T cells only, namely **KLRB1**, **KLRC3**, and **KLRC4**, and two of them were observed only in CD8^+^ T cells (**KLRC3** and **KLRC4**) but at relatively low levels (Table 8). One of them, **KLRG1**, was expressed both in T cells and MCs, and one (**KLRG1**) was identified in CD8^+^ T cells and B cells (Table 8). **NKG2D**, which is encoded by the ***KLRK1*** gene, recognizes induced self-proteins, which appear on stressed, malignant, transformed, and infected cells [80]. In this study, this receptor was found to be highly expressed by CD8^+^ T cells, with 641 and 424 reads, and at low levels in B cells (Table 8). **KLRB1** binds terminal Gal-alpha (1,3) epitopes and N-acetyl-lactose amine epitopes and inhibits NK-cell activation [81]. This lectin was relatively highly expressed by both CD4^+^ and CD8^+^ T cells, with 504, 245, 420, and 183 reads (Table 8). **KLRC3** is a receptor for the recognition of **HLA-E** in NK cells and some CD8^+^ T cells [82]. In this analysis, it was found only in the CD8^+^ T-cell sample and at relatively low levels (Table 8). **KLRC4** was also found to be expressed only in the CD8^+^ T cells, with 71 and 23 reads (Table 8). **KLRG1** plays an inhibitory role on NK and CD8^+^ T cells by binding non-MHC ligands, including conserved sites, on cadherins and may mediate missing self [83]. In this study, it was expressed by both CD4^+^ and CD8^+^ T cells and by MCs (Table 8). In this category, we also have two additional membrane proteins: **NKG7** and **NCR3**. **NKG7** is a small membrane protein essential for the degranulation of NK cells and CD8^+^ T cells and is also important for the activation of CD4^+^ T cells [84]. **NKG7** was very highly expressed by CD8^+^ T cells, with 1423 and 1129 reads, and was at much lower levels in CD4^+^ T cells and even lower in monocytes (Table 8). Natural cytotoxicity triggering receptor 3 (**NCR3)** interacts with a broad range of ligands without an obvious structural similarity, including viral, parasitic, and tumor proteins, and may aid NK-cell lysis of target cells and cytokine secretion by interacting with the T-cell receptor zeta chain [85,86]. It was expressed at low levels in both CD4^+^ and CD8^+^ T cells and B cells but not in monocytes or MCs (Table 8).

### 2.10. Transcript Levels for Complement Components and Receptors

The majority of complement and coagulation components are produced by the hepatocytes of the liver. However, recently, we have shown that also human monocytes and mouse macrophages are major producers of several of these components [16,38]. **C1Q** consists of three chains, **A**, **B**, and **C**, and it was expressed in relatively large amounts by mouse peritoneal macrophages [38]. However, they were not expressed at significant levels in these five cell populations in this study, except for **C1QA**, which was detected at very low levels in monocytes, with 20 and 15 reads (Table 9). Transcripts of the **C2** component were identified in very low numbers in both monocytes and MCs, with 5 and 8 and 13 and 12 reads, respectively (Table 9). The expression of the **C2** component in MCs is in line with what we observed in mouse MCs, although the levels of this component were higher in the murine counterparts [18]. In contrast to **C1Q** and **C2**, three other complement components, namely transcripts for factor P or properdin, **CFP**; complement factor D, **CFD**; and ficolin 1, **FCN1**, were produced in very high amounts in the human monocytes (Table 9). **CFP** was also produced in lower amounts in T cells and **CFD** in MCs (Table 9). **CFD** is one of the activating serine proteases of the complement cascade. None of the other complement components were expressed in any significant levels except for very low levels of the **C3** component by MCs, with eight and seven reads, and very low levels of factor H, **CFH**, by T cells, with four and nine reads (Table 9).

**C3AR1** is the receptor for one of the complement components, the small C3a fragment. This C3 fragment, which is chemotactic and activating on MC, was expressed almost exclusively by MCs but at a low level, with 6 and 31 reads, and at a very low level by T cells, with 1 and 4 reads (Table 9). Low levels of the protein C receptor (**PROCR**) were observed in MCs (Table 9). The two complement receptors **CR1** (**CD35**) and **CR2** (**CD21**) were both primarily expressed by B cells and at relatively low levels. However, both were also expressed at very low levels in T cells (Table 9). This is in line with the important role of these receptors primarily in B-cell biology, where in human B cells, they act as negative regulators of both proliferation and antibody production [87].

### 2.11. Transcript Levels for Toll-like Receptors (TLRs) and Other Pattern Recognition Receptors

Pattern recognition receptors are essential for the recognition of non-self and the initiation of protective inflammatory responses. They originate from a number of different gene families. The most well-known is likely the toll-like receptors (**TLR**s), which is a small family of leucine-rich repeat-containing proteins, which in the human genome, consist of 10 members, **TLR1**-**TLR10**. Interestingly, despite their essential function in immunity, they were expressed at very low levels in both mouse and human immune cells, with no more than 32 reads detected for any of these ten members and in any of the five cell types in the present analysis (Table 10) [18,38]. Low levels were also seen for other pattern recognition receptors such as the **NOD**, **RIG**, and **MDA5.** Still, the maximum level detected was 43 reads, indicating that these low levels of transcription are still sufficient for effective sensing of non-self structures such as LPSs, peptidoglycans, double-stranded RNA, and non-methylated DNA. These receptors were expressed in a cell-specific pattern in the five cell types, suggesting that a certain cell type only responds to some of these non-self structures. We found that B cells expressed **TLR1**, **6**, **7**, **9**, and very low levels of **TLR10**; monocytes expressed **TLR1**, **2**, **4**, **5**, **8**, and **9**; T cells expressed **TLR1** and **5**; and MCs expressed only **TLR4** and very low levels of **TLR2** (Table 10). **NOD1** was more broadly expressed but at low levels. **NOD2** was almost only expressed by monocytes and at slightly higher levels, with 29 and 43 reads (Table 10). **NLRC3** is a short NOD-like receptor, also named **NOD3**, involved in maintaining T-cell activation and preventing T-cell anergy [88]. It was here expressed almost exclusively in lymphocytes and at low levels, between 19 and 45 reads in these three cell types. **DDX58**, also named **RIG-1**, is a pattern recognition protein that recognizes short double-stranded RNA, a characteristic feature of different virus infections [89]. Here, it was expressed in all five cell types, with between 10 and 39 reads, except for monocytes, where the expression was lower, with 5 and 5 reads. **IFIH1**, also named **MDA5**, which in contrast to **RIG-1** bind larger doubled-stranded RNA, with a length of 2000 nucleotides or more, but is also able to detect the lack of 2’-O-methylation in RNA [90,91]. Here, **MDA5** was expressed at very low levels, with between 1 and 18 reads, and it was particularly low in monocytes, with only 1 and 2 reads. As mentioned previously, we have recently shown that LPSs, which are detected by **TLR4** in monocytes, induce a rapid and extremely potent response in freshly isolated human monocytes, with a 75,000 times induction of IL-6 within 4 h of incubation, and they also induce IL-8, thus becoming the most highly expressed protein [16]. **CD14** acts as a coreceptor for LPS recognition by interacting with **TLR4**, and here, we found a remarkable difference in expression levels between these two components of the LPS recognition machinery as the level of **TLR4** in monocytes was 6 and 6 reads and 1304 and 1388 reads for CD14 (Table 5).

FAS-associated death domain protein is encoded by the **FADD** gene. This protein is an adaptor protein for the FAS receptor and procaspases 8 and 10 to form the death-inducing signaling complex during apoptosis [92]. **FADD** is also required for an efficient antiviral response. Upon viral infection, **FADD** is needed to increase the levels of interferon regulatory factor, **IRF7**, a molecule needed for the production of IFN-alpha [93]. Here, **FADD** was expressed in all five cell types but was higher in T cells, with between 23 and 26 reads. Protein kinase R is encoded by the ***EIF2AK2*** gene. This is a kinase activated by double-stranded RNA and induced during cellular stress and by viral infections [94]. It was expressed by all five cell types at higher levels than the other proteins involved in sensing infection, with between 27 and 121 reads. Mitochondrial antiviral-signaling protein ***MAVS*** is the gene for one of the proteins in these lists that show relatively similar expression levels in all five cell types, with between 11 and 35 reads. The protein encoded by this gene is essential for antiviral innate immunity located in the outer membrane of mitochondria, peroxisomes, and mitochondrial-associated ER membranes [95]. **MYD88** acts as an essential signaling adaptor protein to **TLR**s [96]. It was expressed in all five cell types, with between 25 and 100 reads. The protein encoded by the ***PLD4*** gene is a single-stranded acid exonuclease that regulates endosomal nucleic acid sensing, most likely to control **TLR9** responses [97]. It was expressed essentially only in B cells and monocytes, with 83 and 178 reads in B cells and 21 and 23 reads in monocytes. C-type lectin domain family 7 member A or **Dectin-1** is encoded by the ***CLEC7A*** gene. **Dectin-1** is a C-type lectin and pattern recognition receptor that recognizes β-glucans and carbohydrates found in fungal-cell walls, some bacteria, and plants, and has been shown to be expressed by myeloid dendritic cells, monocytes, macrophages and B cells [98]. In our study, ***CLEC7A*** was expressed exclusively by monocytes at a relatively low level of expression, with 30 and 41 reads, but not by B cells.

The formyl peptide receptor **FPR1**, recognizing bacterial N-terminal peptides, **FMLP**, formyl–methionine–leucine–phenylalanine, is involved in sensing the presence of bacteria and was detected only in monocytes, with 48 and 23 reads (Table 10). Interestingly, all of these sensors of non-self were expressed at very low or relatively low levels, indicating that these low levels are sufficient for the strong response seen upon their engagement.

### 2.12. Transcript Levels for Histamine, Leukotriene, and Prostaglandin Synthesis Enzymes

Transcript levels for the enzymes involved in histamine, leukotriene, and prostaglandin production and degradation were generally higher in MCs compared to the other four cell types (Table 11). MCs are known to be initiators of inflammation and potent producers of both histamine and the arachidonic acid metabolites, primarily leukotriene C4 and prostaglandin D2, so the high level of these enzymes primarily in MCs was expected. However, the high selectivity of these enzymes to MCs is still remarkable and shows the major role of MCs as inflammation initiators.

The **HDC** gene encodes the enzyme histidine decarboxylase, an enzyme that removes the carboxyl acid group from the amino acid histidine and thereby generates the highly vasoactive histamine [99]. **HDC** was expressed exclusively by MCs and at high levels, with 853 and 796 reads (Table 11). Two monoamine oxidase genes, **MAOA** and **MAOB**, are genes encoding enzymes involved in the degradation of dopamine, serotonin, and related amines [100]. Both of them were also exclusively expressed by MCs but at very low levels: **MAOA** was detected with 9 and 10 reads and **MAOB** with 105 and 200 reads. Their potential functions in the MC lineage have recently been discussed [101].

Arachidonate 5-lipoxygenase, **ALOX5**, is involved in the early enzymatic steps in the generation of various leukotrienes by adding a hydroperoxyl residue (HO_2_) to arachidonic acid to form 5-HpETE, and in a second step, **ALOX5** can convert 5-HpETE to LTA4 [102]. **ALOX5** was expressed at relatively high levels in B cells and monocytes (250 to 310 reads) and at even higher levels in MCs, with up to 1423 reads (Table 11). In contrast, the coactosin-like protein **COTL1** or **CLP** was more broadly expressed and found at relatively high levels in all five cell types, within the range of 250 to 1132 reads. The protein produced from this gene acts as a stabilizing scaffold for **ALOX5**, hindering its inactivation and thereby promoting its metabolic activity [103]. ALOX5-activating protein or **ALOX5AP** or **FLAP** is an integral protein of the nuclear membrane necessary for the activation of **ALOX5** [104]. It was expressed in all five cell types, within the range of 130 to 312 reads in T cells and MCs and at lower levels in B cells and monocytes. In contrast, leukotriene C4 synthase, **LTC4S**, was almost exclusively expressed by MCs, with 129 and 60 reads, and at very low levels in the other four cell types, not exceeding 4 reads. A similar situation was seen for **HPGD** and **HPGDS**, two enzymes involved in prostaglandin synthesis and inactivation. Both were highly expressed in MCs, with 1296 and 1021 reads for **HPGD** and 705 and 484 reads for **HPGDS**, and they were absent or almost absent in the other cell types. Only low levels of HPGD were seen in T cells within the range of 8 to 10 reads. **HPGD**, or 15-hydroxy prostaglandin dehydrogenase, is an enzyme that catalyzes the first step in the inactivation of prostaglandins [105]. **HPGDS** is instead the enzyme that performs the conversion of PGH2 to PGD2 and thereby has a major role in the formation of PGD2 in MCs [106]. Prostaglandin–endoperoxide synthase 1 or **PTGS1**, also known as cyclooxygenase 1 (**COX-1**), was expressed in MCs, with 378 and 357 reads, which was almost 10 times higher than what was detected in B cells and monocytes, while T cells were negative altogether. **PTGS1** and **PTGS2** (cyclooxygenase 2, COX-2) both catalyze the formation of prostaglandin H2 from arachidonic acid, which is the first step in the synthesis of different prostaglandins [107]. However, in contrast to **PTGS1**, **PTGS2** is inducible and often upregulated during inflammation. We found that **PTGS2** was expressed almost exclusively by MCs and at slightly higher levels than **PTGS1**, with 523 and 392 reads, and only at very low levels in monocytes, with 2 and 4 reads. Phospholipase A2, **PLA2G2A**, is a membrane-associated enzyme involved in releasing arachidonic acid from membrane phospholipids and the first step in the formation of both leukotrienes and prostaglandins [108]. This isoform is a member of one of several structurally unrelated families of PLA2 enzymes, and here, it was expressed exclusively by MCs at relatively high levels, with 272 and 579 reads. Thromboxane A synthase 1, **TBXAS1**, is an endoplasmic reticulum membrane protein that catalyzes the conversion of thromboxane H2 to thromboxane A2 a potent vasoconstrictor and inducer of platelet aggregation [109]. This enzyme was expressed in all five cell types, very low in lymphocytes, within a range of 1 to 13 reads, whereas it was higher in MCs, with 18 and 26 reads, and most highly expressed in monocytes, with 297 and 323 reads. Lysophosphatidylcholine acyltransferase 2, **LPCAT2**, is an enzyme responsible for the generation of the lipid mediator platelet-activating factor (**PAF**). This enzyme plays a key role in macrophage inflammatory gene expression in response to stimulation with bacterial ligands [110]. Here, it was expressed by monocytes and MCs, with 27 and 39 reads for monocytes and 95 and 202 reads for MCs. The ***ENPP2*** gene is encoding autotaxin, an enzyme that is involved in the formation of lysophosphatidic acid (LPA), another lipid mediator [111]. This gene was expressed primarily by MCs, with 86 and 107 reads, and was lower in T cells, with 4 to 18 reads. The ***MBOAT7*** gene encodes the enzyme lysophospholipid acyltransferase 7, an enzyme that is part of the phospholipid remodeling pathway, known as the Land cycle [112]. This enzyme showed a 10-fold higher expression level in monocytes and MCs compared to lymphocytes, within the range of 263 and 332 reads.

Finally, we looked at the expression of annexin A1, **ANXA1**, also known as lipocortin I, a Ca^2+^-dependent phospholipid-binding protein with phospholipase A2 inhibitory activity [113]. **ANXA1** was expressed at a high level in all cell types except B cells and at very high levels in MCs, with 3919 and 5423 reads.

Similar to the observations made for proteases and their inhibitors, here, we found that the controlling proteins were expressed at much higher levels in cell types also producing the mediators they inhibit.

### 2.13. Transcript Levels for Proteoglycan Synthesis and Other Carbohydrate-Related Proteins

A similar situation to histamine and arachidonic acid enzymes was seen in carbohydrate-processing enzymes; here, MCs were also overrepresented, although not to the same extent.

Serglycin, **SRGN**, is the core protein for heparin and chondroitin sulfate synthesis, which is essential for the granule storage of proteases and histamine in MCs [114]. It was expressed in all five cell types, within the range of 690 to 1350 reads in monocytes and T cells and at very high levels in MCs, with 5850 and 7939 reads (Table 12). We found a much lower level in B cells, with 45 and 78 reads. Heparan sulfate glucosamine 3-O-sulfotransferase 1, **HS3ST1**, is a member of the heparan sulfate biosynthetic enzyme family, and here, it was expressed almost exclusively in MCs, with 19 and 46 reads. Another member of this family is heparan sulfate (HS) 6-O-sulfotransferase 1, **HS6ST1**, which catalyzes the transfer of sulfate from 3-prime-phosphoadenonine 5-prime-phosphosulphate to position 6 of the N-sulfoglucosamine residue of heparan sulfate [115]. This gene was expressed at relatively high levels in MCs, with 108 and 293 reads, and had low levels in monocytes and B cells and very low levels in T cells. ***HSPG2*** encodes another core protein known as perlecan, or basement membrane-specific heparan sulfate proteoglycan core protein (**HSPG**) or heparan sulfate proteoglycan 2 (**HSPG2**) [116]. It was expressed almost exclusively in MCs, with 127 and 198 reads. The N-acetylgalactosaminyltransferase 6 gene, ***GALNT6***, encodes an enzyme that initiates mucin-type O-linked glycosylation in the Golgi apparatus [117]. It was most highly expressed in MCs, with 59 and 68 reads, and had lower levels in monocytes and very low levels in B and T lymphocytes. Heparan sulfate N-deacetylase/N-sulfotransferase 1 and 2 (**NDST1** and **NDST2**) are two heparan sulfate modifying enzymes [118,119]. **NDST1** showed the highest expression in MCs, with 54 and 65 reads; it had lower levels in monocytes and very low levels in B cells and was almost totally absent in T cells (Table 12). **NDST2** showed higher expression, with 799 and 183 reads in MCs, 148 and 103 reads in monocytes, 113 and 99 reads in B cells, and 79 and 109 reads in T cells. *N*-acetylglucosamine-6-sulfatase. **GNS** is a lysosomal enzyme found in all cells and is involved in the catabolism of heparin, heparan sulfate, and keratan sulfate [120]. It was here expressed in all five cell types with levels between 9 and 32 reads in all cells, except MCs, where the level was considerably higher, between 135 and 270 reads. A similar picture was seen for the next enzyme in this list, **B4GALT5**, with between 8 and 13 reads in all cells except MCs, where the levels were much higher, between 129 and 255 reads. **B4GALT5** is one out of seven different beta-1,4-galactosyltransferase genes with yet unknown function. **GBE1**, the gene for the 1,4-alpha-glucan-branching enzyme 1, also showed a similar pattern, with low expression in lymphocytes and monocytes and higher in MCs, with 139 and 151 reads. Highest levels of this enzyme have been found in the liver and muscle, and mutations in this gene are associated with glycogen storage disease [121]. The syndecan 3 gene, ***SDC3***, encodes the core protein for a membrane heparan sulfate proteoglycan [122]. This gene was almost exclusively expressed by MCs among these five cell types, with 74 and 84 reads. Exostosin like 3 gene, ***EXTL3***, is a glycosyltransferase involved in heparan sulfate (HS) biosynthesis. Mutations in this gene have been found to cause skeletal dysplasia, immune deficiency, and developmental delay [123]. Here, it was also almost exclusively expressed by MCs, with 25 and 65 reads, and 5 reads or lower in the other four cell types. ***EXT1*** also showed a similar pattern of expression. The protein encoded by this gene is one out of two endoplasmic reticulum-resident type II transmembrane glycotransferases, with the other being **EXT2**. They are involved in the chain elongation step of heparan sulfate biosynthesis [124]. The gene for glycosyltransferase 1 domain-containing protein 1, ***GLT1D1***, was instead almost exclusively expressed by monocytes, with 75 and 103 reads. N-acylglucosamine 2-epimerase is encoded by the ***RENBP*** gene. This protein, which catalyzes the interconversion of N-acetylglucosamine to N-acetylmannosamine, was expressed primarily by monocytes and MCs, with between 63 and 99 reads [125]. **VCAN**, or versican, is a large extracellular matrix proteoglycan that was expressed exclusively in the monocytes and at relatively high expression levels, with 140 and 774 reads.

**Galectin-2** is a soluble beta-galactoside binding lectin encoded by the ***LGALS2*** gene, which has been found to have an apoptosis-inducing effect on activated T cells [126]. Here, it was expressed exclusively by monocytes a, with 88 and 136 reads (Table 12). **Galectin 3** is a cytosolic beta-galactoside binding lectin, encoded from the ***LGALS3*** gene, which is involved in membrane repair and the autophagic removal of damaged organelles [127]. Here, it was expressed by monocytes and T cells at medium to low levels and very high levels in MCs, with 679 and 885 reads.

Collectively, these carbohydrate-related enzymes showed a high expression in MCs compared to most other cell types even if they were also expressed by most other cells.

### 2.14. Transcript Levels for Other Enzymes

In this section, we list a number of enzymes not directly involved in histamine, arachidonic acid metabolism, or carbohydrate processing, whose expression showed an interesting profile across the five cell types (Table 13). We start with the ***CYBB*** gene, which encodes NADPH oxidase 2 (**Nox2**), also known as cytochrome (**b558**) subunit beta, a superoxide-generating enzyme. Here, it was expressed at very high levels in monocytes, with 725 and 1465 reads; it had lower levels in B cells, with 134 and 90 reads, and was almost absent from T cells and MCs (Table 13). Arginase type II (**ARG2**) is an enzyme that catalyzes the hydrolysis of arginine to ornithine and urea. Here, it was expressed exclusively by MCs, with 47 and 68 reads. It may have a role in the regulation of the synthesis of nitric oxide (NO) [128]. **PADI2** and **PADI4** are two members of the peptidyl arginine deiminase family of enzymes. They convert arginine residues in proteins into citrullines in the presence of calcium ions [129]. **PADI2** was expressed in monocytes and MCs at relatively low levels, within the range of 47 to 63 reads, and **PADI4** was exclusively expressed by monocytes, with 98 and 100 reads. Both **Nox2** (***CYBB***) and **PADI4** are involved in the generation of netosis in neutrophils to form nets to trap bacteria during infections [130]. We found that both of these enzymes also were present in monocytes. The ***EPHX2*** gene encodes an epoxide hydrolase that converts epoxides to dioles and was only expressed by T cells at relatively low levels, with between 13 and 50 reads. The heme oxygenase 1 gene, ***HMOX1***, encodes an essential enzyme in the catabolism of heme and plays a key role in iron homeostasis [131]. It was expressed at high levels by monocytes, with 414 and 466 reads, and at much lower levels by MCs. The ***ALAS1*** gene encodes an enzyme, the delta-aminolevulinate synthase 1, which is of importance in the first rate-limiting step in the synthesis of heme [132]. This gene was expressed in all cells, although at very different expression levels: very low in monocytes and B and T lymphocytes and higher in MCs, with 90 and 155 reads. One wonders if heme has a specific function in MCs. The ***SAMHD1*** gene encodes a phosphorylase that has been shown to be of importance for protection against HIV infection [133]. It showed the highest levels in T cells, with between 131 and 155 reads, whereas it was lower in monocytes and MCs and very low in B cells. The protein encoded by the ***NUDT16*** gene plays a role in stabilizing **53BP1** and thereby in the repair of double-strand breaks [134]. This gene was expressed at its highest level in monocytes, with 104 and 152 reads, and much lower in the other four cell types. Monocytes produce oxygen radicals when activated, and such radicals can have damaging effects on DNA, which may be the reason why higher levels of this enzyme are needed. ***ALDH2*** encodes a detoxifying mitochondrial enzyme, an aldehyde dehydrogenase of major importance for alcohol metabolism [135]. Here, it was expressed at very high levels in monocytes, with 905 and 1627 reads, whereas it was much lower in MCs and B cells and almost absent in T cells. A similar situation, namely high levels in monocytes, with 259 and 683 reads, but low levels in B cells and MCs and almost absent in T cells, was seen for the next gene, ***ACSL1***, a gene encoding long chain-fatty-acid CoA ligase 1. This is an enzyme that converts free long-chain fatty acids into fatty acyl-CoA esters and thereby plays a key role in lipid biosynthesis and fatty acid degradation [136]. The enzyme encoded by the ***NUDT18*** gene can degrade 8-oxo-7 and 8-dihydroguanine and thereby take part in the rescue from the mutagenic effect of oxidized nucleosides [137]. As expected, this enzyme was also most highly expressed by monocytes, with 133 and 158 reads, as monocytes use oxygen radicals in antimicrobial killing. The enzyme 6-phosphogluconate dehydrogenase encoded by the ***PGD*** gene catalyzes the oxidative decarboxylation of 6-phosphogluconate into ribose 5-phosphogluconate, generating pentose sugars for nucleic acid synthesis [138]. Here, this enzyme was expressed by all five cell types but at very much higher levels in monocytes, with 1630 and 3408 reads. The question is why monocytes need approximately 40 times more of this enzyme than other immune cells, especially since they do not typically re-enter the cell cycle. The same question can be asked for the enzyme encoded by the ***GLUL*** gene, the glutamate–ammonia ligase, an enzyme that catalyzes the synthesis of glutamine from glutamate and ammonia in an ATP-dependent reaction, and the enzyme is needed to remove excess ammonia. The transcripts of this gene were present in all five cell types but were 10 times higher in monocytes than in lymphocytes and 100 times higher in MCs, with 2816 and 5447 reads in MCs. Why do MCs need such exceptionally high levels of this enzyme?

N-alpha acetyltransferase 10 is encoded by the gene ***NAA10***. This enzyme is bound to the ribosome and acetylate proteins co-translationally, an enzyme found in all kingdoms of life [139]. Here, it was expressed at 10 times higher levels in monocytes, with 167 and 251 reads, compared to the other four cell types (Table 13).

In contrast to the previous two sections, the majority of enzymes in this table showed high dominance in monocytes but not MCs, which may seem as fitting with their role similar to macrophages in cleaning functions, removing rest products, and in immune functions related to the killing of phagocytosed bacteria and also possibly in netosis mechanisms inhibiting bacteria from spreading.

### 2.15. Transcript Levels for Transcription Factors

Transcription factors not only control cell differentiation but also the response to cell activation by increasing the production of cytokines and chemokines to change the expression of cell-surface receptors and cell adhesion molecules among many other functions. The expression of transcription factors is therefore highly characteristic for a specific cell lineage and for the state of activation of a cell. We demonstrate that some transcription factors were specific or almost specific for a certain cell type, such as **GATA1** and **GATA2** for MCs and **GATA3** for T cells (Table 14). We found a relatively large panel of transcription factors that were specifically, or almost specifically, expressed by MCs, such as **MITF**, **HES1**, **HEY1**, **MEIS2**, **EPAS1**, **PTRF**, **NR4A1**, **NFE2L3**, **PBX1**, **GLI1**, and **AFF2**. Several of the transcription factors have been shown to be essential for the fate of the cell type in which they are expressed. We observed that **GATA1**, with 105 and 108 reads, was expressed at relatively low levels, whereas **GATA2** showed very high levels of expression, with 2859 and 1421 reads (Table 14). In knockout experiments, **GATA2** has been shown to be of central importance for both basophil and MC differentiation and maintenance [140]. It was essential for the expression of several MC-related genes, including the high-affinity IgE receptor alpha chain, c-kit, for the production of histamine and the expression of the cytokine IL-4 from basophils and IL-13 from MCs. **MITF** has also previously been shown to be of importance for the expression of MC tryptase and very recently also for the lineage commitment to the MC lineage during hematopoietic differentiation [141]. **GATA1** has been shown to be of importance for the expression of tryptase [142]. Thus, both **GATA1** and **GATA2** are of importance for MC development. What is striking is the large difference in expression levels between some of the factors in cells, which we assume indicates a relatively homogenous population such as MCs. Here, it was **GATA2** that was a bit of an odd bird as most of the other MC-expressed transcription factors listed above were in the lower range, with only a few with more than 500 reads (Table 14). It is unclear why **GATA2** need to be at such a high level compared to the other factors. A possible explanation is related to its involvement in so many of the MC-specific genes compared to **GATA1**, which seem to have a more restricted role in MC development. Interestingly, knocking out only one of the two copies of **GATA2** resulted in a lack of MCs in mice, indicating that high expression levels is needed for MC differentiation but not for basophils [140]. It should be noted here that **GATA1** is not only expressed by MCs but is of key importance for red blood cell development [143]. This is a more general situation, where most transcription factors are not only involved in the differentiation of one single cell type but take part in the differentiation or more than one cell type.

**MITF**, the microphthalmia-associated transcription factor, was primarily detected in MCs, with 195 and 64 reads, and at very low levels in monocytes, with 2 and 4 reads (Table 14). **MITF** is a helix–loop–helix leucine zipper transcription factor involved in the differentiation of melanocytes, osteoclasts, and MCs [144]. **HES1** was primarily expressed by MCs, with 138 and 117 reads, and at very low levels in B cells, with 8 and 6 reads. **HES1** belongs to a helix–loop–helix family of transcription factors, with seven members that repress transcription [145]. **HES1** can also heterodimerize with the next transcription factor in this list, **HEY1,** and has been shown to play a role in both the nervous and digestive systems [145]. **HEY1** is also a helix–loop–helix transcriptional repressor that was only expressed by MCs, with 86 and 206 reads. This gene was found to be expressed primarily during embryogenesis in mice [146]. ***MEIS2*** is a homeobox gene that was expressed exclusively by MCs, with 193 and 275 reads, but that has previously been found among other activities to suppress myeloid cell development [147]. **EPAS1** was also exclusively expressed by MCs, with 436 and 835 reads. **EPAS1**, also known as hypoxia-inducible factor 2-alpha, is a hypoxia-inducible factor in which several alleles contribute to high-altitude adaptation in humans [148]. **PTRF**, also named Cavin-1, was found to be MC-specific, expressed at relatively high levels, with 465 and 707 reads. **PTRF** has been shown to regulate ribosomal transcription in response to metabolic changes in adipocytes [149]. ***NR4A1*** was primarily expressed by MCs, with 147 and 997 reads (Table 13). The protein produced from this gene is involved in the regulation of cell cycle, inflammation, and apoptosis and has been shown to play a proinflammatory role in macrophages [150]. Then, we have four genes that were (almost) MC-specific but were expressed at much lower levels, namely ***NFE2L3***, ***PBX1***, ***GLI3***, and ***AFF2***, with between 18 and 95 reads. **NFE2L3** is a leucine zipper transcription factor with still unknown function. **PBX1** is a global regulator of embryonic development, which has been shown to be essential for lymphoid cell development and one of the earliest-acting transcription factors that regulate de novo B-lineage lymphopoiesis but is not needed after the pre-B-cell stage [151]. **GLI3** belongs to the family of C2H2 zinc finger transcription factors, acting both as a negative and positive regulator. **GLI3**-mutant mice have abnormalities in the CNS, lungs, and limbs [152,153]. **AFF2**, an X-linked gene, encodes a protein that is not a classical transcription factor but involved in mRNA processing, more precisely in the regulation of alternative splicing [154]. The role of several of these factors have not been analyzed for their role in MC development and biology, which warrants future studies.

We also found a number of transcription factors that were B-cell-specific, some that were T-cell-specific, and a few that were also primarily expressed by monocytes. **GATA3** is considered to be specifically expressed by CD4^+^ T cells and is particularly important for the development of Th2 cells. However, we detected **GATA3** at the same levels in CD8^+^ cells, within the range of 150 and 281 reads (Table 13 and Table 14). **GATA3** is also expressed by naïve T cells. **FOXP3**, a transcription factor that is specific for regulatory T cells, was here expressed at low levels only in CD4^+^ T cells, with 30 and 23 reads (Table 14). Five to 10% of CD4^+^ T cells in the circulation have been estimated to be regulatory T cells, which is why the expression level of **FOXP3** in the regulatory T cells can be estimated to be between 150 and 600 reads [155]. **TBX21**, or **T-bet**, is a transcription factor used as a marker for Th1 cells, and this transcription factor was expressed at low levels in both CD4^+^ and CD8^+^ T cells, at levels between 15 and 78 reads (Table 13) [156]. **RORC**, a marker for Th17 cells, was also expressed at very low levels in both CD4^+^ and CD8^+^ T cells, at levels between 5 and 17 reads (Table 14) [157]. In line with what was mentioned above concerning the cell-type specificity of the majority of transcription factors, **RORC** also plays an important role in lymphoid organogenesis in particular lymph nodes and Peyer´s patches, and it may also be involved in regulating circadian rhythms [158]. **FOXP3**, **T-bet**, and **RORC** were all expressed at relatively low levels, with between 10 and 78 reads, most likely primarily due to the fact that they were only expressed by a small subpopulation of these T cells. Also interesting is the expression of both **GATA3** and **T-bet** at similar levels in both CD4^+^ and CD8^+^ T cells and that **RORC** was also expressed in CD8^+^ T cells, indicating that we have a similar separation into Type 1, Type 2, and Type 17 cells in CD8^+^-positive cells as we have in CD4^+^ cells. It is only **FOXP3** that seems exclusively expressed in CD4^+^ T cells.

We also found a number of transcription factors that are of major importance for B-cell development, including the two POU domain-containing transcription factors, **POU2F2**, also named **OCT2**, and the associated factor **POU2AF1. POU2F2** was primarily expressed in B cells and monocytes, within the range of 240 to 328 reads, but were also at lower levels in T cells (Table 14). **POU2F2** has been shown to regulate immunoglobulin and IL-6 gene transcription in B cells [159]. **POU2AF1** is an associated factor binding to **OCT2** and stabilizing the complex, thereby enhancing transcription [160]. Here, **POU2AF1** was almost B-cell-specific, with 307 and 435 reads, and had only 100-fold lower levels in CD4^+^ T cells. **POU2AF1** is of importance for multiple stages of B-cell development and the knockout shows impaired production of transitional B cells and defective maturation of recirculating B cells [161]. In this study, a related POU domain-containing factor, **POU2F1**, also named **OCT1**, was expressed only at very low levels in all five cell types.

Two other B-cell-related genes that belong to the **SP** family of transcriptional regulators are **SPI1** (**PU1**) and **SPIB** [160]. The **SPI1** gene encodes the transcription factor **PU.1**, which was expressed at a relatively high level in monocytes, with 369 and 252 reads, and had lower levels in both B cells and MCs. Humans with mutations in the **SPI1** gene lack circulating B cells and the majority of dendritic cells, and the mutations also affect myeloid differentiation [162]. Mice lacking **SPI1 (PU.1)** die in utero [162]. Here, **SPIB** expression was specific for B cells, with 48 and 73 reads. It has been shown to be expressed in plasmacytoid dendritic cell precursors and to be a negative regulator of T-, B-, and NK-cell development [163]. However, in contrast to published data, we did not detect expression of this factor in T cells.

Additional factors that have been shown to be of major importance for B-cell development or were primarily expressed by B cells are **PAX5**, **AFF3**, **KHDRBS2**, **EBF1**, **E2F5**, **BACH2**, and **SETBP1**. **PAX5** is a member of the paired box family and a transcription factor of major importance for B-cell development [164]. Here, it was expressed only by B cells at relatively high levels, with 590 and 707 reads. **PAX5** has been shown to be essential for B-cell commitment by suppressing alternative lineage choices [164]. **AFF3** is a transcriptional activator that was also almost B-cell-specific, with 148 and 125 reads (Table 14). **AFF3**-deficient mice exhibit low levels of immunoglobulins and appear to regulate class switching by binding to the switch regions and facilitating the recruitment of **AID** [165]. Here, **KHDRBS2** was also essentially B-cell-specific, with an expression level of 212 and 137 reads. **KHDRBS2** is not a classical transcription factor but an RNA-binding protein with affinity for polyA and polyU tails and is also involved in differential splicing [166]. Early B-cell factor 1 (**EBF1**) controls the expression of key proteins required for B-cell differentiation and signal transduction [167]. In this study, this gene was almost exclusively expressed by B cells, with 53 and 68 reads. **E2F5** belongs to the E2F family of transcription factors, which are major regulators of cell proliferation [168]. This gene was expressed primarily in B cells at a low level, with 51 and 67 reads. **BACH2** is a transcription factor of major importance for both B- and T-cell development. In B cells, this factor is needed both for early development and late development, during isotype switch and plasma cell development, and in T cells, it seems to be of importance for Th2 development [169]. Here, **BACH2** was expressed in both B and T cells, with 351 and 392 reads in B cells and 12 and 76 reads in T cells. **SETBP1**, which has been shown to act as an epigenetic hub and be involved in several hematological malignancies, was expressed primarily in B cells, with 114 and 91 reads [170].

We also identified a few additional transcription factors that were expressed primarily in T cells, in addition to the ones described further above. **LEF1**, a member of the T-cell factor/lymphoid enhancer factor family, was expressed at a relatively high level and almost exclusively in both CD4^+^ and CD8^+^ T cells, at a level between 365 and 632 reads (Table 14). **LEF1** has been shown to be essential for T-cell development and to enhance **GATA3** expression, thereby promoting Th2 development, and it is also known to be essential for innate NK cells and T-cell development [171]. **RNF157** is a ring finger protein that was primarily expressed by T cells, with between 40 and 56 reads. **TCEA3** is a member of the translation elongation factor TFIIS family that was expressed primarily by T cells, with a range of 70 to 140 reads (Table 14).

**BHLHE40** is a helix–loop–helix transcription factor that is described as a key regulator of immunity during infection autoimmunity and inflammatory conditions, which was expressed at very high levels by MCs, with 1449 and 969 reads, and at relatively high levels in the other cell types, except for B cells, where the expression was very low (Table 14) [172]. The hematopoietically expressed homeobox protein **HHEX** is a protein involved in developmental processes. Without the expression of this factor, mice die in utero [173]. Here, **HHEX** was expressed at a higher level in B cells, with 170 and 173 reads, and had lower levels in monocytes and MCs but was essentially absent in T cells.

A number of transcription factors were expressed in more than one cell type and at various levels. We start with a large group of zinc finger proteins. The human genome contains a number of different families of zinc finger proteins, with the majority belonging to the C2H2 type, with two cysteines and two histidines, which coordinate the zinc binding. This class of zinc finger proteins is the second largest gene family in the human genome with an estimated number of 700 members, which is only surpassed by the seven transmembrane receptors, with approximately 900 members. C2H2 zinc finger proteins are a highly evolving family of regulators in which the members differ markedly between mice and humans, and we have only started to understand their role in human biology; many are expressed in multiple cells and at relatively low levels [174,175,176].

We start the analysis of zinc finger proteins with three members of the **IKZF** family. **IKZF1** (**Ikaros**) was expressed by all five cell types but was higher in T cells and MCs, within a range of 56 to 158 reads. **IKZF1** has been shown to be of major importance for early B-cell development and in the function of Th cells, and it is also of major importance for erythrocyte and granulocyte differentiation [177,178]. **IKZF2** (**Helios**) was expressed in very low levels in B cells, CD4^+^ T cells, and MCs, and had slightly higher levels in CD8^+^ T cells, with 16 and 12 reads, but not in monocytes. **Helios** has been shown to be expressed by both CD4^+^ and CD8^+^ T-regulatory cells, and to interact with **FOXP3 [179]**. However, here, we primarily detected expression in CD8^+^ T cells. **IKZF3** (**Aiolos**) forms a heterodimer with **Ikaros**, and it has been described as lymphocyte-specific [180]. In agreement, we also found **IKZF3** transcripts in the lymphocytes with slightly higher levels in B cells, with 112 and 183 reads, compared to T cells, and they had very low levels in MCs (Table 14).

**GFI1** is a transcriptional repressor and a zinc finger protein of importance for normal hematopoiesis. It was primarily expressed in T cells and MCs, with between 11 and 116 reads (Table 14) [181]. **ZEB1** is a zinc finger and homeodomain transcription factor that was expressed at a relatively low level in four out of these five cell types but not in monocytes. **ZEB1** has been shown to inhibit IL-2 and E-cadherin expression [182]. In contrast to **ZEB1**, **ZEB2** was most highly expressed in monocytes and MCs, with between 236 and 463 reads, and it was lower in CD8^+^ T cells and B cells and almost absent in CD4^+^ T cells (Table 14). **ZEB2** is an R-SMAD-binding protein involved in neural development [183]. **ZEB2** has also been shown to be of major importance for the final development of NK cells and effector CD8^+^ T cells and also for monocytes and plasmacytoid dendritic cells [184].

**KLF1** is a C2H2 zinc finger protein that is essential for the proper maturation of erythrocytes. **KLF1** was not expressed by any of the five cells in this analysis. This was in contrast to **KLF4**, which is primarily expressed by resting non-dividing cells, and its overexpression induces cell cycle arrest [185]. Here, **KLF4** was expressed at very high levels by freshly isolated MCs, with 1751 and 1041 reads, and it had high levels in monocytes, lower levels in B cells, but it was almost absent in the T cells. Its expression was dramatically downregulated upon in vitro culturing of the MCs for 3 weeks to levels below one read ([3] and Appendix A). **ZNF385A** is not a regular transcription factor but an RNA-binding protein involved in the regulation of cell cycle and apoptosis [186]. It was primarily expressed in monocytes, with 311 and 269 reads. **ZNF467**, which was primarily expressed in monocytes, with 131 and 120 reads, is a C2H2 zinc finger protein involved in gene regulation by augmenting **STAT3** activity by keeping it in the nucleus [187]. Here, **ZNF513** was expressed in all five cell types at low levels, except for in monocytes, where the levels were approximately 10 times higher in the range of 226 to 271 reads. This zinc finger protein has been shown to be involved in the development and maintenance of the retina [188]. Its role in monocyte–macrophage biology seems to be unknown. **ZNF521** showed only low-level expression in MCs. This gene has been shown to be associated with erythrocyte differentiation by directly binding to **GATA1** [189]. To our knowledge, its potential role in MC development is not known. **ZNF703** is a transcriptional repressor by recruiting histone deacetylases to promoter regions but does not bind directly to DNA [190]. Here, it was expressed primarily in monocytes, with 74 and 83 reads, and at lower levels in MCs (Table 14). **ZNF787** has been found to interact with **HDAC1** and regulate tight junction proteins, thereby affecting the blood–brain barrier function [191]. Here, it was expressed by monocytes and only at low levels. **ZNF827** is a single-strand DNA-binding protein promoting homologous recombination-mediated DNA repair [192]. This gene was primarily expressed in B and T lymphocytes at low levels. **ZNF831** is another C2H2 zinc finger protein with a very similar expression pattern as **ZNF827**. **ZNF831** acts as a transcriptional suppressor and enhances apoptosis signals by inhibiting the expression of STAT3/Bcl2 [193]. **ZCCHC24** has been associated with adult-onset severe asthma [194]. Here, this gene was expressed primarily in MCs, with 53 and 65 reads (Table 14). **ZMIZ1** has been described as a coactivator of several transcription factors, including p53, the androgen receptor, and **NOTCH1**, and to be involved in neural development [195]. Here, it was expressed by all five cell types but at very much higher levels in monocytes and MCs than in B and T cells, with 2280 and 1103 reads in monocytes and 190 and 300 reads in MCs. **CXXC1**, the last member of the zinc finger proteins in this list, is a CXXC-type zinc finger protein as well as a CpG-binding protein that recognizes unmethylated CpG sequences and regulates gene expression [196]. It was expressed at very low levels in all five cell types; however, it had higher levels in monocytes, with 76 and 85 reads.

The interferon regulatory factor **IRF5** is a transcription factor with diverse roles, including the virus-mediated activation of interferon and polarizing macrophages into M1-type [197]. It was expressed at high levels by monocytes, with 444 and 500 reads, whereas it had lower levels in B cells and even lower in MCs, and was almost absent in T cells (Table 14). CCAAT/enhancer-binding protein alpha that is encoded by the **CEBPA** gene has been found to be essential for the development of mature granulocytes and monocytes and also impairs adipocyte maturation [198]. Here, it was essentially expressed only by monocytes, with 54 and 96 reads. The protein encoded by the **MAFG** gene is a leucine zipper-containing transcription factor. Mice lacking **MAFG** show mild neuronal phenotype and mild thrombocytopenia [199]. Here, this gene was expressed at relatively high levels in monocytes, with 411 and 390 reads, but it was at lower levels in MCs and almost absent in the lymphoid cells. **ELF3** and **VENTX** were both essentially only expressed by monocytes and were at similar levels, within the range of 46 to 76 reads. **ELF3** has been shown to be involved in epithelial–mesenchymal transition, but its role in monocyte biology seems less well known [200]. **VENTX** has been found to be primarily expressed by hematopoietic cells and to be involved in both proliferation and differentiation [201]. **RUNX2** and **MAF** also showed a similar expression pattern, with low level expression in T cells and MCs. **RUNX2** has been described as a master regulator of bone due to its importance in osteoblast differentiation [202]. **MAF** transcription factors are involved in the development of many different organs and have been indicated to be regulators of general energy homeostasis [203]. The ***ETS2*** gene encodes C-ETS2 a protein belonging to the ETS family of transcription factors. **ETS2** was expressed in all five cells, but it was much higher and variable in monocytes and MCs, with 177 and 206 reads in monocytes and 395 and 9645 reads in MCs.

**MYC** is involved in stem cell biology and often connected to proliferation and therefore classified as a proto-oncogene [204]. Here, it was expressed at relatively high levels in all cell types, within the range of 79 to 259 reads, except in monocytes, which were essentially negative (Table 14). **STRBP** is a microtubule-associated RNA-binding protein involved in spermatid formation [205]. It was expressed essentially only in lymphoid cells, with ten times higher levels in B cells than in T cells, with 258 and 270 reads in B cells. **EIF2AK3** showed an expression profile similar to that of **MYC**, with expression in all the cell types analyzed here except monocytes. **EIF2AK3** is an enzyme that phosphorylates the alpha subunit of the translational initiation factor 2, leading to its inactivation and thereby reducing translational initiation. **CTDSP1** was expressed at relatively high levels in all cell types except for the monocytes, where the level was very high, 6–8 times higher, with 1508 and 1675 reads. **CTDSP1** is an N-terminal domain of RNA polymerase II with phosphatase activity [206]. **CCDC9** is an RNA-binding protein and most likely a component of the exon junction complex [207]. Here, it was expressed in all five cell types but was 10 times higher in monocytes, with 1118 and 585 reads, compared to the other cell types. The question is why monocytes need such exceptionally high levels of the polymerase component **CTDSP1** and the exon–intron junction binding protein **CCDC9**. We may speculate that this is needed for monocytes to be able to rapidly produce extremely high levels of inflammatory cytokines and chemokines when encountering LPS, as we have seen previously [16]. **SCML4** encodes a protein that is predicted to be a histone-binding protein and acts as a negative regulator of transcription (NCBI, http://ncbi.nlm.nih.gov). Here, it was expressed at low levels only in lymphocytes.

The regulation of transcription is complex, and many of the transcription factors that are important for a particular cell type can be expressed in many cell types in a non-exclusive manner; factors with broader expression may still be crucial to particular lineages [160].

### 2.16. Transcript Levels for SOX Members of Regulators of Tissue Development

The SRY-related HMG box (SOX) genes constitute a large family of transcriptional regulators of differentiation that has been shown to regulate the development of the eye lens, hair follicles, gut, B cells, muscle, and blood vessels, to name just a few [208]. As can be expected, based on their well-established role as regulators of cell fate decisions during early development, the members of the **SOX** family were almost totally absent in our analysis (Table 15) [208]. Only a few of them such as **SOX4**, **13**, **17,** and **18** were expressed at very low levels in MCs, and **SOX13** was also detected at very low levels in CD8^+^ T cells.

### 2.17. Transcript Levels for STATs

In contrast to many of the other protein families described in this communication, the transcript levels for the seven signal transducers and activators of transcription (**STATs)** showed a relatively even distribution among these five cell types. **STATs** are important for connecting cytokine receptor triggering with transcriptional activation. Here, they were expressed at very similar levels in all five cell types, except for **STAT4**, which was almost T-cell-specific (Table 16) [209]. **STATs** are cytoplasmic transcription factors that, upon phosphorylation by receptor-associated kinases, dimerize by forming homo- or heterodimers that translocate into the nucleus and act as transcription factors. **STATs** are of major importance for many immune cells, and their activity is regulated by key interleukins, interferons, and other growth and differentiation factors [210]. As mentioned above, **STAT4** was in contrast to the other **STATs** that were essentially only expressed in T cells in this study, with between 34 and 77 reads. It has been shown to be of major importance for Th1 differentiation [211].

### 2.18. Transcript Levels for Cytokines, Chemokines, and Other Growth and Differentiation Factors

Cytokines are primarily used in intercellular communication to send a message from one cell to another. In most cases, they are therefore silent during nonactivated states but can rapidly be turned on to send out a signal to other cells. As previously mentioned, we recently published a very clear example of this situation in a study of the response to LPS by human blood monocytes [16]. In that study, monocytes rapidly induced the expression of the major inflammatory cytokines and expressed these at very high levels. As an example, **IL-6** increased 75,000 times within 4 h of the addition of LPSs to the culture medium [16]. In line with this finding, we found that most of the cytokines in this analysis were expressed at very low levels, with the exception of a few cytokines that seemed to be constitutively expressed most likely due to their functions in tissue homeostasis. Examples of the latter situation included lymphotoxin B, **LTB**, which was very highly expressed by B and T cells, and leukemia inhibiting factor, **LIF**, which was exclusively expressed by MCs (Table 17). To these cytokines, we can also add one of the colony-stimulating factors, the macrophage colony-stimulating factor **M-CSF**, expressed by MCs and **IL32** expressed by T cells, as well as two chemokines, **CCL2**, expressed by MCs, and **CCL5**, expressed by T cells, primarily CD8^+^ T cells.

**LTB** was highly expressed in B and T cells, within the range of 784 and 2219 reads, and at much lower levels in monocytes, with 17 and 20 reads (Table 17). **LTB** is involved in promoting lymphoid tissue development and maintenance. It has also been reported to protect the host against pathogenic insults and to regulate the composition of the host microbiota [212]. **LIF** was expressed only by MCs and at relatively high levels, with 659 and 307 reads (Table 17). **LIF** is of major importance for stem cells to maintain their multipotency and appear to have non-redundant actions in maternal receptivity to blastocyst implantation, placental formation, and the development of the nervous system [213]. **LIF** is also preferentially expressed by mouse peritoneal MCs, showing that the expression pattern is conserved between different mammals [18].

Tumor necrosis factor-alpha (**TNF**) is an important inflammatory cytokine that was expressed by all five cell types but at relatively low and variable levels. A high individual variation was seen in CD8^+^ T cells, where one individual showed 63 reads and the second individual 371 reads, indicating that its expression is highly sensitive to external signals, similar to what we found for monocyte responses to LPS (Table 17) [16]. **TNFSF10** showed a similar pattern as **TNF**, with expression in all five cell types and at relatively low levels (Table 17). **TNFSF10**, which is also named **TRAIL** or **CD253**, acts as a ligand to induce programmed cell death by binding to **TRAILR1** and **2** [214].

Of the three colony-stimulating factors, we only detected high expression for one, namely **CSF1**, also named the macrophage colony-stimulating factor **M-CSF**, with 1579 and 933 reads in MCs (Table 17). This may indicate that MCs have an important role in the maintenance of macrophage tissue homeostasis. A low level of **CSF2**, also named **GM-CSF**, was seen in MCs, with 9 and 61 reads, but not in any of the other four cell types (Table 17). Upon the in vitro culturing of the MCs for 3 weeks, **CSF1** levels decreased by approximately 50–70%, but the **CSF2** levels did instead increase 10–100-fold, reaching between 276 and 799 reads ([3] and Appendix A).

For the majority of the other cytokine genes, including erythropoietin (**EPO**), **IL2**, **IL3**, **IL4**, **IL5**, **IL9**, **IL10**, **IL11**, **IL19**, **IL12B**, **IL17A**, **IL17F**, **IL20**, **IL21**, **IL22**, **IL25**, **IL26**, **IL28A**, **IL28B**, **IL29**, **IL31**, **IL34**, **IL36A**, **IL36B**, **IL36G**, and **IL37**, we found no or very low levels of expression in all five cell types (Table 17). A low or very low level of **IL1A** was seen in MCs, with 4 and 5 reads. **IL1B** was expressed at very low levels by monocytes and MCs (Table 17). Low levels of both **IL6** and **IL7** were seen in B cells. **IL8** was detected in monocytes and MCs, with 32 and 38 reads and 65 and 80 reads, respectively (Table 17). **IL13** expression was seen in MCs, with 29 and 47 reads, and **IL12A** in B cells, with 24 and 18 reads (Table 17). A low level of **IL15** was also detected in B cells, monocytes, and both CD4^+^ and CD8^+^ T cells but not in MCs (Table 17). Very low levels were also observed in **IL16** and **IL17B** in MCs and **IL23A** and **IL24** in both populations of T cells (Table 17). Interestingly, in contrast to the majority of the other interleukins, we found very high levels of **IL32** mRNA in both CD4^+^ and CD8^+^ T cells, in the range of 1800 to 3500 reads, and much lower levels in MCs (Table 17). **IL32** has been shown to have both pro- and anti-inflammatory properties, to be a metabolic regulator, and to have a role in antimicrobial host defense. It shows a very low degree of sequence conservation among many mammalian species and is not found in rodents [215]. Transcripts for **IL33**, an important alarmin, were only observed in MCs and at a very low level, with four and two reads (Table 17).

When it comes to chemokines, we found that MCs, but not any of the other cell types, expressed very high levels of **CCL2**, with 1518 and 1644 reads (Table 17). **CCL2** (alternative name: monocyte chemoattractant protein 1, **MCP1**) is important for recruiting monocytes, memory T cells, and dendritic cells to the sites of inflammation [216]. High levels of this chemokine were also found in mouse peritoneal MCs, showing that this pattern of expression is evolutionarily conserved [18]. Low levels of **CCL3** were seen in monocytes, CD8^+^ T cells, and MCs. Very high levels of **CCL5** were observed in CD8^+^ T cells, with 1778 and 1157 reads, and it had lower levels in CD4^+^ T cells (Table 17). In contrast to the above chemokines, **CXCL16** was more broadly expressed, within the range of 12 to 54 reads, and was at higher levels in MCs, with 263 and 160 reads. Very low levels of pro-platelet basic protein (**PPBP**), also named **CXCL7**, were seen in three of the cell types, and it was expressed at high levels in monocytes, with 154 and 97 reads, but was not seen in MCs. As described above, we observed the constitutive expression of some chemokines, such as **CCL2** in MCs, **CCL3** in monocytes, **CCL4** in CD8^+^ T cells, and MCs and **CCL5** in T cells (Table 17). One can only speculate about the functions of these constitutively expressed chemokines. A possibility is to maintain a steady state influx of a low number of immune cells into the tissue to allow for their rapid responses to inflammatory stimuli. Another question is whether they are continuously released or require activation. Recently, **CCL5** has been shown to be released by CD8^+^ T cells upon activation and to contribute to the pathology of primary Sjögren´s syndrome, indicating that activation is of importance at least for T cells [217]. Conversely, substantial baseline release was reported for skin MCs [218].

Among the vascular endothelial growth factors, which are of importance for blood vessel formation, we found a very high level of expression of **VEGFA** in MCs, with 1948 and 1323 reads, and it had relatively high levels in monocytes, with 222 and 109 reads, but no expression was observed in any of the other cell types. All five cell types expressed **VEGFB** but at variable levels. In contrast, **VEGFC** was negative in all cell types.

The thymidine phosphorylase **TYMP** is not a classical growth factor but an enzyme with a prominent role in angiogenesis similar to **VEGF**s, and therefore it is listed here. **TYMP** was expressed in all five cell types but at very different levels, with 574 and 637 reads in monocytes and in the range of 6 to 37 reads in the other four cell types.

Of the three members of the platelet-derived growth factor family, we observed the expression of **PDGFA** and **PDGFC** only in MCs, with 270 and 366 reads for **PDGFA** and 17 and 20 treads for **PDGFC** (Table 17). **VEGFs** and **PDGFs** are closely related and form a separate gene family named the **VEGF/PDGF** family.

**GDF15**, also named macrophage inhibitory cytokine 1 (**MIC-1**), belongs to the transforming growth factor-beta family. Its function is not fully clear but seems to have a role in injured tissues, where it contributes to regulating inflammation, cell survival, proliferation, and apoptosis [219]. Here, **GDF15** was expressed primarily in MCs, with 136 and 27 reads. Transforming growth factor A (**TGFA**) was only expressed by MCs and at a relatively low level. **TGFB1I1** was also essentially MC-specific, with expression levels of 137 and 149 reads. Studies suggest that **TGFB1I1** is involved in an array of different functions, including cell growth, proliferation, migration, differentiation, and senescence, and is known to be located primarily at focal adhesion points [220]. **ADM** is a gene encoding adrenomedullin. This gene was basically only expressed by monocytes and MCs, within the range of 50 to 140 reads. **ADM** knockout is embryonically lethal, and the haploinsufficiency of its receptor has been shown to cause reduced fertility, hyperprolactinemia, skeletal abnormalities, and endocrine abnormalities in mice [221]. Platelet factor 4 (**PF4**), also known as **CXCL4**, was almost exclusively expressed by monocytes, with 122 and 64 reads. **PF4** is known to be produced by platelets and macrophages and released from alpha granules from platelets, but our findings suggest that it is also produced by resting monocytes. It is chemotactic for neutrophils and fibroblasts and has a high affinity to various glucosaminoglycans (GAGs), including heparin, and it promotes leucocyte adhesion to endothelium [222]. EGF-like module-containing mucin-like hormone receptor-like 2, also known as **CD312** or **EMR2**, has been shown to be expressed by monocytes–macrophages and all granulocytes, which fits our data, according to which it was expressed at high levels in monocytes and MCs, within the range of 150 to 400 reads [223]. The ligation of **EMR2** by antibodies promotes neutrophil and macrophage effector functions, indicating a role in potentiating inflammation [224]. Here, the fibroblast growth factor binding protein 2 (**FGFBP2**) was expressed exclusively in CD8^+^ T cells and at relatively high levels, with 66 and 118 reads. In the Human Protein Atlas (www.proteinatlas.org), it was also found to be expressed primarily by CD8^+^ T cells and NK cells among different immune cells. Its function seems to be still relatively unknown.

The expression of cytokines and chemokines is also very complex, similar to transcription factors, with some being expressed at high steady-state levels and others only after activation. The question related to the ones being expressed at high steady-state levels is whether they are released directly after synthesis for more general tissue maintenance purposes.

### 2.19. Transcript Levels for Cytokine-Induced Proteins

We also observed a number of genes that are known to be induced by various cytokines, including ***TNFAIP2*** and ***3***, ***TGFBI***, and ***IFITM1*** (Table 18). They have most likely several different roles in tissue homeostasis.

The full function of **TNFAIP2**, the primary response gene of TNF-α, is not known, but it may have a role as the regulator of cell proliferation and migration [225] (Table 18). Here, it was expressed primarily in monocytes at a high level, with 697 and 483 reads, and at a much lower level in MCs, with 44 and 58 reads. **TNFAIP3** is a zinc finger protein and a deubiquitinating enzyme that has been shown to inhibit NFκB activation [226]. Knockout experiments have shown that **TNFAIP3** is critical for limiting endotoxin and TNF-induced NFκB responses [226]. Here, this gene was expressed at very high levels in T cells and MCs, with between 189 and 5148 reads, and it had lower levels in monocytes and B cells (Table 18). Transforming growth factor-beta (**TGFBI**) is a 68kDa protein that binds Type I, II, and IV collagens, regulates cell adhesion, and serves as a ligand for several integrins through its RGD motif [227]. Here, **TGFBI** was primarily expressed by monocytes, with 380 and 244 reads. The gene encoding interferon-induced membrane protein 1 (**IFTIM1**) was expressed in all five cell types but at considerably higher levels in both CD4^+^ and CD8^+^ T cells, with levels between 475 and 713 reads. The proteins encoded by members of this small gene family are involved in the control of cell proliferation, the promotion of homotypic cell adhesion, protection against viral infection, the promotion of bone matrix maturation and mineralization, and germ cell development [228].

### 2.20. Transcript Levels for the Major Cytokine and Chemokine Receptors

To respond to the messages sent by different cytokines and chemokines, cells need to express the corresponding receptors. By evaluating the expression of such receptors, we found a highly selective pattern, which is logical as only some cells are supposed to respond to a particular cytokine or chemokine. These receptors are also typically expressed at relatively low levels but with several exceptions to this rule, where the cytokine and its receptor most likely have functions related to tissue maintenance. One such exception is **KIT**, the receptor for stem cell factor (**SCF**). **KIT** and **SCF** are central to the development of MCs, and among the cells in this analysis, **KIT** was highly expressed only by MCs, with 1023 and 458 reads (Table 19). Two other receptors showed a similar MC-specific expression pattern. These were the receptors for erythropoietin (EPO) and IL-33, **EPOR** and **IL1RL1** (9). They were expressed in MCs, with 105 and 35 and 125 and 289 reads, respectively. Very low levels were detected of the IL-1 receptor (**IL1R1**) in MCs and **IL1R2** in monocytes and MCs. A complex pattern was seen for the different components of the IL-2 receptor. The alpha-chain **IL2RA** was expressed at low levels in B cells, CD4^+^ T cells, and MCs, whereas the beta-chain **IL2RB** was relatively highly expressed in CD4^+^ and CD8^+^ T cells only, with between 148 and 316 reads, and the gamma-chain **IL2RG** was expressed in all five cell types but with much higher levels in T cells. The IL-3 receptor **IL3RA** was expressed at very low levels in monocytes and MCs and at variable levels in the three other cell types. The very high level in one of the B-cell samples is questionable, which is why we have marked it with a question mark. Interestingly, the IL-4 receptor **IL4R** was highly to very highly expressed in all five cell types, with reads reaching almost 1200 reads in one of the CD4^+^ T-cell samples. This contradicts a previous transcriptomic study, where this receptor was primarily expressed by CD19^+^ B cells [229]. Here, the IL-5 receptor **IL5RA** was only detected in MCs and at low levels. The IL-6 receptor **IL6RA** was expressed in all cells, except B cells, at varying and relatively low levels, in the range of 6–98 reads. The IL-7 receptor **IL7R** was very highly expressed in both CD4^+^ and CD8^+^ T cells, with between 390 and 762 reads and very low in MCs. The IL-9 receptor **IL9R** was expressed only in MCs and at very low levels. The IL-10 receptor alpha-chain **IL10RA** was relatively highly expressed in lymphocytes and monocytes, within the range of 167 to 334 reads. **IL10RB** and the IL-11 receptor **IL11RA** were expressed at very low levels in all five cell types. **IL12RB1** was expressed at very low levels, ranging from 3 to 14 reads, except MCs, in which it was negative. The second IL-12 receptor, **IL12RB2**, was only expressed at very low levels, with one to three reads in T cells. Of the two IL-13 receptors, **IL13RA1** and **IL13RA2**, only one was expressed at very low levels in B cells, monocytes, and MCs. The IL-15 receptor **IL15R** was expressed in all cell types at very low levels, except monocytes, with 35 and 45 reads. The IL-16 receptor is **CD4**, which has been described in previous sections. The different IL-17 receptors showed very different patterns of expression. **IL17RA** was expressed in all cell types, with slightly higher levels in monocytes, with 160 and 96 reads. **IL17RB** was negative in all cell types, **IL17RC** was expressed only at very low levels in monocytes and MCs, and **IL17D** only at very low levels in MCs. **IL17RE** was only expressed at very low levels in T cells. The IL-18 receptor **IL18R1** was expressed in T cells and MCs but at much higher levels in MCs than in T cells, with 280 and 216 reads in MCs. The two receptor subunits for IL-20, **IL20RA** and **IL20RB**, were essentially negative for all five cell types. The IL-21 receptor, **IL21R**, was expressed at low levels in B cells and T cells, with expression in the range of 7 to 30 reads. The IL22 receptor subunits, **IL22RA** and **IL22RB**, and the IL-23 receptor, **IL23R**, were all negative. In contrast, the IL-27 receptor, **IL27RA**, was expressed in all five cell types but at relatively low levels. Both the IL-28 receptor (**IL28RA**) and the IL-31 receptor (**IL31RA)** were negative in all cell types.

We also analyzed the expression of the receptors for the colony-stimulating factors **G-CSF**, **M-CSF**, and **GM-CSF**. The expression of **CSF1R**, the receptor for macrophage colony-stimulating factor (**M-CSF**), was expressed essentially only by monocytes, with 226 and 210 reads. The **CSF2RA**, which is the alpha subunit of the granulocyte–macrophage colony-stimulating factor (**GM-CSF**), was also expressed primarily in monocytes, with 83 and 63 reads, but it was also at lower levels in MCs, with 14 and 6 reads. The beta chain of the **GM-CSF** receptor (**CSF2RB)** was expressed at relatively high levels by both monocytes and MCs, with expression levels in the range of 300 to 500 reads and at lower levels in B cells. **CSF3R**, the receptor for granulocyte colony-stimulating factor (**G-CSF**), was almost exclusively expressed by monocytes and at a very high level, with 2682 and 2117 reads. The expression of the **M-CSF** receptor by monocytes makes sense as it is the receptor that regulates their proliferative response [230]. In contrast, the extremely high levels of the **G-CSF** receptor were less expected. It is essential for granulocyte proliferation and differentiation but has also been found to be highly expressed by human macrophages in a large analysis of the human proteome (proteinatlas.org).

Several of the tumor necrosis factor receptor superfamily receptors were differentially expressed among these five cell types. Two were almost B-cell-specific, and two were almost MC-specific, **TNFRSF13B** and **C** and **TNFRSF9** and **21**, respectively (Table 19). **TNFRSF13B** has been shown to play a very important role in B-cell development by controlling T-cell-independent B-cell antibody responses, isotype switching, and B-cell homeostasis by binding **APRIL**, **BAFF**, and **CAML** [231,232,233]. This receptor was exclusively expressed by B cells, with 93 and 55 reads. **TNFRSF13C**, also named **BAFF-R** or **BR3**, is a receptor for **BAFF**, which is an essential factor for B-cell maturation and survival [234]. This receptor was also almost exclusively expressed by B cells, with 478 and 308 reads. **TNFRSF9**, also named **CD137**, is a costimulatory molecule functioning to stimulate T-cell proliferation, dendritic cell maturation, and the promotion of antibody secretion, but it is also described as a checkpoint inhibitor [235]. Here, this receptor was almost exclusively expressed by MCs, with 159 and 351 reads. **TNFRSF9** has been described as being expressed primarily on activated T cells. Here, it showed a very low expression level in T cells, with one and two reads, indicating that almost all T cells in this analysis were resting T cells. Interestingly, the role of **TNFRSF9** in MC function is less well known. However, it has been shown to be upregulated on MCs in response to an interaction with tumor cells [236]. **TNFRSF21**, also named death receptor 6 (**DR6**), interacts with **TRADD**, and knockout experiments have indicated that this protein plays a role in T-helper cell activation and may be involved in inflammation and immune regulation [237]. This receptor was almost exclusively expressed by MCs, with 292 and 226 reads.

The lymphotoxin beta receptor **LTBR**, also known as the **TNFRSF3**, can induce IL-8 secretion in apoptosis and the development of secondary lymphoid organs [238]. This receptor was expressed only by monocytes and MCs, with 198 and 146 reads and 74 and 65 reads, respectively. This receptor was not expressed by B and T cells, which are the major producers of the ligand (lymphotoxin), indicating a direct communication between lymphocytes, monocytes, and MCs by this cytokine (Table 19). The transforming growth factor beta receptor 3 (**TGFBR3**) was expressed by both T cells and MCs, with between 50 and 182 reads in T cells and 22 and 29 reads in MCs. The activin receptor **ACVR1B** was expressed in all cells at low levels but was slightly higher in monocytes and MCs. Activin belongs to the TGF-beta family of growth factors.

Some of the chemokine receptors were also differentially expressed among these five cell types. **CXCR4** was highly expressed by B and T cells, with levels in the range of 180 to 1200 reads, and it was expressed at much lower levels in monocytes and MCs (Table 19). **CXCR4** is the receptor for **CXCL12**, which acts as a strong attractant for lymphocytes and has also an essential role in the homing of hematopoietic stem cells to the bone marrow, cell migration during inflammation, the establishment of functional lymphoid microenvironments, and organogenesis [239,240]. **CXCR5**, a receptor for **CXCL13**, has been shown to be essential for B-cell migration to the B-cell areas within the spleen and lymph nodes [241,242]. This receptor was only expressed by B cells, with 126 and 136 reads, and at 10 times lower levels in CD4^+^ T cells. **CCR6** is a receptor for **CCL20**, also named **MIP-3 alpha**. This receptor binds only **CCL20**, which is relatively unique compared to other chemokine receptors, which often bind several ligands [243]. Here, this receptor was only expressed by T cells with low expression levels. The **CCR6**-**CCL20** chemokine receptor pair has been shown to be important for B-lineage maturation and antigen-driven B-cell differentiation, as well as for the recruitment of Th17 and T-reg cells during inflammatory responses [244]. We could, however, only detect mRNA for this receptor in T cells and not in B cells, in contrast to previous findings [243]. **CCR7** is a receptor for both **CCL19** and **CCL21**, and its main function is to guide immune cells to different organs such as lymph nodes and to promote the development of T cells in the thymus; it is also involved in CD8^+^ T-cell function in the spleen during infection [245,246]. This receptor was expressed by both B and T cells, with between 52 and 286 reads, and at very low levels in MCs, with 6 and 8 reads.

Cytokine receptor-like 2 (**CRLF2**) forms complexes with the IL-7 receptor alpha chain and binds **TSLP**, an early alarmin [247,248]. This receptor has been detected on many cell types, but in our analysis, it was expressed exclusively and at high levels by MCs, with 194 and 427 reads. NFAT-activating protein with ITAM motif 1 (**NFAM1**) is a receptor that has been reported to be involved in B-cell signaling and development [249]. However, we found no expression of this gene in B cells, but they were detected in monocytes, with 700 and 705 reads and at very low levels in MCs, which is in line with a more recent study showing the effects of this receptor on monocytes’ cytokine production [249].

In this analysis, platelet-activating factor receptor (**PTAFR**) was expressed only at very low levels in monocytes, with 6 and 10 reads, and in MCs, with 53 and 25 reads.

### 2.21. Transcript Levels for Other Receptors

Among other cell-surface receptors, the ATP receptor **P2RX5** was expressed at very high levels in B cells, with 658 and 1081 reads, and at much lower levels in T cells and monocytes and it was absent in MCs (Table 20). This receptor has been shown to be of major importance for osteoclasts and in inflammatory bone loss, but its role in B-cell biology seems less well characterized [250]. Mice lacking this receptor also show defects in inflammasome activation and increased sensitivity to Listeria infection [251]. The receptor for adiponectin, **ADIPOR2**, was weakly expressed in all cells, while MCs displayed higher levels, with 48 and 101 reads (Table 20). This receptor is primarily expressed by the liver and regulates glucose levels and fatty acid breakdown, but to our knowledge, its role in MCs has not been investigated [252]. The estrogen-related receptor alpha **ESRRA** is an orphan nuclear receptor, with no ligand yet identified. It is involved in regulating fat metabolism and absorption [253]. This gene was primarily expressed by monocytes, with 450 and 224 reads, and it was at very low levels in the other cell types. The beta-2 adrenergic receptor **ADRB2** is a G-coupled receptor (GPCR) for epinephrine (adrenalin) and activates cells via adenylate signaling [254]. This receptor was expressed at low levels in all cell types; however, the highest levels were seen in CD8^+^ T cells and MCs, with 73 and 205 reads and 561 and 590 reads, respectively. The downregulation of **ADRB2** is associated with asthma, obesity, and type 2 diabetes [255].

The cannabinoid receptor type 2 (**CNR2**), which also is a GPCR, was expressed essentially only by B cells, with 39 and 60 reads, and it was at very low levels in CD8^+^ T cells (Table 20). This receptor has been shown to have an immunoregulatory function and to be localized to monocytes, macrophages, B cells, and T cells; however, as shown in Table 20, we primarily found it in B cells and not in monocytes, CD4^+^ T cells, or MCs [256]. **CNRIP1** is a cannabinoid receptor-interacting protein that was exclusively expressed by MCs. Retinoic X receptor alpha (**RXRA**) is a receptor for retinoids and acts as a transcription factor for genes involved in lipid metabolism [257]. It showed low-level expression in some of the cell types, but very high levels were observed in monocytes, with 1420 and 1336 reads, and it was also relatively high in MCs, with 181 and 109 reads. Neurophilin 2, **NRP2**, is a member of the neuropilin family of receptor proteins; it has been reported to bind a large number of ligands, including VEGF members and integrins, and to be involved in multiple processes, including cardiovascular development and inflammation [258]. This protein was almost exclusively expressed in MCs, with 87 and 132 reads. **LRPAP1** is a chaperone that is involved in the trafficking of low-density lipoprotein (**LDL**) receptor family members [259]. It was expressed at high levels in monocytes (813 and 1307 reads) and at medium levels in the other cells.

Then, we have a number of GPCRs among which **GPR18** was expressed in both B- and T cells, within an expression in the range of 23 to 87 reads (Table 20). **GPR56** was expressed only in CD8-positive T cells, with 189 and 162 reads, while **GPR141** was detected only in MCs, with 80 and 79 reads, and **GPR174** was only in B and T lymphocytes, with 76 to 156 reads. **GPR18** is a receptor for N-arachidonyl glycyl, an endogenous lipid neurotransmitter with effects on microglial migration and resolvin (**RvD2**), both affecting the resolution of an inflammatory response [260]. **GPR174** has been shown to be a receptor for lysophosphatidylserine and has also previously been associated with hypertensive retinopathy [261,262]. **GPR56**/**ADGRG1** is a member of the adhesion GPCR family, which, in addition to several tissues, has been shown to be a marker for CD8^+^ T cells and a subgroup of NK cells [263]. Finally, **GPR141** is a member of the rhodopsin family of GPCRs, with an unknown function [264]. The neuronal pentraxin receptor **NPTXR** was expressed at low levels only in T cells. This receptor is thought to mediate the neuronal uptake of synaptic material, but its role in T-cell biology is not known [265]. The **MAS1L** gene encodes a GPCR that has been described as an oncogene [266]. Here, it was expressed exclusively in MCs, with 26 and 174 reads.

We assessed two receptors for histamine, **HRH2** and **HRH4**. **HRH2** was only expressed at very low levels, with 5 and 4 reads in monocytes, and **HRH4** only in MCs, with 27 and 7 reads (Table 20). **HRH4** has also been shown to be exclusively expressed by MCs in the mouse [18]. The endothelin receptor B, **EDNRB**, which is also a GPCR, was only expressed in MCs, with 101 and 63 reads (Table 20). This receptor, which is regulated by the transcription factor **MITF**, has a role in the migration of melanocyte and enteric neuron precursors, and a homozygous mutation in this gene in horses causes lethal white syndrome [267].

Dopamine receptors have a number of functions not only in the nervous system but also in immune cells [268]. Dopamine receptor D2, **DRD2**, which is a GPCR, is the main receptor for most antipsychotic drugs [269]. Among these five cell types, it was expressed only in MCs and at relatively low levels, with 35 and 19 reads. In contrast, **DRD5** was only expressed by monocytes and at relatively low levels, with 33 and 18 reads. **AMHR2** is the anti-Müllerian hormone receptor type 2, a receptor for this hormone and testosterone and is involved in sex differentiation. This receptor is responsible for the regression of the Müllerian ducts in mammalian fetuses during male differentiation [270]. It was expressed only by MCs, with 59 and 26 reads. **ADORA3** is a GPCR for adenosine that may be involved in inhibiting neutrophil degranulation, and an antagonist to this receptor may have a therapeutic potential in bronchial asthma [271]. This receptor was expressed only by MCs and at a very low level. **STAB1** is the gene for stabilin-1, a protein that may act as a scavenger receptor, and has been shown to be expressed primarily on sinusoidal endothelial cells of the liver, spleen, and lymph node [272]. We found relatively high levels of **STAB1** in monocytes, with 159 and 192 reads, and lower levels in MCs. **GPBAR1** is a GPCR for bile salts and is implicated in the suppression of macrophage functions [273]. It was expressed only at low levels in monocytes. **PTGIR** is the prostacyclin or prostaglandin I2 receptor, and the activation of this receptor results in potent vasodilation [274]. This receptor was only expressed by monocytes and at relatively low levels, with 36 and 55 reads. **LDLR** is the low-density lipoprotein receptor, which mediates the endocytosis of cholesterol-rich LDL. This receptor was expressed by all cells except B cells, and it was higher in monocytes and MCs than in T cells, in the range of 78 to 535 reads. **LRP1** is the low-density lipoprotein receptor-related protein 1, also known as the alpha-2-macroglobulin receptor (**A2MR**) or the apolipoprotein E receptor (**APOER**). This receptor was almost exclusively expressed by monocytes and at high levels, with 424 and 841 reads. It is involved in numerous cellular processes, including lipoprotein metabolism, clearance of matrix proteinases, and cell motility [275,276,277]. **APOBR** is the apolipoprotein B receptor that binds to the apolipoprotein B48 of dietary triglyceride-rich lipoproteins [278]. Similar to **LDLR**, it was expressed by all of the cells in this analysis, except B cells, and was highest in monocytes. **VDR** is the vitamin D receptor, also known as the calcitriol receptor [279]. It is a member of the nuclear receptor family of transcription factors and is involved in a number of metabolic processes and downstream target genes, including several genes involved in bone metabolism [280]. It was primarily expressed by monocytes, with 222 and 128 reads, and had lower levels in MCs. **Frizzled-1** and **Frizzled-5** are encoded by **FDZ1** and **FDZ5**, both of which are receptors for Wnt signaling, and both were expressed at low levels in monocytes and MCs. Here, the scavenger receptor **SCARF1** was found only in monocytes and MCs at a relatively low level of expression, with 42 and 27 reads and 22 and 21 reads, respectively. This receptor is involved in the clearance of a number of molecules, including acetylated low-density lipoprotein, heat shock proteins, calreticulin, and apoptotic cells [281]. **TSPAN4** showed a similar expression pattern as **SCARF1** with expression essentially only in monocytes and MCs. **TSPAN4** is a member of the transmembrane 4 superfamily that is involved in regulating cell development, activation, growth, and motility and is known to complex with integrins and other members of the transmembrane 4 superfamily [282]. It has also recently been shown to be involved in the formation of migrasomes, a newly discovered cell organelle formed by the local swelling of retraction fibers [283]. **MCOLN1** is encoding mucolipin-1, a receptor that transports iron ions across the endosome/lysosome membrane and that is a transmembrane protein important for lysosome function and vesicular trafficking, exocytosis, and autophagy [284,285]. This gene was likewise expressed primarily in monocytes, with 227 and 133 reads. The plexin-B2 gene **PLXNB2** was expressed at high levels in monocytes, with 898 and 473 reads, and at lower levels in MCs. This protein is essential for the normal differentiation and migration of neuronal cells, but little is known about its function in monocytes [286]. **TRPM2** was also almost exclusively expressed in monocytes and at relatively high levels, with 324 and 243 reads. **TRPM2** is a non-selective calcium-permeable cation channel that is associated with a number of autoinflammatory and metabolic diseases and has also been shown to mediate induction of chemokine production in monocytes [287,288].

### 2.22. Transcript Levels for Calcium, Chloride, and Potassium Channels and Transporters

A number of ion channels and transporters were found to be differentially expressed in the five cell populations (Table 21). These channels and transporters function as gateways for charged ions that cannot freely diffuse across lipid membranes. They are involved in a number of different physiological processes, including the regulation of ion homeostasis, but they can also act as regulators of for example Ca^2+^, which acts as a second messenger for granule release by MCs. **CACNA2D1** and **CACNA2D2** are two voltage-dependent calcium channels (Table 21). **CACNA2D1** was only expressed at low levels in MCs, with 35 and 43 reads, whereas **CACNA2D2** was expressed at low levels in both CD8^+^ T cells and MCs, within a range of 25 to 84 reads (Table 21). **KCTD12** is a BTB/POZ domain protein and a potassium channel that has been shown to be an auxiliary receptor to the GABA-B receptor [289]. We found the expression of this receptor primarily in monocytes at a relatively high level, with 155 and 269 reads. **KCNA3** is a voltage-dependent potassium channel that has been shown to have an essential role in T-cell proliferation and activation [290]. Here, as expected, this channel was expressed by T cells, within a range of 107 to 263 reads, but also at slightly lower levels in B cells. **KCNH8** is another voltage-gated potassium channel that is expressed in the central nervous system. It was expressed at relatively low levels in B cells, with 47 and 71 reads [291]. **ATP1B1** is the gene encoding the sodium/potassium-transporting ATPase beta subunit. This is an enzyme that is responsible for establishing and maintaining the electrochemical gradients of sodium and potassium ions across the plasma membrane [292]. We found the expression of this enzyme at very low levels in all cells, except for MCs, where the expression was much higher, with 187 and 204 reads. The **TTYH3** gene has been reported to encode a Ca^2+^ and cell volume-regulated anion channel; however, conclusive evidence still seems to be lacking for this function [293]. This gene was expressed at high levels in monocytes, with between 500 and 900 reads, and at much lower levels in B cells and MCs (Table 21). **FXYD6** belongs to a family of ion transport regulators that likely affect the activity of sodium/potassium ATPases [294]. This gene was primarily expressed in monocytes, with a range of 60 to 127 reads, and had lower levels in MCs.

### 2.23. Transcript Levels for Angiogenesis Inhibitors and Promoters

In this screening, we looked at the expression levels of two proteins, **VSAH1** and **ENG**, that are involved in angiogenesis but do not belong to the VEGF family (Table 22). The **VSAH1** gene encodes vasohibin an endothelium-derived negative feedback regulator of angiogenesis [295]. Here, this protein was expressed by monocytes at low levels, with 31 and 47 reads, and at very low levels by T cells. The **ENG** gene encodes the protein endoglin, which is a type I membrane glycoprotein that has been shown to have a role in the development of the cardiovascular system and vascular remodeling. The knockout of **ENG** results in the death of mice due to cardiovascular abnormalities [296]. This gene showed a low-level expression primarily in monocytes and MCs, with 113 and 178 reads in MCs.

### 2.24. Transcript Levels for the Sialic Acid-Binding Ig-Lectin Family Members (Siglecs)

Sialic acid has several important regulatory functions, and there are numerous proteins that bind sialic acid moieties. The major family of sialic acid-binding proteins is the sialic acid-binding Ig-lectin family, **SIGLEC**s. Here, we have summarized the expression levels of 15 members of this family in the five cell types. No expression of **SIGLEC1** was seen in any of the five cell types, and a very low level of **SIGLEC4** (also named **MAG**) was observed only in MCs, with three and seven reads (Table 23). **SIGLEC2**, also named **CD22**, and **SIGLC3**, also named **CD33**, were described previously in Table 5. **SIGLEC2** was expressed primarily in B cells as well as by MCs, with 737 and 655 reads for B cells and 146 and 110 reads for MCs. **SIGLEC3** was expressed exclusively by monocytes and MCs at relatively low levels, with between 28 and 83 reads. **SIGLEC3** has been described as myeloid-specific with a function in inhibiting phagocytosis [297]. **SIGLEC5**, also named **CD170**, was expressed primarily by monocytes, with 47 and 58 reads, but also at low levels by B cells and MCs (Table 23). **SIGLEC5** binds to linked sialic acid as well as lipid compounds and has been shown to have a role in inhibiting the activation of human monocytes [298,299]. An important paralog of **SIGLEC5** is **SIGLEC14**, which had a very similar expression pattern, with a minor difference by a higher expression in B cells [300]. **SIGLEC6** was almost exclusively expressed by MCs, with 484 and 289 reads, and at a very low level by B cells (Table 23). Its expression on the surface of MCs has been shown previously at the protein level, but no physiological ligand to this protein with a connection to MCs has been identified, and it does not appear to be dependent on the glycolyl group of sialic acid [301,302]. **SIGLEC7**, or **CD328**, was produced only at low levels in monocytes, with 15 and 32 reads, and even lower in MCs, with 6 and 5 reads. **SIGLEC7** binds alpha-linked sialic acid and di-sialo-gangliosides and is predicted to have an inhibitory role in immunity, primarily on NK cells and monocytes [303]. **SIGLEC7** is a paralog of **SIGLEC9**, which shows a very similar expression pattern with medium expression in monocytes, with 70 and 70 reads and lower in MCs. Very little is known about the function of this molecule. **SIGLEC8** was expressed in MCs, with 129 and 33 reads. This is in line with previous studies where it has been shown to be expressed almost exclusively by eosinophils and MCs and to a lower extent in basophils [304]. The ligand to **SIGLEC8** has not been determined but it binds strongly to 6-sulfo-sialyl Lewis X, and the cross-linking of **SIGLEC8** has been shown to inhibit the release of prostaglandin D2 and histamine by MCs [305]. **SIGLEC10** and **11** showed very similar patterns, with between 90 and 130 reads in monocytes and lower levels in B cells. They are paralogs that both bind alpha-linked sialic acid residues, and both have a cytoplasmic immune cell tyrosine-based inhibitory motif (**ITIM**). Additionally, at least **SIGLEC10** may have an inhibitory function in controlling the response to danger-associated molecular patterns (**DAMPs**) [306]. **SIGLEC**s **12**, **15**, and **16** were essentially negative in all five cell populations of this study.

### 2.25. Transcript Levels for S100 Proteins

The members of the relatively large family of **S100** proteins contain a calcium-binding motif. This family includes more than 20 members in which the encoded proteins are located in the cytoplasm or nucleus of a wide range of cells [307]. They are only found in vertebrates and are involved in the regulation of proliferation, differentiation, apoptosis, Ca^2+^ homeostasis, energy metabolism, inflammation, and migration/invasion through interactions with a variety of target proteins [308].

**S100A4** was expressed in all five cell types and at relatively high levels, within a range of 147 to 1189, except for in B cells, where we observed a lower level of expression (Table 24). **S100A4** has been shown to interact with another member of this family, namely **S100A1** [309]. **S100A6** showed an almost identical pattern of expression as **S100A4**, with only slightly lower expression levels. **S100A6** interacts with a number of molecules in a calcium-dependent manner and has been indicated to take part in a number of cellular processes [310]. Both **S100A8** and **S100A9** were here essentially monocyte-specific and expressed at a very high level, in the range of 600 to 3600 reads (Table 24). They have been shown to be expressed constitutively in monocytes and neutrophils and to act as calcium sensors participating in cytoskeletal rearrangements and arachidonic acid metabolism [311]. **S100A10** showed an expression pattern very similar to **S100A4** and **S100A6,** with relatively high expression in all cells except in B cells, where the expression level was lower. **S100A10** is unique among S100 proteins in that it does not bind calcium. However, it has been shown to interact with a number of other molecules and to have several intracellular functions similar to other members of this family; it primarily seems to exist in a heterotetrameric complex with annexin A2 [307]. The expression of **S100Z** was only detected in monocytes, within a range of 30–40 reads.

**S100A11** showed low-level expression in monocytes and T cells and much higher levels in MCs, within the range of 300 to 750 reads, whereas **S100A12** was exclusively expressed by monocytes and in the range of 250 reads. **S100A11** has been implicated in vesicular transport and membrane and cytoskeletal dynamics, and it interacts with a number of proteins, including tubulin, actin, intermediate filaments, and annexin I and II [312,313,314]. **S100A12** has been shown to be expressed and secreted primarily by neutrophils and to have cytokine-like activity; it interacts with the receptor for advanced glycosylation end products (**RAGE**) and plays a role in proinflammatory reactions [315].

### 2.26. Transcript Levels for Cell Adhesion Molecules and Other Membrane Proteins

Cell adhesion molecules are essential for organ formation, cell migration, and many other cellular processes, and there are several large gene families of such molecules. One of them comprises integrins, of which we list the expression levels of 20 members (Table 25). Some integrins were relatively highly expressed in all five cell types, such as **ITGB1,** with expression levels varying between 82 and 331 reads, whereas others are totally absent, such as **ITGB5**, **ITGB6**, **ITGB8**, **ITGA2**, **ITGA7**, and **ITGA8** (Table 25). **ITGB2** was very highly expressed in monocytes, with 1079 and 1024 reads, and was also very high in both CD4^+^ and CD8^+^ T cells, with a range of 349 to 996 reads, and had lower levels in B cells, with 137 and 120 reads, and very low in MCs, with 10 and 7 reads (Table 25). In contrast, **ITGB3** was primarily expressed by MCs but at low levels, with 23 and 34 reads (Table 25). **ITGB4** was only expressed at very low levels by MCs, whereas **ITGB7** was expressed only by B and T cells, with between 16 and 71 reads (Table 25). **ITGA1** was only seen in T cells and at very low levels (Table 25). **ITGA3** was expressed primarily in MCs, with 75 and 98 reads, and much lower in lymphocytes. **ITGA4** was instead expressed at relatively high levels in all five cell types, with between 71 and 257 in lymphocytes and monocytes. **ITGA5** was most highly expressed in monocytes and MCs, with between 162 and 447 reads, and was lower in T cells and almost absent in B cells. **ITGA6** was most highly expressed by CD4^+^ T cells, with 106 and 135 reads, and was lower in CD8^+^ T cells and MCs and absent in both B cells and monocytes. **ITGA9** was detected only in MCs, with 127 and 148 reads, and **ITGAL** showed the opposite pattern by being expressed at high levels in all cells except MCs. **ITGAM** showed expression primarily in monocytes, with 387 and 292 reads, and was lower in MCs, with 146 and 108 reads. **ITGAV** was also essentially only expressed by MCs, with 29 and 148 reads. Finally, **ITGAX** was found to be expressed essentially only by monocytes and MCs, with between 84 and 231 reads. We observed a very large variability in the expression of integrins in this study. Since the expression level of integrins can change quite markedly by activation, it is vital to handle the cells very carefully upon purification to obtain information concerning their true in vivo expression profile.

Another family of cell adhesion molecules is the intercellular adhesion molecules **ICAMs**, which belong to the immunoglobulin superfamily of proteins. Most of them were expressed at relatively low levels, with **ICAM1** at very low levels primarily in monocytes and MCs (Table 25). **ICAM2** was instead broadly expressed but also at relatively low levels, within the range of 13 to 82 reads (Table 25). **ICAM3** was also relatively broadly expressed, in the range of 42 to 278 reads, except for being absent in MCs. (Table 22). Both **ICAM4** and **5** were only expressed at very low levels by monocytes and MCs (Table 25).

The focal adhesion protein paxillin, encoded by the **PXN** gene, is a signal transduction adaptor protein that interacts directly with the cytoplasmic tail of beta integrins [316]. **PXN** was expressed by all five cell types, with between 127 and 405 reads in T cells and MCs, and it had lower levels in monocytes and very low levels in B cells. The P selectin gene **SELP** was almost totally absent in all five cell types, with expression levels not exceeding three reads (Table 25). In contrast, its ligand **SELPLG** was relatively highly expressed by monocytes, T cells, and MCs, within the range of 21 to 192 reads, and it was almost absent in B cells. This adhesion protein is of major importance for the interaction of immune cells with blood vessel endothelium and platelets, which under inflammatory conditions, express P and E selectin [317]. The gene **CELSR1**, which encodes cadherin family member 9, was instead almost exclusively expressed by B cells, with 63 and 129 reads. This protein has been shown to have an important role in early brain development, but its role in B-cell development and function seems to be less well known [318]. Protocadherin FAT1 is encoded by the ***FAT1*** gene and was expressed exclusively by MCs but at low levels, with 34 and 36 reads. It has been mainly associated with fetal epithelia, and its knockout in mice resulted in perinatal death due to renal failure [319]. The gene encoding the epithelial membrane protein, **EMP1**, was expressed almost exclusively in MCs and at a relatively high level, with 333 and 517 reads. The tight junction protein **ZO-2**, encoded by the ***TJP2*** gene, was also almost exclusively expressed by MCs, with 60 and 110 reads. It is a cytoplasmic protein that interacts with several tight junction proteins, including occludin [320]. The **MAL** gene encodes a proteolipid that is found in the endoplasmic reticulum of T cells and in myelin cells, where it is thought to be involved in T-cell receptor signaling in the former [321]. It was expressed at relatively high levels almost exclusively by T cells and at approximately the same levels in CD4^+^ and CD8^+^ T cells, within the range of 179 to 488 reads. Transmembrane glycoprotein NMB and its mouse ortholog osteoactivin is encoded by the **GPNMB** gene. In the present study, this type I glycoprotein, which has been shown to be expressed in melanocytes, osteoclasts, osteoblasts, and dendritic cells, was expressed exclusively in MCs and at relatively high levels, with 467 and 543 reads (Table 25) [322]. The **GPNMB** gene has been shown to be regulated by the transcription factor **MITF**, which most likely is the reason why it is also expressed in MCs [323]. Transmembrane protein 176B, encoded by the **TMEM176B** gene, is thought to have a role in the maturation of dendritic cells [324]. Here, it was expressed at very high but variable levels in monocytes, with 149 and 1605 reads, and at low levels in MCs, with 34 and 36 reads. **AMICA1**, or junctional adhesion molecule like (**JAML**), binds and activates the coxsackie and adenovirus receptor (human CXADR and mouse CAR), which is needed for neutrophil extravasation [325]. This gene was expressed almost exclusively by monocytes, with 85 and 188 reads. SID1 transmembrane family member 1 is encoded by the ***SIDT1*** gene and is a transmembrane dsRNA-gated channel that facilitates the transport of dsRNA into cells and is required for systemic RNA interference [326]. It was expressed only by B and T lymphocytes, within a range of 29 to 50 reads (Table 25). Aquaporin 3, encoded by the ***AQP3*** gene, is a pore-forming protein that allows for the exit of water but also glycerol, ammonia, urea, and hydrogen peroxide [327]. Here, it was expressed almost exclusively by T cells and at similar levels in both CD4^+^ and CD8^+^ T cells, in the range of 170 to 393 reads. The **ITM2A** gene encodes the integral membrane protein 2A, which has been shown to be involved in the activation of T cells [328]. It was primarily expressed by T cells, within a range of 94 to 290 reads, but also by MCs with only slightly lower levels and at very low levels in B cells.

### 2.27. Transcript Levels for Cell Signaling Proteins

A complex network of cell signaling proteins is involved in transferring the information received from external signals, such as the binding of cytokines and chemokines to their receptors and the interaction with other cells through cell adhesion molecules. A large number of such signaling proteins were identified in this screening of the five cell types. In general, cell signaling by a majority of these proteins is regulated by tyrosine or serine/threonine phosphorylation, but as suggested by our findings, they may also be regulated by the level of transcription and thereby also by the levels of these proteins in the cytoplasm of these cells (Table 26). We found that some of these signaling proteins were expressed exclusively in one or a few of these cell types. **BLK** was almost exclusively expressed in B cells, and **LCK**, **THEMIS**, and **ZAP70** were exclusively expressed by T cells (Table 26). Interestingly, some other signaling proteins, like **BTK**, which has an important role in B-cell biology, were also expressed by monocytes in addition to B cells, actually at 2–3 times higher levels in MCs than in B cells (Table 26). Due to the massive size of our report, here, we will just present the reads for this large set of signaling molecules without further details concerning their direct functions. However, we think the expression data provide very valuable information for further analysis of the complex network of intracellular signaling in various immune cells, which is why we keep the entire list for facilitating future analysis. A general observation was that there were very minor differences between CD4^+^ and CD8+ T cells in the expression of these signaling proteins, indicating that the signaling machinery in these two subtypes of T cells is very similar. This was in contrast to the clear differences found between monocytes, B cells, and MCs.

### 2.28. Transcript Levels for Apoptosis-Related Proteins

Regulated cell death is an important cellular mechanism during organ development and infection and involves both pro- and anti-apoptotic proteins. One of the inhibitory proteins is the Fas apoptosis inhibitory molecule 2, which is encoded by the **FAIM2** gene. Transcripts of this inhibitor of Fas/CD95 mediated apoptosis have previously primarily been found in various brain cells, and here, they were found only in MCs, with 48 and 50 reads (Table 27) [329]. **BCL2** is another anti-apoptotic protein that was highly expressed by both B and T lymphocytes, within the range of 251 and 432 reads, and they were lower in MCs with 66 and 75 reads and almost absent in monocytes. The anti-apoptotic protein Mcl-1, encoded by the **MCL1** gene, which has been shown to be the most important anti-apoptotic protein for skin MCs, was found to be expressed at high levels in all five cell types but was much higher in MCs, with 2704 and 3365 reads [330]. ***CARD11*** encodes a CARD domain protein that interacts with **BCL10**, is activated upon B- and T-cell receptor stimulation, and is critical for B- and T-cell activation [331]. It was expressed exclusively in B and T cells at relatively high levels, within the range of 198 to 293 reads (Table 27). Clusterin, encoded by the ***CLU*** gene, is an extracellular molecular chaperone and can also be found in the cytosol. It binds to misfolded proteins in body fluids, neutralizes their toxicity, and mediates their uptake by receptor-mediated endocytosis; its overexpression can protect cells from apoptosis induced by cellular stress [332]. **CLU** was expressed exclusively by MCs and at very high levels, with 799 and 958 reads. As for many genes in this analysis that have shown an interesting expression pattern, further studies concerning their role in the biology of these cells are warranted.

### 2.29. Transcript Levels for Matrix Proteins

Immune cells are not generally considered major players in matrix protein production. However, we found that some members of the large collagen family were expressed at relatively high levels in B cells, T cells, and MCs, and other matrix proteins were also detected in other cells. The two collagens **COL4A3** and **COL19A1** were exclusively expressed by B cells but at quite different expression levels, with 31 and 54 reads for **COL4A3** and 403 and 447 reads for **COL19A1** (Table 28). **COL4A3** is one of the subunits of collagen type IV, the major component of basement membranes. **COL19A1** is a subunit of a fibril collagen of unknown function. **COL6A2** is a subunit of beaded filament collagen found in most connective tissues. Here, it was expressed by T cells and MCs, with a major difference in expression levels between CD4^+^ and CD8^+^ T cells, with 23 and 39 reads in CD4^+^ T cells and 119 and 190 reads in CD8^+^ T cells. **COL13A1** is the alpha chain in one of the membrane-bound nonfibrillar collagens, a collagen of unknown function that was expressed exclusively by MCs at a relatively low level. **FERMT2**, also known as kindlin-2, is a component of the extracellular matrix and has a role in regulating the activation of integrins, and here, it was expressed exclusively by MCs, with 92 and 103 reads (Table 28) [333]. Extracellular microfibril interface 2 is encoded by the **EMILIN2** gene. The encoded protein is an extracellular glycoprotein that is predicted to confer elasticity of the extracellular matrix [334]. Here, it was expressed at high levels by both monocytes and MCs, with expression levels between 117 and 750 reads and at much lower levels in B cells. Laminin subunit alpha-5 is encoded by the gene ***LAMA5***, which was expressed at relatively low levels in B cells and MCs, with between 27 and 70 reads. This protein exerts a myriad of effects, including mediating the attachment, migration, and organization of cells in various tissues [335]. These findings indicate a larger role of various immune cells in the regulation of the extracellular matrix not only by directly interacting with the matrix cells but also by producing selective matrix components.

### 2.30. Transcript Levels for Solute Carriers

Cells need controlled access to ions, amino acids, glucose, nucleosides, and other small molecules. The large family of solute carriers plays a major role herein, and entities of this family showed a complex expression pattern in these five cell types (Table 29). **SLC1A5** is a sodium-dependent amino acid transporter with a broad substrate range. **SLC1A5** was expressed in the five cell types but at a higher level in MCs, with 344 and 376 reads, and it was lower in monocytes and much lower in the other cell types (Table 29). **SLC2A9** is a glucose transporter. It was expressed almost exclusively by monocytes, at a level of 51 and 118 reads. **SLC6A8**, which is a creatine transporter, was expressed only by MCs, with 30 and 72 reads. **SLC7A7** is a sodium-independent amino acid transporter of basic and large neutral amino acids from the cytoplasm to the extracellular space. This transporter was found primarily in monocytes, with 102 and 95 reads, and at a lower level in B cells. **SLC8A3** is a Na^+^/Ca^2+^ exchange protein involved in maintaining Ca^2+^ homeostasis. This protein was expressed exclusively in MCs, with 47 and 10. **SLC9A1**, alias **NHE1**, is a membrane-bound Na^+^/H^+^ transporter involved in volume and pH regulation [336]. **SLC9A1** was broadly expressed but with higher levels in MCs, with 210 and 150 reads. The **SLC9A7** gene encodes a sodium/potassium antiporter that is located primarily in the trans-Golgi network and is involved in maintaining pH in organelles [337]. Here, it was expressed primarily in B cells, with 244 and 258 reads. The ***SLC9A9*** gene was expressed at low levels in all five cell types. The ***SLC9A9*** gene encodes a sodium/proton exchanger that is localized to late recycling endosomes and that may play an important role in maintaining cation homeostasis [338]. The ***SLC12A4*** gene encodes a potassium–chloride transporter that was expressed in all five cell types but higher in monocytes with 151 and 185 reads (Table 29) [339]. The ***SLC15A3*** gene encodes a dipeptide transporter that transports histidine, certain dipeptides, and peptidomimetics from inside the lysosome to the cytosol. Here, it was expressed in all five cells but at much higher levels in monocytes, with 154 and 173 reads, which is in line with previous studies showing that this gene is primarily expressed by macrophages [340]. The monocarboxylate transporter 4 (MCT4) is encoded by the ***SLC16A3*** gene. This gene was expressed in all five cell types but was higher in monocytes and MCs, with between 19 and 181 reads. The vesicular monoamine transporter 2 (VMAT2) is a protein that, in humans, is encoded by the ***SLC18A2*** gene. This protein is an integral membrane protein that transports monoamines such as dopamine, norepinephrine, serotonin, and histamine from cytosol to secretory vesicles [341]. It was expressed exclusively by MCs and at a very high level, with 698 and 702 reads (Table 29). The ***SLC18B1*** gene is a sister gene to the previous ***SLC18A2*** and to other vesicular monoamine and acetylcholine transporters, and it is the only known polyamine transporter, with an unknown physiological role [342]. ***SLC18B1*** was expressed by all five cell types at very low levels but with slightly higher expression in lymphocytes. The ***SLC25A44*** gene encodes a transporter of branched amino acids, valine, leucine, and isoleucine, into mitochondria in brown fat tissue for thermogenesis [343]. It was expressed at low levels in all cell types except MCs, where the expression was high, with 189 and 467 reads. Equilibrative nucleoside transporter 1 (**ENT1**) is a protein that, in humans, is encoded by the ***SLC29A1*** gene. This transporter, which mediates the cellular uptake of nucleosides from the surrounding medium, was expressed by all five cell types but at higher levels in MCs, with 70 and 102 reads (Table 29) [344]. In humans, the zinc transporter 1 is encoded by the ***SLC30A1*** gene. It was expressed by all cells but at higher levels in monocytes and MCs, within a range of 50 to 98 reads. This protein downregulates not only Zn^2+^ influx but also Ca^2+^ influx, thereby protecting cells from the effects of excessive cation permeation [345]. The sodium-coupled neutral amino acid transporter 1 is encoded by the ***SLC38A1*** gene. It is an important transporter of glutamine, an intermediate in the detoxification of ammonia and the production of urea [346]. **SLC38A1** was expressed almost exclusively by lymphocytes and at relatively high levels, with between 74 and 186 reads. The protein encoded from the ***SLC39A11*** gene (also named **ZIP11**) is predicted to be involved in the transport of zinc across the Golgi, nucleus, and plasma membranes [347]. It was expressed in all five cell types, but was higher in monocytes, with 68 and 260 reads. The ***SLC40A1*** gene encodes the so far only known iron exporter from cells to blood [348]. It was expressed at very low levels in monocytes and CD8^+^ T cells, not in B cells, and it was higher in CD4^+^ T cells and MCs, within a range of 33 to 68 reads (Table 29). The protein encoded by the ***SLC40A3*** gene has been shown to mediate the uptake of purine nucleobases such as adenine, guanine, and hypoxanthine without requiring typical driving ions such as Na^+^ and H^+^, but it does not mediate the uptake of nucleosides [349]. It was expressed by all five cell types but at much higher levels in MCs, with 352 and 387 reads. Choline transporter-like protein 1 is a protein that is encoded by the ***SLC44A1*** gene, where the loss of function mutations has implicated its role in ethanolamine transport [350]. It was expressed at very low levels in all five cell types except in MCs, which expressed high levels, with 118 and 203 reads.

### 2.31. Transcript Levels for Cell Cycle and Immediate-Early Response-Related Proteins

Depending on the proliferative potential of a cell and the state of activation, the cell expresses different levels of cell cycle and immediate-early response-related genes. ***CDK14*** is one such gene that is predicted to encode a cyclin-dependent serine/threonine kinase, involved in the G2/M transition of the mitotic cell cycle and the regulation of the canonical Wnt signaling pathway. This gene was only expressed by B cells, with 113 and 144 reads (Table 30). Another such protein is the G0/G1 switch 2 protein, which is encoded by the ***GOS2*** gene. It is located in the mitochondrion and is involved in positive extrinsic apoptosis signaling [351]. Here, it was expressed almost exclusively by monocytes and MCs, at a very high level in monocytes, with 850 and 2437 reads, and at a lower level by MCs, with 45 and 252 reads.

Cyclins and cyclin-dependent kinases (CDKs) were in general expressed at relatively low levels, except for **CDK4** and cyclin D1 (**CCND1**) in MCs, with 129 and 152 reads and 131 and 178 reads, but also cyclin D2 and D3 in all five cell types, with between 18 and 833 reads (Table 30).

The ***IER3*** gene encodes the radiation-inducible immediate-early gene IEX-1, a protein that functions in the protection of cells from FAS or tumor necrosis factor alpha-induced apoptosis [352]. It was expressed at very low levels in B and T cells and at very high levels in both monocytes and MCs, within a range of 510 and 1693 reads (Table 30). Early growth response protein 3 is a protein encoded by the ***EGR3*** gene. It was expressed almost exclusively by MCs and at a high level, with 194 and 494 reads. The gene encodes a transcriptional regulator that belongs to the EGR family of C2H2-type zinc finger proteins. It is an immediate-early growth response gene that is induced by mitogenic stimulation [353].

### 2.32. Transcript Levels for Nuclear Proteins and Splicing Factors

Nuclear proteins and splicing factors are of importance for nuclear organization, transcript transport, accurate splicing, and regulating differential splicing. Pleckstrin homology-like domain family A member 1, also known as **TDAG51**, is a protein that is encoded by the ***PHLDA1*** gene. This protein is an evolutionarily conserved proline–histidine-rich nuclear protein, which may play an important role in the anti-apoptotic effects of insulin-like growth factor 1 [354]. It was expressed at very low levels in monocytes and T cells and at higher levels in MCs, with 154 and 220 reads (Table 31). The ***AHNAK2*** gene encodes a large nuclear protein that may play a role in calcium signaling by associating with calcium channel proteins. It was expressed exclusively by MCs, with 75 and 154 reads. Human Schlafen family member 5 is a transcriptional corepressor of STAT-1-mediated interferon responses and is encoded by the ***SLFN5*** gene [355]. It was expressed at low levels in B cells and monocytes and at higher levels in T cells and MCs, with between 65 and 208 reads. Prelamin-A/C, or lamin A/C is a nuclear lamina protein, whereby the lamina consists of a two-dimensional matrix of proteins located next to the inner nuclear membrane [356]. The gene encoding this protein, ***LMNA***, was expressed by all five cell types, with between 15 and 77 reads in the lymphocytes, between 227 and 840 reads in the monocytes, and very high levels in the MCs with 4302 and 6983 reads (Table 31). B-cell CLL/lymphoma 7 protein family member A is a protein encoded by the ***BCL7A*** gene. This protein belongs to the SWI/SNF chromatin remodeling complex, a complex that is able to modify the interactions between DNA and histones [357]. It was expressed at very low levels in all of these cells except for in B cells, where we detected 43 and 87 reads.

The splicing factor encoded by the ***SF3B4*** gene was expressed by all five cell types but was considerably higher in monocytes, with 102 and 110 reads.

### 2.33. Transcript Levels for Cytoskeleton-Related Proteins

Cytoskeletal proteins are of major importance for many cellular processes such as vesicle transport, cell shape, cell division, and migration. Cortactin-binding protein 2 is a protein with six ankyrin repeats and several proline-rich regions that may act as a regular of the actin cytoskeleton, and it is encoded by the ***CTTNBP2*** gene [358]. It was exclusively expressed by MCs with 116 and 138 reads (Table 32). Tensin-1 is a protein that localizes to focal adhesions and is encoded by the ***TNS1*** gene [359]. It was expressed primarily in MCs at very high levels, with 683 and 848 reads, and also at low levels in monocytes. Myosin-IF is a protein that is encoded by the ***MYO1F*** gene. It was expressed mainly in the immune system and might be involved in cell adhesion and motility [360]. It is expressed by all five cell types at levels except for in monocytes where the levels were much higher with 868 and 2491 reads (Table 32). We may speculate that this protein contributes to the high mobility of monocytes in infected tissues. Another myosin, ***MYO10***, was expressed by MCs only, with 85 and 110 reads. The protein encoded by this gene is an actin-based motor protein that can localize to the tips of the finger-like cellular protrusions known as filipodia [361]. Tubulin polymerization-promoting protein 3 (**TPPP3**) is a member of the tubulin polymerization family. It was expressed only by monocytes and MCs, within a range of 48 to 179 reads. Lymphocyte-specific protein 1 is a protein that is encoded by the ***LSP1*** gene. It is an intracellular F-actin-binding protein that is expressed by lymphocytes, neutrophils, macrophages, and endothelium, and it may regulate neutrophil motility, adhesion to fibrinogen, matrix proteins, and transendothelial migration [362]. It was expressed at very high levels in the lymphocyte populations, with between 917 and 1459 reads, very low in MCs, and very high also in monocytes, with 4812 and 5211 reads (Table 32). **Drebrin** is a cytoplasmic actin-binding protein that is encoded by the ***DBN1*** gene. It was expressed primarily by CD8^+^ T cells and MCs, within a range of 65 to 120 reads. PDZ and LIM domain protein 7 (also named **ENIGMA**) is a protein that is encoded by the ***PDLIM7*** gene. This protein is involved in the assembly of an actin filament-associated complex essential for the transmission of ret/ptc2 mitogenic signaling [363]. Here, it was expressed by all five cell types but was much higher in monocytes and MCs, where the expression was between 95 and 216 reads (Table 32). The ***TNNI2*** gene encodes the fast-twitch skeletal muscle troponin I. It was expressed at very low levels in B cells and MCs and at much higher levels in monocytes, with 171 and 439 reads. The ***ADD2*** gene encodes beta-adducin, a protein involved in the assembly of spectrin–actin network in erythrocytes and at sites of cell–cell contact in epithelial tissues [364]. It was only expressed by B cells and at low levels. Tubulin beta 6, encoded by the ***TUBB6*** gene, controls microtubule and actin dynamics [365]. It was expressed at relatively low levels in B cells, monocytes, and MCs, with between 23 and 145 reads. Gelsolin is an actin-binding protein that is a key regulator of actin filament assembly and disassembly, encoded by the ***GSN*** gene. It was expressed in all five cells but at very low levels by lymphocytes and much higher by monocytes and MCs, with between 135 and 205 reads. It has been found to play an important role in host innate immunity by activating and localizing macrophages to sites of inflammation [366].

### 2.34. Transcript Levels for Vesicle and Protein Transport

Several immune cells store large amounts of material in cytoplasmic vesicles, which are released upon receptor triggering, such as upon allergen stimulation in MCs and upon targeting virus-infected cells in cytotoxic T cells. Proteins involved in vesicle transport and the transport of proteins into various types of vesicles are then of importance in this process. The amount of such proteins may vary considerably across different cells. One such example is the ***NSG1*** gene, which encodes the neuronal vesicle trafficking associated 1, a protein located in the endoplasmic reticulum that is predicted to enable clathrin light chain binding activity. It was expressed almost exclusively by T cells with between 52 and 13 reads (Table 33). The ***STX3*** gene encodes the protein syntaxin 3, which is a member of the syntaxin family of cellular receptors for transport vesicles that participate in exocytosis in neutrophils [367]. It was expressed at very low levels in all cells except for in MCs, where the expression was higher, with 47 and 149 reads. The ***DYNLL1*** gene encodes the dynein light chain 1, a cytoplasmic protein involved in intracellular transport and motility [368]. It was expressed at relatively low levels in all cells, except for in MCs, where the expression was at least 10-fold higher, with 457 and 756 reads. Syntaxin 11, a protein encoded by the ***STX11*** gene, is a member of the t-SNARe family involved in intracellular membrane trafficking. The enhanced expression of Syntaxin 11 has been shown to augment the secretion and killing of tumor targets by NK cells and CTLs [369]. It was expressed in all cells but 5–10 times higher in monocytes and MCs, within between 196 and 327 reads (Table 33). The ***SNX21*** gene encodes sorting nexin-21, a protein involved in intracellular trafficking. It was expressed essentially only in monocytes, with 88 and 90 reads, and was lower in MCs. Another member of this family, ***SNX22***, was expressed exclusively in B cells, with 82 and 210 reads. The ***VPS37C*** gene encodes a subunit of ESCRT-I (endosomal sorting complex required for transport I), a complex in the class E vacuolar protein sorting (VPS) pathway required for sorting ubiquitinated transmembrane proteins into internal vesicles of multivesicular bodies [370]. ***VPS37C*** was expressed at very low levels in lymphocytes, but was higher in MCs and much higher in monocytes, with 167 and 177 reads. The ***AP5B1*** gene encodes the beta subunit of the AP-5 complex of clathrin adaptor proteins involved in vesicular transport [371]. It was expressed in all cells at low levels except for in monocytes, where the levels were approximately 10 times higher, with 457 and 588 reads. A similar situation was seen for the translocator protein, encoded by the ***TSPO*** gene, a protein that may be involved in transporting cholesterol into mitochondria. It was expressed in all five cell types but was 10 times higher in monocytes, with 955 and 967 reads.

The EH domain binding protein 1 like 1, encoded by the ***EHBP1L1*** gene, was also expressed in all five cell types but was 25–50 times higher in monocytes with 5040 and 5041 reads. This protein is located in the cell membrane and is predicted to be involved in cytoskeletal organization and potentially in vesicle trafficking. The reason why monocytes need so high level of this protein compared to MCs and cytotoxic T cells is intriguing but not known.

### 2.35. Transcript Levels for Endogenous Retroviruses and Oncogenes

Finally, we assessed the expression levels of an endogenous retrovirus and a few oncogenes, as shown in Table 34.

The ***ERVFRD-1*** gene is part of a human endogenous retrovirus provirus on chromosome 6 that has inactivating mutations in the gag and pol genes. It was expressed exclusively in MCs, with 85 and 174 reads (Table 34).

Deleted in liver cancer 1, encoded by the ***DLC1*** gene, was also expressed exclusively in MCs and at a very high level, with 419 and 690 reads. It is a candidate tumor suppressor gene [372]. Tyrosine–protein kinase Fes/Fps is also known as proto-oncogene c-Fes/Fps. The ***FES*** gene encodes the human cellular counterpart of a feline sarcoma retrovirus protein. ***FES*** was expressed primarily in monocytes, with 154 and 171 reads, and was lower in MCs, with 56 and 60 reads.

## 3. Materials and Methods

### 3.1. Purification of Human Peripheral Blood B Cells, Monocytes, and CD4^+^ and CD8^+^ T Cells by FACS Sorting

Enriched peripheral blood mononuclear cells (PBMCs) were obtained from buffy coats of anonymized blood donors at Uppsala University Hospital. Platelets were removed by centrifugation (2 × 200 g, 10 min). The white blood cells were further enriched using Ficoll–Paque Premium (ρ = 1.076 g/mL) (GE Healthcare, Little Chalfont, UK) in SepMate™-50 tubes (Stemcell Technologies, Vancouver, Canada). Enriched PBMCs were vital frozen at −80 °C in heat-inactivated fetal calf serum (FCS) (Sigma-Aldrich, St. Louis, MO, USA) with 10% of DMSO. After thawing, the PBMCs were incubated in FACS buffer (PBS, pH 7.4, containing 2% heat-inactivated FCS) and stained with fluorescent-conjugated antibodies from BD Bioscience and eBioscience, San Diego, CA, USA, targeting (clone names) CD4 (RPA-T4), CD8 (RPA-T8), CD14 (M5E2), and CD19 (HIB19). Cell sorting was performed on a FACSAria III (BD Biosciences). One million cells of each population were collected in FACS buffer and pelleted for RNA preparation, and a small fraction was used to acquire the purity check controls. Data analysis was performed using FlowJo software version 9.8. Total RNA was prepared using the RNeasy Plus mini kit from (Qiagen, Hilden, Germany), according to the manufacturer’s recommendations. The RNA was eluted with 30 μL of DEPC-treated water, and the concentration of RNA was determined by using a Nanodrop ND-1000 (Nano Drop Technologies, Wilmington, Delaware, USA). Later, the integrity of the RNA was confirmed by visualization on 1.2% agarose gel using ethidium bromide staining.

### 3.2. Purification of Human Skin MCs

MC purification was performed as previously described [218,373], with modifications specified in more recent work [374]. Skin was cut into strips and treated with dispase (24.5 mL per preparation, activity: 50 U/mL; Corning, Kaiserslautern, Germany) at 4 °C overnight. After the removal of the epidermis, the dermis was chopped into small pieces and digested with 2.29 mg/mL collagenase (Worthington, Lakewood, NJ, USA), 0.75 mg/mL hyaluronidase (Sigma, Deisenhofen, Germany), DNase I at 10 µg/mL (both from Roche, Basel, Switzerland), and 5 mM MgSO_4_ for 1 h at 37 °C.

The cell suspensions were separated from remaining tissue by three steps of filtration. In the case of breast skin, the undigested tissue still remaining after the first digestion was subjected to a second digestion step of 1 h at 37 °C after the first filtration. MC purification from the dispersates was achieved by positive selection with anti-human c-Kit microbeads and an Auto-MACS separation device (both from Miltenyi Biotec, Bergisch Gladbach, Germany). MC purity always exceeded 98%, as assessed by acidic toluidine-blue staining (0.1% in 0.5 N HCl). Viability by trypan blue exclusion exceeded 99%. We used between 4.8 and 6.2 × 10^6^ MCs for one RNA isolation (ex vivo samples). Cultures were started from 1.5–6 × 10^6^ MCs, and around 3 × 10^6^ were eventually used for one RNA preparation.

### 3.3. RNA Isolation and Heparinase Treatment of Human MCs

Total RNA was prepared from freshly isolated MCs following an established protocol for each preparation. Briefly, MCs were lysed in 700 µL QIAzol^®^lysis reagent (Qiagen, Hilden, Germany), mixed with 140 µL chloroform (Sigma) and 60 µL DEPC-treated water and transferred to a 2 mL gel tube (Quanta bio/VWR, Dresden, Germany). After centrifugation, the supernatant was transferred to a NucleoSpin^®^ filter, and RNA was isolated using the NucleoSpin RNA kit from Machery-Nagel (Düren, Germany) following the manufacturer’s instructions. For heparinase (BioLab, Braunschweig, Germany) treatment, the resulting RNA solution was mixed with RNAse inhibitor (Thermo Fisher Scientific) and heparinase buffer (BioLab, Braunschweig, Germany) and incubated for 3 h at 25 °C. Another RNA isolation procedure was followed, using the NucleoSpin RNA kit from Machery-Nagel (Düren, Germany) according to the manufacturer’s protocol. To further concentrate preparations, RNA was precipitated overnight at −80 °C using 100% ethanol and sodium acetate (Merck, Darmstadt, Germany). RNA of each preparation was eventually solved in 20 µL DEPC-treated water. After each treatment step, RNA concentration was determined by using a Nanodrop ND-1000 (Nano Drop Technologies, Wilmington, Delaware, USA).

### 3.4. Ampliseq Analysis of the Total Transcriptome

The transcriptome of the four different peripheral blood cell fractions and the freshly isolated MCs were analyzed for their total transcriptome by the Thermo-Fisher chip-based Ampliseq transcriptomic platform at the SciLife lab in Uppsala, Sweden (Ion-Torrent next-generation sequencing system; Thermofisher.com). The sequence results were delivered in the form of a large Excel file.

### 3.5. Quantitative Transcriptome Analysis

Two samples for each of the purified cell populations were analyzed. The data, comprising ~21,000 transcripts, were then manually processed and presented in a series of Tables where the numbers of reads for the respective genes are presented, and where the different tables focus on different categories of genes. Altogether, the conducted analysis identified more than 780 transcripts that were differentially expressed among the assessed cell populations, and these transcripts were therefore selected for a detailed comparison of the phenotypes of the chosen cell populations. In our analysis, genes were divided according to functional characteristics and by particularly emphasizing genes with a known impact on the respective cell populations.

The entire 15-sample Excel file, including all 20,803 listed genes, is available in Appendix A. This file also includes the 4 samples of cultured human skin MCs analyzed in a previous comparative analysis of similarities and differences in the phenotype between freshly isolated and cultured human skin MCs [3]. The cultured human skin MCs originate from both female breast skin and male foreskin, with two samples each.

## 4. Conclusions

Analyses of the total transcriptome of individual cells or cell fractions have opened new ways to obtain detailed information concerning the phenotype of different cell populations. Here, we analyzed the entire transcriptome of five major human immune cell populations: B cells, monocytes, CD4^+^ T cells, CD8^+^ T cells, and MCs. We identified more than 780 transcripts that were differentially expressed among these cells, making them particularly interesting for analysis of their roles in human immunity. The results were almost school-book-clear and confirmed the findings of many years of detailed work on the phenotype of these five cell populations. However, this study also added a number of new findings and clarified inconsistencies in the literature concerning the expression of many genes and proteins, for which previous results were often derived from analysis in cell lines or ex vivo cultured cells.

In spite of uncertainties such as the half-life of the transcripts, the proteins produced, and also how efficiently the mRNAs are translated, a transcriptomic analysis probably is still the best available technique to obtain the full picture of the expression levels of all 21000 human genes in a cell. It should be noted that regardless of methodology, mRNA or protein levels may differ depending on age, sex, weight, ethnicity, health status, and disease burden. In the present study, we used freshly isolated cells from healthy individuals, and for MCs, we analyzed both male and female cells with very similar results. One additional limitation when trying to translate mRNA to protein levels is that some proteins are short-lived, and some are targeted for lysosomal degradation, which is why mRNA levels do not always reflect the amount of protein present in the tissue. However, other techniques such as antibody-based studies of the abundance of proteins in a tissue are dependent on the number of epitopes the antibodies bind; the type of protein used for developing the antibodies, such as native or denatured; and if the entire protein was used or a part, or only peptides. The fixation method is also of key importance, which is why protein quantifications by antibodies are only rarely quantitative; this is also the situation for mass spectrometry (MS), which is not quantitative but is still often used. So, despite its limitations, quantitative transcriptomics is most likely presently the most reliable method to obtain a good quantitative estimate of the phenotype of a cell.

The findings from this large comparative analysis combined with previous studies of mouse MCs and organs demonstrate that the majority of MC granule proteases are expressed exclusively by MCs and therefore can be used as markers to estimate the fraction of MCs in a whole tissue sample [18]. In mice, we can also obtain a quite accurate estimate of the distribution of their respective subtypes, mucosal and connective tissue MCs, based on their differential expression of granule proteases [18]. The possibility to use of these MC proteases to map subpopulations of MCs in different tissues was recently also confirmed by single-cell analysis in the mouse [375].

Interestingly, our data strongly suggest that human skin MCs are not even a minor producer of any granzymes, which has been debated for many years. The human skin MCs did not express any of the five human granzymes, at least not under the non-inflammatory conditions applied in this study (Table 1). We could also show that there was a correlation between the expression level of proteases and their respective protease inhibitors in a particular cell type.

When we analyzed CD markers that had been identified as good lineage-related surface markers, our results adhered nicely to the published information in most cases. However, we also obtained very interesting quantitative information concerning their expression levels in different human immune cells. For example, in T cells, the 2–3 times higher levels of **CD3E** compared to **CD3G** and **CD3D** makes sense, as they are needed in twice the amount as **CD3E** forms heterodimers with both **CD3G** and **CD3D** (Table 5). The high levels of **CD4** in both monocytes and MCs were also of value for questions concerning the purification of **CD4**^+^ T cells and the biological function of **CD4**. It was also interesting to see the very low expression level of the lineage-specific marker **CD19** on B cells, which still seems sufficient for cell surface expression and for efficient antibody-mediated purification of these cells. The very large difference in expression levels between the two signaling components of the B-cell receptor, the Igα and Igβ (**CD79A** and **B**) was also remarkable as in theory they should be expressed at equal levels. However, we could see that **CD79A** was expressed at 10 times higher levels than **CD79B** (Table 5).

Our analyses of Fc receptors showed that human skin MCs express the high-affinity IgE receptor at an expression level 20–30 times higher than human blood monocytes and that they only express one Fc receptor for IgG, **FCGR2A**. Quite varying results have been presented concerning the expression of Fc receptors for IgG on MCs, which is why this information now clarifies these inconsistencies. We also found that the only **MRGPRX** receptor expressed among the five immune cells was **MRGPRX2** and that **MRGPRX2** was expressed exclusively by the skin MCs and at a relatively high level. This receptor has received a lot of attention due to its role in non-IgE-dependent MC activation and thereby as an alternative mechanism causing allergy exacerbations. The present study identified the expression of the Fc receptor-like molecules **LILRs** and **KIRs** in normal freshly isolated and nonactivated cells, with an almost exclusive expression of **LILRs** in monocytes and **KIRs** in T cells. We also confirmed that both human MCs and T cells express very low levels of **MHC** Class II and are therefore relatively poor antigen-presenting cells compared to both B cells and monocytes. A highly interesting finding was the almost total lack of expression of **MHC** Class I **B** and **C** genes in one of the individuals, indicating that we can live with a quite limited MHC Class I repertoire, at least in a modern society without heavy parasite load and with effective vaccines. Several complement and coagulation components were found to be expressed by human monocytes and, as earlier shown, also by mouse peritoneal macrophages [38].

The analysis of genes related to the complement systems showed that human monocytes express two components of the alternative pathway, properdin (**CFP**) and factor D (**CFD**), and one component of the lectin pathway, ficolin (**FCN1**). We have previously reported that, in addition to this, mouse peritoneal macrophages also express the three genes for **C1Q (C1QA**, **B, and C**), the complement component **C4a**, factor **H,** and factor **B**, as well as the coagulation factors **V**, **VII**, and **X** [38]. Together, these findings indicate that both human monocytes and mouse macrophages are important players in the complement system and that the mouse macrophages are also in the coagulation system.

Interestingly, we found remarkably low levels of the different pattern recognition receptors, including **TLR**, **NOD**, and **RIG** receptors in the five human immune cells, and this was previously shown also in mouse peritoneal MCs and macrophages (15) [38]. However, as mentioned earlier, higher expression levels have been obtained in other studies [39]. In spite of the apparent low levels of expression, these pattern recognition receptors are extremely potent in inducing cytokine expression in various immune cells, as we recently showed for human monocytes [16].

When it comes to enzymes involved in histamine and arachidonic acid synthesis, we found that MCs were the only cells in the present study expressing the histidine decarboxylase, which is essential for the formation of histamine. MCs were the dominating cell type in terms of the expression of arachidonic acid-metabolizing enzymes, which makes sense as MCs are major producers of both leukotriene C4 and prostaglandin D2, **LTC4** and **PGD2**. MCs were also dominant regarding enzymes involved in the synthesis of heparin, chondroitin sulfate, and other major carbohydrate-containing compounds.

When analyzing transcription factors, we found that not only CD4^+^ T cells but also CD8^+^ T cells expressed transcription factors separating the three major Th subtypes, Th1, Th2, and Th17. Both CD4^+^ and CD8^+^ T cells expressed equal levels of the Th2-related transcription factor **GATA3** and the Th1-related **T-bet**, but compared to the CD4^+^ cells, CD8^+^ expressed lower levels of **RORC**, which specifies Th17 cells. This finding indicates that CD8^+^ T cells may have similar subtypes as CD4^+^ cells, which is speculative but interesting. Notably, we identified a panel of additional transcription factors in MCs, which opens for future analysis of their potential role in MC differentiation.

The list of novel and interesting findings can be very long when looking at all the quantitative expression levels emerging from this large comparative analysis. We hope that this detailed map across five major human immune cells can serve as a starting point, in combination with the Human Proteome Atlas (https://www.proteinatlas.org), for continued in-depth analysis of their respective roles in the steady state, during normal human immune response and in disease states.

We have summarized some of the most characteristic features of these five cell types in Figure 2.

We have also made the entire Excel file available for the research community to study the remaining more than 20,000 transcripts not analyzed in detail in this communication. This file is also easily searchable by the gene name and the search tool in Excel without the need for any advanced bioinformatic skills.

## Figures and Tables

**Figure 1 ijms-25-13050-f001:**
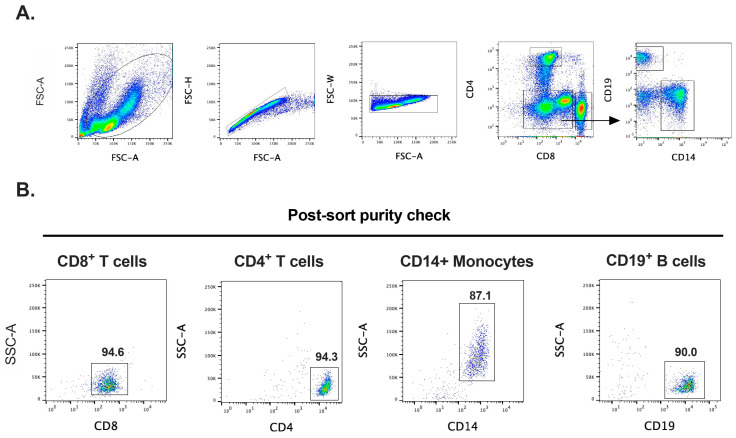
**Isolation of human peripheral blood cell subpopulations by FACS**. Blood PBMCs isolated from buffy coats were stained with antibodies against CD4, CD8, CD14, and CD19 to isolate CD8^+^ T cells, CD4^+^ T cells, CD19^+^ B cells, and CD14^+^ monocytes, respectively: (**A**) Gating strategy for the FACS of the different immune cell populations. (**B**) A fraction of isolated cells was used for the assessment of the purity in percent of the populations. The data are representative of the two different samples.

**Figure 2 ijms-25-13050-f002:**
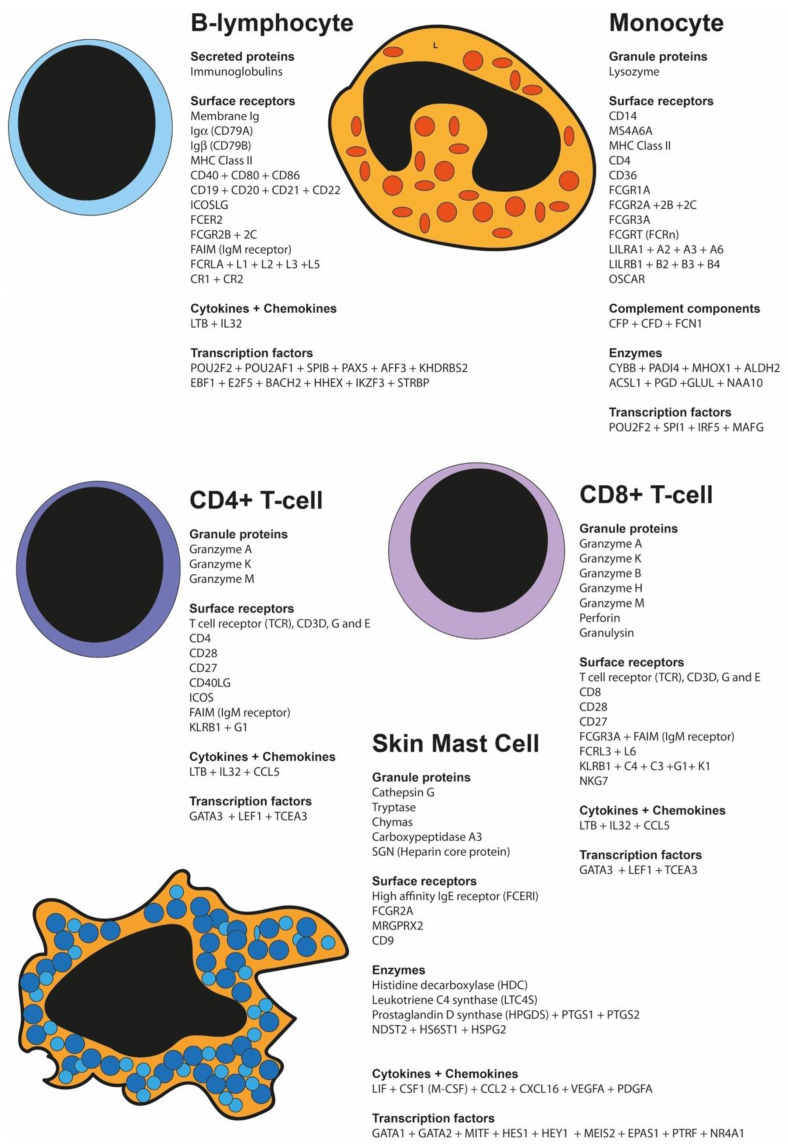
**A summary figure of some of the most important genes for the phenotype of these five cell types.** The genes have been listed in a few major categories including secreted proteins, granule proteins, surface receptors, complement components, cytokines and chemokines, transcription factors, and enzymes.

**Table 1 ijms-25-13050-t001:** Granule proteases.

Gene	B Cells	B Cells	Monocytes	Monocytes	CD4^+^ T Cells	CD4^+^ T Cells	CD8^+^ T Cells	CD8^+^ T Cells	MCs	MCs
** *CTSG* **	0.1	0	0.5	1	0.3	0.1	0.2	2	9445	6194
** *TPSB2* **	0.5	0	0	0.4	0.1	0.4	0	0	7096	6177
** *CPA3* **	0	0	0	0.3	0	0.1	0.2	0	4403	3086
** *CMA1* **	3	0	1	20	3	0	7	1	1713	1359
** *TPSD1* **	0	0	0	0	0.2	0	0.2	0	418	67
** *TPSG1* **	0	0	0	0	0	0	0	0	86	49
** *CPM* **	0.2	0.8	0.6	0.9	0.1	0.2	0	0	140	81
** *GZMA* **	0.1	0	0.4	0.1	68	38	219	268	0	0
** *GZMK* **	0	0	0	0.1	148	51	228	146	0	0
** *GZMH* **	0	0	0	0	9	0.4	280	103	0	0
** *GZMB* **	0.4	0.1	0	0	2	0.6	155	70	0.4	0.2
** *GZMM* **	0	0	0	0.2	65	41	160	279	0	0
** *PRF1 (perforin)* **	0.1	0	0	0	41	15	274	632	0	0.4
** *GNLY (Granulycin)* **	0.7	0.8	6	2	109	50	722	205	1	2
** *PRTN3* **	0	0	0.1	0	0	0	0	0	0	0.1
** *ELANE* **	0	0.2	0.4	4	0	0	0	0	0.4	0.5
** *PRSS57 (NSP4)* **	0	0	0	0	0	0	0	2	0	0
** *CTSC* **	5	6	30	21	22	17	40	13	23	28

**Table 2 ijms-25-13050-t002:** Other proteases.

Gene	B Cells	B Cells	Monocytes	Monocytes	CD4^+^ T Cells	CD4^+^ T Cells	CD8^+^ T Cells	CD8^+^ T Cells	MCs	MCs
** *CTSD* **	60	55	1792	1517	139	175	210	511	2356	1723
** *CTSS* **	147	175	538	1011	122	89	147	22	24	15
** *CTSB* **	16	18	88	28	91	72	32	26	241	315
** *CTSL1* **	0	0.1	4	3	4	1	0.8	0	48	158
** *CTSA* **	47	30	296	352	64	105	104	208	49	49
** *CTSW* **	1	4	0.3	0.3	104	156	627	1153	32	12
** *SCPEP1* **	35	62	206	303	9	12	9	8	10	10
** *ADAM15* **	6	9	207	169	12	12	12	19	23	27
** *ADAM19* **	147	195	14	61	38	38	9	4	4	5
** *ADAM23* **	11	15	0	0	47	49	9	6	1	2
** *ADAM28* **	37	38	2	1	0.2	0	0.1	0.7	0.4	0
** *ADAMTS7* **	0.2	0	0	0	0.1	0	0	0.1	80	58
** *ADAMTS6* **	22	14	0	0	7	4	24	6	1	4
** *PSMB10* **	118	91	796	1314	73	78	82	59	31	21
** *NAPSB* **	113	103	410	459	2	2	0.9	0	0.7	0.1
** *PRSS12* **	0.1	0	0	0.2	0	0	0	0	38	59
** *MMP17* **	21	54	47	79	0	0	0	0	4	4
** *PLAU* **	0.1	0	3	2	0	0	0	0.1	102	38
** *(PLAUR)* **	0.5	0.2	970	1371	8	4	5	8	888	415
** *DPP4* **	1	0.3	0	0	129	111	71	81	32	15
** *CASP3* **	5	5	5	5	7	5	9	18	88	74
** *CAPN10* **	8	10	47	43	10	13	14	9	5	6
** *BACE2* **	10	12	0	0.1	0.1	0	0	0	103	49

**Table 3 ijms-25-13050-t003:** Protease inhibitors.

Gene	B Cells	B Cells	Monocytes	Monocytes	CD4^+^ T Cells	CD4^+^ T Cells	CD8^+^ T Cells	CD8^+^ T Cells	MCs	MCs
** *CST3* **	7	4	4677	6287	21	25	32	18	564	586
** *CST7* **	0.1	0	0	0.3	89	23	290	50	111	91
** *TIMP1* **	7	6	216	129	153	218	105	329	407	357
** *TIMP2* **	28	18	491	488	10	9	7	8	52	48
** *TIMP3* **	0	0	0.3	0	0	0	0	0	1270	995
** *SERPINB1* **	29	31	42	61	68	50	66	16	333	156
** *SERPINH1* **	0.8	1	4	1	5	5	4	4	270	352
** *SERPINE1* **	0	0	2	0.8	0	0.1	0.5	0	200	339
** *LXN* **	0.8	2	0.1	0	2	2	1	1	27	132
** *ITIH4* **	3	2	78	48	3	2	1	1	3	2
** *SPINT1* **	13	11	100	75	4	3	2	1	23	18

**Table 4 ijms-25-13050-t004:** Eosinophil, neutrophil, and macrophage granule proteins.

Gene	B Cells	B Cells	Monocytes	Monocytes	CD4^+^ T Cells	CD4^+^ T Cells	CD8^+^ T Cells	CD8^+^ T Cells	MCs	MCs
** *RNASE3 (ECP)* **	0	0	0.3	0.5	0	0	0	0	0.1	1
** *RNASE2 (EDN)* **	0	0.2	14	19	0	0	0.2	0.4	0.1	2
** *(RNASE6)* **	35	77	40	37	1	2	3	0.2	1	0
** *EPX (EPO)* **	0	0	0	0	0	0	0	0	0	0
** *PRG2 (MBP)* **	0	0.1	0.6	0.6	0	0	0.5	0.2	13	29
** *CLC* **	0	0	0	0	0.1	0.2	0	0.1	0	0
** *MPO* **	0.2	0	13	9	0	0	0	0	0	0
** *LTF* **	0	0.2	0.6	0	0	0	0	0	0.5	0.7
** *NCF1* **	194	198	604	618	0.2	0.2	0.8	0.2	2	0.7
** *NCF2* **	12	15	358	225	7	1	7	1	0.3	0.6
** *LYZ* **	12	6	18,602	23,009	38	23	78	15	29	13
** *MPEG1* **	52	28	36	52	0.6	0.2	1	0.2	1	0
** *COCH* **	63	59	0.6	0	0	0.1	0.5	1	3	5
** *MNDA* **	29	4	152	26	1	0	2	0.9	1	1

**Table 5 ijms-25-13050-t005:** Surface markers.

Gene	B Cells	B Cells	Monocytes	Monocytes	CD4+ T Cells	CD4+ T Cells	CD8+ T Cells	CD8+ T Cells	MCs	MCs
** *CD2* **	0	0	0.1	0.1	128	74	128	26	0.7	1
** *SLAMF6* **	40	41	0	0	18	19	47	21	0.1	0.1
** *CD4* **	0.4	0.1	193	197	536	501	18	1	238	157
** *CD5* **	24	24	0.6	0.3	908	798	564	702	0	0.7
** *CD6* **	10	6	0.4	0	468	371	261	407	0.3	0.1
** *CD7* **	0.2	0.1	29	6	406	490	600	2355	4	11
** *CD8A* **	0.1	0	0	0	10	0.5	781	332	0.1	0.2
** *CD8B* **	0	0	0.4	0.5	9	0.7	875	1479	0.4	0.3
** *CD3D* **	0	0	0	0	358	358	352	216	0	0.2
** *CD3E* **	0.2	0.2	0.3	0.1	1311	1688	1468	2349	0.4	1
** *CD3G* **	0.8	0	0	0	444	588	567	479	0.3	0
** *CD247* ** ** *(CD3Z)* **	0.1	0	0.3	0.6	328	335	371	353	0	0.4
** *CD28* **	0	0	0.1	0	158	198	94	66	0.1	0
** *TESPA1* **	26	26	0.4	0.3	229	351	196	229	528	272
** *BTLA* ** ** *(CD272)* **	69	103	0.3	0.1	9	10	6	1	0.1	0.2
** *CTLA4* **	0.1	0	0	0	23	19	4	2	0	0.3
** *ICOS* **	0	0	0.1	0	51	50	28	3	0	0.5
** *ICOSLG* **	533	372	216	160	13	7	9	12	26	32
** *PDCD1* ** ** *(PD1)* **	0.2	0.2	0.3	0.4	14	3	15	14	0	0.1
** *CD274* ** ** *(PD-L1)* **	2	0.7	0	0.3	4	5	5	3	106	158
** *PDCD1LG2 PD-L2* **	0.7	0	0	0	0.6	0.5	0.5	0.1	8	5
** *CD27* **	45	35	0	0.1	106	153	109	541	0	0
** *CD80 (B7:1)* **	21	14	0.1	0.4	0.8	1	0.8	0	3	9
** *CD86 (B7:2)* **	15	25	28	29	1	0.1	0.6	0	1	0.8
** *CD276* ** ** *(B7-H3)* **	0	0	0.1	0	0	0	0	0	17	32
** *CD40* **	164	120	11	9	2	2	2	5	9	12
** *CD40LG* **	0.1	0.1	0.3	0.3	84	68	19	14	0.4	0.2
** *CD9* **	4	15	6	2	1	2	8	3	931	1078
** *CD14* **	1	0.3	1304	1388	3	2	4	2	22	26
** *MS4A6A* **	2	1	578	378	2	1	5	2	49	18
** *CD33* **	0.7	0	83	38	0.8	0	1	0	42	28
** *CD36* **	0.2	0.2	117	138	0.8	0.7	2	0	3	7
** *LY86* **	97	143	69	62	0.9	2	0.6	0.6	0.7	0.1
** *CD19* **	11	16	0	0	0	0	0.1	0	0	0
** *MS4A1* ** ** *(CD20)* **	214	359	0.1	0	2	0.4	3	0.4	0	0
** *CR2* ** ** *(CD21)* **	40	40	0	0.1	4	5	5	5	0.3	0
** *CD22* ** ** *(Siglec-2)* **	737	655	3	0.4	0.2	0	1	0	146	110
** *CD53* **	18	36	0.3	0.3	0	0	0.2	0.2	0	0
** *CD34* **	0	0	0.1	0	0	0	0	0	4	2
** *CD72* **	42	53	5	3	0.9	2	3	4	4	2
** *CD79A (Igα)* **	3902	4312	10	4	6	6	33	33	0	0.2
** *CD79B (Igβ)* **	314	304	4	2	12	12	6	23	8	5
** *IGLL5* **	207	686	0.3	0	0.6	0	0.5	0	0.3	0
** *VPREB1* **	0.2	0.1	0.3	0	0.2	0	0	0	0	0
** *VPREB3* **	135	162	0.1	0.1	0.1	0.1	0.2	0	0.3	0
** *IGJ* **	206	270	0.4	0.4	1	0.9	0.5	0	0.3	0
** *MZB1* **	66	51	0.3	0	0.1	0.2	0.2	0.5	0.8	0.5
** *CD63* **	33	33	369	658	37	37	56	41	265	306
** *CD68* **	15	20	347	663	6	7	8	3	355	305
** *CD69* **	632	1447	1	1	1044	1129	880	773	2143	341
** *CD81* **	397	507	29	30	284	235	351	854	682	1025
** *CD82* **	137	198	53	26	79	77	32	64	549	596
** *CD83* **	481	221	57	11	9	10	8	48	1555	969
** *CD96* **	3	34	0.1	0	198	190	251	58	0.1	0.4
** *CD97* **	57	32	68	89	162	145	187	71	533	606
** *CD163* **	0.5	0	37	48	0	0.2	0.5	0.1	3	2
** *CD180* **	40	44	8	3	0.1	0	0.1	0	0	0
** *LY86* ** ** *(MD-1)* **	97	143	69	62	0.9	2	0.6	0.6	0.7	0.1
** *CD200* **	44	63	0	0.1	2	4	0.5	0.1	10	13
** *CD248* **	3	2	1	3	4	6	123	244	44	2
** *CD300A* **	1	2	31	22	21	14	59	92	56	35
** *CD300C* **	1	0.3	47	53	0.7	0.2	1	0.1	16	15
** *CD300E* **	0.2	0	344	171	0.2	0.1	2	0.2	0.3	0.1
** *CD300LB* **	0.2	0	23	15	0	0	0.5	0.2	1	0.5
** *CD300LF* **	0.4	0.1	161	119	0.1	0	0.2	0.2	12	5

**Table 6 ijms-25-13050-t006:** MHC and related transcripts.

Gene	B Cells	B Cells	Monocytes	Monocytes	CD4^+^ T Cells	CD4^+^ T Cells	CD8^+^ T Cells	CD8^+^ T Cells	MCs	MCs
** *HLA-DRA* **	6459	5685	1208	994	63	35	71	28	75	65
** *HLA-DPA1* **	6014	5205	2486	2516	116	119	212	200	93	60
** *HLA-DRB1* **	2356	4672	1130	3761	33	70	55	68	146	121
** *HLA-DPB1* **	2055	2907	956	1153	83	80	198	189	39	28
** *HLA-DQA1* **	11	458	4	83	0	3	0.3	4	17	8
** *HLA-DQA2* **	0	0	0	0.1	0	0	0	0	0	0.1
** *HLA-DQB2* **	0	0	0	0	0	0	0	0	0.1	1
** *HLA-DMA* **	762	538	582	542	12	16	20	27	31	21
** *HLA-DOA* **	93	137	8	8	2	2	14	26	2	0.6
** *HLA-DOB* **	157	195	1	0.6	0.4	0.1	0.9	0	0.1	0.1
** *CD74* **	1134	1030	168	156	99	68	98	58	111	63
** *CIITA* **	900	804	408	244	3	2	9	12	7	3
** *CD1A* **	3	7	4	6	0	0	0.2	0	1	0
** *CD1B* **	0	0	0	0.1	0	0	0	0	0.3	0.1
** *CD1C* **	3	5	1	2	0.2	0.4	0	0	0	0
** *CD1D* **	40	15	466	441	0	0	0.8	0	0.3	0
** *B2M* **	4231	5756	1574	2081	11,150	8743	11,103	5838	2244	2218
** *HLA-A* **	1470	521	1754	1087	2498	1218	6862	344	326	1470
** *HLA-B* **	33	2159	45	5368	29	3042	28	3134	297	188
** *HLA-C* **	2	2193	5	514	17	4421	10	9613	552	241
** *HLA-E* **	3951	3493	4625	3166	4026	4662	4306	7743	1533	1614

**Table 7 ijms-25-13050-t007:** Fc receptors.

Gene	B Cells	B Cells	Monocytes	Monocytes	CD4^+^ T Cells	CD4^+^ T Cells	CD8^+^ T Cells	CD8^+^ T Cells	MCs	MCs
** *FCER1A* **	0.1	0	8	5	0.6	0.9	0.5	0.1	214	127
** *MS4A2* **	0	0	0	0.1	0	0	0.1	0	817	441
** *FCER1G* **	0.6	0.7	741	816	17	7	34	57	910	848
** *FCER2* **	598	643	6	4	0.6	0.1	0.3	0.1	0	0
** *FCGR1A* **	0.2	0.2	13	15	0.1	0	0.3	0	0	0
** *FCGR2A* **	1	0.6	66	61	0.3	0	0.8	1	142	199
** *FCGR2B* **	31	34	38	188	0.3	0	1	1	2	0.4
** *FCGR2C* **	30	32	38	190	0.3	0.1	2	2	2	1
** *FCGR3A* **	0	0.3	6	8	1	1	58	65	0.3	0
** *FCGR3B* **	0	0	0	0	0	0	0	0.1	0	0
** *FCGRT* **	9	16	135	235	19	21	17	4	28	19
** *FAIM3* **	468	422	2	0	297	362	282	199	0.1	0.1
** *FCRLA* **	114	184	0.1	0	0	0	0.5	0	0	0
** *FCRLB* **	1	2	0.3	0.1	0.3	0.1	0.8	1	0.8	0
** *FCRL1* **	443	518	0.4	0	2	3	2	1	0	0
** *FCRL2* **	25	23	0	0	0.1	0	0.2	0	0	0
** *FCRL3* **	66	47	0	0	4	5	24	16	0	0
** *FCRL4* **	0	0	0	0	0	0	0	0	0	0
** *FCRL5* **	160	87	1	0.1	0	0	0.5	0	0	0
** *FCRL6* **	0.1	0	0.4	0	0.8	0.4	126	72	0.1	0.5
** *MILR1* **	33	27	42	37	0.3	0.2	0.7	0.2	57	29
** *CD200R1* **	0	0.2	0	0	2	1	3	0	24	6
** *MRGPRX1* **	0	0.1	0	0	0	0	0	0	0	0.1
** *MRGPRX2* **	0	0	0	0	0	0	0.1	0	365	253
** *MRGPRX3* **	0.2	0	0.5	0.6	0	0	0.1	0.1	0	0.1
** *MRGPRX4* **	0.7	0	0	0	0	0	0	0	0	0

**Table 8 ijms-25-13050-t008:** Leukocyte immunoglobulin-like and killer cell lectin-like receptors.

Gene	B Cells	B Cells	Monocytes	Monocytes	CD4^+^ T Cells	CD4^+^ T Cells	CD8^+^ T Cells	CD8^+^ T Cells	MCs	MCs
** *LILRA1* **	0.4	0.1	42	53	0.4	0	0.2	0.2	2	0.3
** *LILRA2* **	0.4	0.7	96	106	0.1	0.4	2	0	17	4
** *LILRA3* **	0.2	0	78	34	0.2	0	1	0.4	0.1	0
** *LILRA6* **	0	0.3	141	128	2	1	2	2	2	4
** *LILRB1* **	34	37	47	48	0.2	0	10	1	0.1	0
** *LILRB2* **	0.1	0.4	245	240	0.8	0.5	1	0.6	4	1
** *LILRB3* **	2	0.7	335	500	2	2	2	0.2	8	2
** *LILRB4* **	0.4	0.4	107	147	0.1	0.5	0.3	0	1	0.3
** *OSCAR* **	0	0	327	284	0	0	0.5	0.1	0	0.7
** *KLRB1* **	0.2	0.3	0.5	0	504	245	420	183	0.4	1
** *KLRC4* **	0	0	0	0	0.4	0	71	23	0	0.2
** *KLRC3* **	0	0	0.1	0	0.4	0	82	58	0	0.1
** *KLRG1* **	3	4	0.4	0.9	101	97	197	178	202	96
** *KLRK1* **	34	19	0.3	0	5	0.4	641	424	1	0.8
** *NKG7* **	1	2	13	13	84	32	1423	1129	0.4	0.5
** *NCR3* **	39	56	0.3	0	18	19	46	72	0.7	0

**Table 9 ijms-25-13050-t009:** Complement and coagulation components and receptors.

Gene	B Cells	B Cells	Monocytes	Monocytes	CD4^+^ T Cells	CD4^+^ T Cells	CD8^+^ T Cells	CD8^+^ T Cells	MCs	MCs
** *C1QA* **	0	0	20	15	0	0	0.2	0	2	0.9
** *C1QB* **	0.2	0	2	0.4	0	0	0	0.1	5	2
** *C1QC* **	0	0	2	0.1	0	0	0.1	0	10	2
** *C2* **	0	0.2	5	8	0	0	0	0.1	13	12
** *CFP* **	3	1	732	837	26	37	11	17	1	1
** *CFD* **	3	0.7	478	431	3	1	4	7	107	93
** *FCN1* **	3	0.6	1166	766	5	2	10	4	0.7	0
** *FCN2* **	0	0	0	0	0	0	0	0.1	0	0
** *FCN3* **	0	0	0	0	0	0	0	0	0	0.2
** *C3* **	0	0	0.1	2	0	0	0.1	0	8	7
** *C9* **	0	0	0	0	0	0	0	0	0	0.1
** *CFH* **	0	0	0	0	7	9	8	4	14	1
** *C3AR1* **	0	0	0.9	1	4	1	3	0.6	31	6
** *PROCR* **	0	0	0.6	2	0.2	0.1	1	0.5	36	19
** *CR1* **	24	23	1	12	3	4	6	2	0.1	0
** *CR2* **	40	40	0	0.1	4	5	5	5	0.3	0

**Table 10 ijms-25-13050-t010:** Pattern recognition receptors.

Gene	B Cells	B Cells	Monocytes	Monocytes	CD4^+^ T Cells	CD4^+^ T Cells	CD8^+^ T Cells	CD8^+^ T Cells	MCs	MCs
** *TLR1* **	12	10	3	7	2	3	2	0.3	0.3	0.1
** *TLR2* **	0.7	0.4	7	4	0.8	1	0.2	0.6	2	1
** *TLR3* **	0	0	0	0	1	0	2	0.1	1	0.5
** *TLR4* **	0	1	6	6	0	0	0	0	14	8
** *TLR5* **	0	0	3	4	6	5	3	0.1	1	0.4
** *TLR6* **	11	8	1	1	0.2	0.4	0	0.1	0	0
** *TLR7* **	7	13	2	1	0	0.1	0	0	0.4	0.1
** *TLR8* **	0.1	0	32	11	0.3	0	0.6	0	0	0
** *TLR9* **	12	11	5	3	0.3	0.4	0.1	0	0.7	0.7
** *TLR10* **	2	5	0	0	0	0	0	0	0	0
** *NOD1* **	18	23	5	6	8	8	8	6	11	5
** *NOD2* **	0.1	0.2	29	43	3	3	1	0.9	0.1	0.5
** *NLRC3* **	19	23	0	0	39	42	45	23	4	2
** *DDX58 (RIG1)* **	16	17	5	5	10	14	15	13	39	33
** *IFIH1 (MDA5)* **	17	18	1	2	9	10	12	2	17	13
** *FADD* **	9	8	8	6	26	25	23	23	8	16
** *EIF2AK2 (PKR)* **	95	95	29	27	76	121	86	100	44	44
** *MAVS* **	35	30	21	11	22	32	26	22	26	22
** *MYD88* **	30	26	100	99	27	25	39	36	49	43
** *PLD4* **	83	178	23	21	1	2	0.3	0.1	2	0
** *CLEC7A* **	0.2	0.2	41	30	0.2	0.2	0.2	0.1	0	0
** *FPR1 (Formyl pep. R)* **	0.1	0.2	48	23	0	0	0.4	0	0	0.3

**Table 11 ijms-25-13050-t011:** Histamine, leukotriene, and prostaglandin synthesis.

Gene	B Cells	B Cells	Monocytes	Monocyte	sCD4^+^ T Cells	CD4^+^ T Cells	CD8^+^ T Cells	CD8^+^ T Cells	MCs	MCs
** *HDC* **	0	0	0	0	0.2	0	0	0	853	796
** *MAOA* **	0	0	0.3	0	0	0.5	0.2	0	9	10
** *MAOB* **	0	0	0	0	0	0.6	0.1	0	200	105
** *ALOX5* **	311	249	235	289	6	5	3	0.4	307	1423
** *COTL1 (CLP)* **	439	596	1032	780	541	515	516	536	259	297
** *ALOX5AP (FLAP)* **	72	82	35	47	175	134	183	281	312	130
** *LTC4S* **	0.1	0.2	2	2	3	4	3	2	129	60
** *HPGD* **	0.2	0.2	0.1	0.1	8	9	8	10	1296	1021
** *HPGDS* **	0	0	0.1	0	0	0.2	0	0	705	484
** *PTGS1* **	40	30	50	40	0	0.5	0.5	0.5	378	357
** *PTGS2* **	0	0.1	4	2	0.1	0	0	0	523	392
** *PLA2G2A* **	0	0	0	0	0	0	0	0	272	579
** *TBXAS1* **	5	9	297	323	13	7	3	1	26	18
** *LPCAT2* **	0	0.2	27	39	0	0.9	0.3	0	202	95
** *ENPP2* **	1	2	0	0.1	5	18	4	12	107	86
** *MBOAT7* **	28	30	332	263	20	35	20	51	291	324
** *ANXA1* **	0.8	0.8	439	366	585	361	547	204	3919	5423

**Table 12 ijms-25-13050-t012:** Proteoglycan synthesis and binding and other carbohydrate-containing or interacting proteins.

Gene	B Cells	B Cells	Monocytes	Monocytes	CD4^+^ T Cells	CD4^+^ T Cells	CD8^+^ T Cells	CD8^+^ T Cells	MCs	MCs
** *SRGN* **	78	45	1125	1350	1117	690	1258	897	5850	7939
** *HS3ST1* **	2	5	0	0	0	0.1	0	0	46	19
** *HS6ST1* **	13	20	64	29	9	5	6	3	293	108
** *HSPG2* **	1	0.2	0.4	0.4	0.7	3	0.7	0.2	198	127
** *GALNT6* **	3	5	13	17	4	2	5	3	59	68
** *NDST1* **	5	6	19	22	0.4	1	0.8	0.5	65	54
** *NDST2* **	113	99	148	103	84	109	79	91	799	183
** *GNS* **	10	9	29	23	24	19	32	16	135	270
** *B4GALT5* **	8	8	12	5	8	9	13	11	129	255
** *GBE1* **	11	12	1	1	6	8	8	9	139	151
** *SDC3* **	1	1	4	4	0.7	0.2	0.2	0	87	74
** *B3GNT5* **	0	0.1	14	12	0	0.1	0	0.9	79	100
** *EXTL3* **	1	3	2	5	4	4	5	2	65	25
** *EXT1* **	5	7	10	18	8	8	10	6	119	91
** *GLT1D1* **	0.2	0	75	103	0	0.2	0.3	0	0	0.1
** *RENBP* **	6	4	90	88	5	4	4	2	99	63
** *VCAN* **	0.4	0	140	774	0.9	0.4	2	0.4	3	2
** *LGALS2* **	0.5	0.2	88	136	0.2	0.7	1	0.2	1	0.4
** *LGALS3* **	1	1	96	59	53	24	24	10	679	885

**Table 13 ijms-25-13050-t013:** Other enzymes.

Gene	B Cells	B Cells	Monocytes	Monocytes	CD4^+^ T Cells	CD4^+^ T Cells	CD8^+^ T Cells	CD8^+^ T Cells	MCs	MCs
** *CYBB* **	134	190	725	1465	2	2	7	0.4	4	0.9
** *ARG2* **	0	0.7	0.9	0	0.1	0.4	0.4	5	68	47
** *PADI2* **	0.2	0.2	63	64	0.1	0.2	0.4	0	59	47
** *PADI4* **	0	0.1	100	98	0.4	0.1	0	0	0	0
** *EPHX2* **	1	1	0	0	32	50	25	13	1	2
** *HMOX1* **	4	3	414	466	1	0.5	2	1	29	13
** *ALAS1* **	6	5	18	17	11	9	11	12	155	90
** *SAMHD1* **	7	7	91	48	154	155	149	131	36	49
** *NUDT16* **	7	7	104	152	5	6	3	9	16	12
** *ALDH2* **	11	17	905	1627	1	0.5	2	3	65	63
** *ACSL1* **	16	13	259	683	7	7	7	2	40	40
** *NUDT18* **	17	24	158	133	5	8	7	19	2	2
** *PGD* **	45	39	1630	3408	33	53	49	83	91	139
** *GLUL* **	26	22	462	536	40	32	22	6	2816	5447
** *NAA10* **	21	19	167	251	21	28	20	10	30	33

**Table 14 ijms-25-13050-t014:** Transcription factors.

Gene	B Cells	B Cells	Monocytes	Monocytes	CD4^+^ T Cells	CD4^+^ T Cells	CD8^+^ T Cells	CD8^+^ T Cells	MCs	MCs
** *GATA1* **	0	0.1	0	0	0	0	0.1	0.1	105	108
** *GATA2* **	0.1	0	0.6	0.3	0.1	0.2	0.1	0	2859	1421
** *GATA3* **	0.1	0.4	0.4	0	196	174	150	281	0.4	2
** *FOXP3* **	0	0.3	0	0.1	30	23	2	0.2	0.3	0.1
** *TBX21 (T-bet)* **	5	3	0	0	27	15	78	21	0.1	0.6
** *RORC* **	0	0	0	0.5	17	10	6	5	0	0.2
** *POU2F1 (Oct1)* **	9	12	3	2	8	9	8	2	15	4
** *POU2F2 (Oct2)* **	292	249	328	240	25	33	27	83	0.5	1
** *POU2AF1* **	307	435	2	0.1	2	4	1	0.1	0.1	0
** *MITF* **	0	0	2	4	0	0	0	0	195	64
** *HES1* **	8	6	1	0.8	0	0	0	0.4	138	117
** *HEY1* **	0	0.1	0	0	0	0.1	0	0.1	86	206
** *MEIS2* **	0.1	0.2	0	0.1	0	0	0.2	0	275	193
** *EPAS1* **	0.1	0.2	0.6	0.1	3	1	5	2	436	835
** *PTRF* **	0	0.2	0.8	0.6	0.4	2	0.3	0.4	465	707
** *NR4A1* **	11	9	8	24	4	3	2	7	997	147
** *NFE2L3* **	1	0.8	2	0.9	4	3	1	1	18	19
** *PBX1* **	0	0	0.5	0.1	0.9	0.1	0.5	0.1	53	44
** *GLI3* **	0	0	0	0	0.3	0.6	1	1	57	47
** *AFF2* **	5	5	4	2	2	9	4	2	95	48
** *SPI1 (PU.1)* **	63	66	369	252	1	1	3	0.9	56	17
** *SPIB* **	48	73	0.3	0.3	0.1	0.2	0	0	0.5	0
** *PAX5* **	590	707	1	0.1	0.8	0	1	0.1	0	0
** *AFF3* **	148	125	2	0.6	4	7	4	1	1	2
** *KHDRBS2* **	212	137	2	0.4	0	0	0.1	0	0	0
** *EBF1* **	53	68	0	0.5	0.2	0	0.2	0	16	0.1
** *E2F5* **	51	67	1	0	1	2	4	1	2	0.8
** *BACH2* **	351	392	4	0.8	73	61	76	12	3	4
** *SETBP1* **	114	91	3	1	0.3	0.9	9	4	8	11
** *LEF1* **	4	4	0	0	365	501	399	632	2	1
** *RNF157* **	0.4	0.1	0	0	50	56	40	48	3	3
** *TCEA3* **	0	0.1	4	1	99	168	70	140	21	29
** *BHLHE40* **	9	9	165	190	226	145	167	66	1449	969
** *HHEX* **	170	173	50	43	1	1	1	1	33	27
** *IKZF1 (Ikaros)* **	50	79	20	19	129	112	133	56	158	133
** *IKZF2 (Helios)* **	1	3	0	0	3	5	16	12	4	1
** *IKZF3 (Aiolos)* **	112	183	0.3	0.1	36	42	85	20	5	5
** *GFI1* **	0.8	0.7	4	4	30	21	39	11	116	41
** *ZEB1* **	76	60	0	0	92	79	78	28	18	16
** *ZEB2* **	69	49	469	236	6	2	75	17	316	346
** *KLF1* **	0	0	0	0	0	0	0	0	0.1	0
** *KLF4* **	84	42	328	207	3	6	1	3	1751	1041
** *ZNF385A* **	13	16	311	269	1	1	3	1	12	16
** *ZNF467* **	0.8	0.7	131	120	4	6	3	4	4	6
** *ZNF513* **	27	26	271	226	18	25	15	18	17	24
** *ZNF521* **	0.1	0.2	0	0	0	0	0.1	0	43	19
** *ZNF703* **	9	7	74	83	1	1	1	3	20	23
** *ZNF787* **	1	0.7	12	32	0.9	0.9	0.5	0.5	2	0.8
** *ZNF827* **	48	22	0.5	0	10	17	29	33	4	8
** *ZNF831* **	12	13	0.1	0	56	50	57	34	0.1	0
** *ZCCHC24* **	2	2	15	8	0.9	1	1	5	53	65
** *ZMIZ1* **	5	4	2280	1103	38	49	28	28	190	300
** *CXXC1* **	8	6	76	85	10	15	7	9	6	5
** *IRF5* **	66	60	444	500	5	5	2	6	16	11
** *CEBPA* **	0.5	0.3	96	54	0.2	0.5	3	6	2	4
** *MAFG* **	9	6	411	390	4	5	5	0.9	38	69
** *ELF3* **	1	0.5	51	46	0.3	1	0.7	0.1	0	0.2
** *VENTX* **	1	1	76	46	0	0	0.1	0	3	2
** *RUNX2* **	6	4	10	9	41	44	33	53	61	23
** *MAF* **	0.1	0	0.4	0.5	42	30	30	22	23	59
** *ETS2* **	4	3	206	177	14	15	14	17	395	9645
** *MYC* **	209	216	2	1	115	135	79	82	149	259
** *STRBP* **	270	258	2	2	26	29	25	10	4	9
** *EIF2AK3* **	246	178	4	2	22	30	25	37	92	254
** *CTDSP1* **	218	241	1508	1675	170	209	184	284	254	221
** *CCDC9* **	170	75	1118	585	33	38	38	49	66	70
** *SCML4* **	14	16	0	0	26	44	35	18	0.1	0

**Table 15 ijms-25-13050-t015:** SOX members.

Gene	B Cells	B Cells	Monocytes	Monocytes	CD4^+^ T Cells	CD4^+^ T Cells	CD8^+^ T Cells	CD8^+^ T Cells	MCs	MCs
** *SOX1* **	0	0	0	0.2	0.1	0	0.2	0.1	0.7	0.4
** *SOX2* **	0	0	0	0.2	0	0	0	0.1	0.5	1
** *SOX3* **	0	0	0	0	0	0	0.2	0	0	0.2
** *SOX4* **	2	6	3	4	0.6	7	2	3	62	29
** *SOX5* **	0	0	0	0.1	0	0	0	0	0.8	0.3
** *SOX6* **	0	0	0	0	0	0	0	0.1	0.4	0.4
** *SOX7* **	3	1	0	0	0	0	0	0	46	2
** *SOX8* **	0	0	0.1	0	4	7	2	11	4	5
** *SOX9* **	0	0	0	0	0	0	0	0	1	1
** *SOX10* **	0	0	0	0.1	0	0	0	0	17	2
** *SOX11* **	0	0	0	0	0	0	0	0.1	0.3	0.1
** *SOX12* **	3	2	2	7	6	3	5	1	7	1
** *SOX13* **	0.1	0.1	0.3	0.1	5	2	10	11	105	135
** *SOX14* **	0	0	0	0	0	0	0	0	0	0
** *SOX15* **	0.2	0.1	2	2	0.1	0	0	0	4	4
** *SOX17* **	0	0	0	0	0	0	0	0	37	60
** *SOX18* **	0	0	0	0	0	0	0	0	33	38

**Table 16 ijms-25-13050-t016:** STATs.

Gene	B Cells	B Cells	Monocytes	Monocytes	CD4^+^ T Cells	CD4^+^ T Cells	CD8^+^ T Cells	CD8^+^ T Cells	MCs	MCs
** *STAT1* **	24	32	53	168	156	150	126	25	46	25
** *STAT2* **	104	97	109	152	49	75	50	34	60	48
** *STAT3* **	46	42	46	85	121	101	117	36	222	199
** *STAT4* **	3	3	0.8	2	34	77	60	53	0.5	0.2
** *STAT5A* **	48	35	111	80	84	63	88	77	80	56
** *STAT5B* **	108	108	39	28	225	230	204	240	111	100
** *STAT6* **	557	427	646	561	160	211	167	274	158	116

**Table 17 ijms-25-13050-t017:** Cytokines and chemokines.

Gene	B Cells	B Cells	Monocytes	Monocytes	CD4^+^ T Cells	CD4^+^ T Cells	CD8^+^ T Cells	CD8^+^ T Cells	MCs	MCs
** *LTB* **	1087	1240	17	20	1403	1944	784	2219	3	2
** *LIF* **	0	0	0	0	0	0.2	0	0.4	659	307
** *TNF* **	8	18	25	36	58	136	63	371	138	124
** *TNFSF10* **	3	4	55	46	28	33	12	14	89	93
** *CSF1* **	0	0.4	3	2	8	8	4	1	1579	933
** *CSF2* **	0	0	0	0	1	0	0.5	0.2	9	61
** *CSF3* **	0	0.2	0	0.1	0.1	0	0	0	65	3
** *EPO* **	0	0	0	0.1	0	0	0	0	0	0.4
** *IL1A* **	0	0	0	0	0	0	0	0	4	5
** *IL1B* **	0	0.1	19	1	0	0	0	0.1	20	10
** *IL2* **	0	0	0	0	0.7	0.9	1	0	0	0
** *IL3* **	0	0	0	0	0	0.5	0.4	0.6	0.3	0.4
** *IL4* **	0	0	0	0	0	0	0	0	0	0.1
** *IL5* **	0	0	0	0	0	0	0.4	0	0.1	0
** *IL6* **	28	25	0.4	0.1	0	0	0	0	262	7
** *IL7* **	6	13	0.1	0.1	0.6	0.4	0.2	0.1	0.8	2
** *IL8* **	0	0.2	32	38	0.2	0	0.4	0.2	65	80
** *IL9* **	0	0	0	0	0	0	0	0	0	0
** *IL10* **	0.7	0.2	0.6	0.8	0.1	0.4	0.4	0	1	0.5
** *IL13* **	0.5	0	0	0.3	0.4	0.2	0.5	0	29	47
** *IL11* **	0.2	0.2	0	0	0.2	0.1	0.2	0.1	0.8	2
** *IL19* **	0.1	0	0	0	0	0	0.1	0	0	0
** *IL12A* **	24	18	1	2	0.4	0.4	2	4	0.5	0.8
** *IL12B* **	0	0	0	0	0	0	0	0	0	0
** *IL15* **	13	9	8	12	6	7	4	3	0.7	0.9
** *IL16* **	0	0	0	0	0	0	0	0.2	3	4
** *IL17A* **	0	0	0	0	0.2	0	0	0	0	0
** *IL17B* **	0	0	0	0	0	0	0	0	2	3
** *IL17F* **	0	0	0,.1	0	0	0	0	0.1	0.1	0
** *IL20* **	0	0	0	0	0	0	0	0	0	0
** *IL21* **	0	0	0	0	0	0	0	0	0	0
** *IL22* **	0	0	0.4	0.6	0	0	0	0	0	0
** *IL23A* **	1	0.5	0	0	8	6	4	1	0.7	0.2
** *IL24* **	0.5	3	0	0	8	7	8	0.7	0.3	0.4
** *IL25* **	0	0	0	0	0	0	0	0	0	0
** *IL26* **	0.7	0.8	0	0.1	0.4	0	0.8	0	0	0
** *IL28A* **	0	0	0	0	0.3	0	0.6	0	0.4	0.1
** *IL28B* **	0.2	0	0.9	1	0.1	0	0	0	0	0
** *IL29* **	0	0	0	0.1	0	0.1	0	0	0	0
** *IL31* **	0	0	0	0	0	0	0	0	0	0
** *IL32* **	0.4	0.3	2	0.3	2636	1843	2728	3462	23	70
** *IL33* **	0	0	0	0	0	0	0	0	4	2
** *IL34* **	0	0	0	0	0	0	0.3	0	2	0.8
** *IL36A* **	0	0	0	0	0.4	2	0.6	1	0	0
** *IL36B* **	0	0	0	0	0	0	0	0	0	0
** *IL36G* **	0	0	0	0	0	0	0	0	0	0
** *IL37* **	0	0	0.1	0	0	0.4	0	0	0	0
** *CCL2* **	0	0	0.6	0.1	0	0.1	0.1	0	1518	1644
** *CCL3* **	0.8	1	24	16	0.2	0.2	7	3	6	4
** *CCL4* **	0.2	1	3	0.8	3	1	81	69	76	34
** *CCL5* **	3	1	7	5	283	142	1778	1157	0.3	0.7
** *CXCL16* **	33	24	50	54	26	12	22	25	263	160
** *PPBP* ** ** *(CXCL7)* **	6	15	154	97	3	8	6	9	0	0
** *VEGFA* **	0.7	0.1	222	109	0.9	0.1	0.6	0.5	1948	1323
** *VEGFB* **	116	117	12	7	53	59	49	96	138	87
** *VEGFC* **	0	0	0	0	0	0	0	0	1	0
** *TYMP* **	12	9	574	637	8	7	6	12	23	37
** *PDGFA* **	0.7	0.1	3	0.4	0.9	0.4	3	0.6	270	366
** *PDGFB* **	0	0	0	0	0	0.5	0.2	1	0	1
** *PDGFC* **	0.1	0	4	3	0	0	0	0	17	20
** *GDF15* **	0	0.2	5	3	0	0	0	0	136	27
** *TGFA* **	0	0	1	0.6	1	0.2	2	5	31	21
** *TGFB1I1* **	0.1	0.3	0.4	0.2	0.3	0	0.5	0.4	137	149
** *ADM* **	0.1	0.2	107	135	0.7	2	1	6	55	113
** *PF4 (CXCL4)* **	4	7	122	64	1	4	4	4	0	0
** *EMR2* **	0.5	0.3	394	472	1	0.6	1	0.5	154	207
** *FGFBP2* **	0.2	0.1	0	0.1	3	0.9	118	66	0.5	0.5

**Table 18 ijms-25-13050-t018:** Cytokine-induced proteins.

Gene	B Cells	B Cells	Monocytes	Monocytes	CD4^+^ T Cells	CD4^+^ T Cells	CD8^+^ T Cells	CD8^+^ T Cells	MCs	MCs
** *TNFAIP2* **	0.1	0.1	697	483	0.1	0.2	1	0.6	44	58
** *TNFAIP3* **	21	35	43	223	1074	710	437	189	5148	1318
** *TGFBI* **	3	3	380	244	8	5	6	2	92	7
** *IFITM1* **	15	26	71	98	515	475	567	713	87	90

**Table 19 ijms-25-13050-t019:** Cytokine and chemokine receptors.

Gene	B Cells	B Cells	Monocytes	Monocytes	CD4^+^ T Cells	CD4^+^ T Cells	CD8^+^ T Cells	CD8^+^ T Cells	MCs	MCs
** *KIT* **	0	0	0	0.1	2	2	0.7	1	1023	458
** *EPOR* **	2	4	0.8	0.8	0.4	1	0.5	1	105	38
** *IL1RL1* ** ** *(IL33R)* **	0	0	0	0	0	0	0	0	125	289
** *IL1R1* **	0	0	0.1	0.3	1	0.7	0.5	0.2	10	6
** *IL1R2* **	0	0	4	6	0.2	0.1	0	0	6	2
** *IL2RA* **	20	13	0	0	24	18	4	2	17	32
** *IL2RB* **	2	0.7	1	1	198	148	316	201	0.4	2
** *IL2RG* **	71	128	10	22	430	357	391	92	28	29
** *IL3RA* **	162?	3	14	4	10	0	2	0	10	11
** *IL4R* **	245	405	151	249	1189	909	829	339	169	291
** *IL5RA* **	0	0	0	0	0	0	0	0	17	11
** *IL6RA* **	0.8	2	71	89	65	66	27	6	69	98
** *IL6ST* **	13	8	3	3	76	134	52	58	124	252
** *IL7R* **	0.1	0.1	0	0	762	605	614	390	8	11
** *IL9R* **	0	0	0	0	2	1	1	0.4	4	11
** *IL10RA* **	249	171	334	264	323	218	324	167	4	2
** *IL10RB* **	11	15	5	12	5	10	7	7	11	11
** *IL11RA* **	2	2	4	4	14	9	9	0.9	13	11
** *IL12RB1* **	3	1	7	14	10	7	7	3	0	0
** *IL12RB2* **	0	0	0.1	0.3	3	2	3	1	0	0
** *IL13RA1* **	3	3	5	5	0.5	0	0.4	0	3	3
** *IL13RA2* **	0	0	0	0	0	0	0	0	0	0.1
** *IL15R* **	3	3	45	35	8	7	6	4	5	8
** *IL16R* ** ** *(CD4)* **	0.4	0.1	193	197	536	501	18	1	238	157
** *IL17RA* **	21	17	160	96	59	63	60	51	35	37
** *IL17RB* **	0.7	0	0	0	0	0.6	0.5	0	0.4	0.3
** *IL17RC* **	0	0	8	6	0.2	0	0	0.2	5	5
** *IL17RD* **	0	0	0	0	0	0	0	0	3	6
** *IL17RE* **	0	0	0	0	2	2	2	1	0.1	0.2
** *IL18R1* **	1	1	0.5	0.5	10	12	6	17	280	216
** *IL20RA* **	0	0	0	0	0	0	0	0	0	0.1
** *IL20RB* **	0.1	0.1	0.3	0.3	0	0.4	0.6	0.7	2	0.7
** *IL21R* **	7	18	1	1	30	16	30	9	0	0.2
** *IL22RA1* **	0	0	0	0	0	0	0	0.1	0.3	0
** *IL22RA2* **	0	0	0	0	0	0	0	0	0	0
** *IL23R* **	0	0	0	0	0.7	1	0.6	2	0	0.2
** *IL27RA* **	27	29	74	70	42	54	35	47	9	14
** *IL28RA* **	7	7	0	0	0.8	0.1	1	2	0	0
** *IL31RA* **	0	0	0.3	0.1	0.1	0	0	0.1	0	0.2
** *CSF1R* **	0.7	0.2	226	210	2	1	1	0.1	2	1
** *CSF2RA* **	0.2	0.1	83	64	0.3	0.6	0.2	0.7	14	6
** *CSF2RB* **	69	50	440	389	2	4	0.7	0.1	512	347
** *CSF3R* **	2	1	2682	2117	2	0.4	2	3	2	0.3
** *TNFRSF13B* **	93	55	0	0	0	0	0.3	0.1	0	0
** *TNFRSF13C* **	478	308	4	0.7	7	3	8	6	0	0
** *TNFRSF9* **	0.1	0	0.3	2	2	1	2	2	159	351
** *TNFRSF21* **	1	3	0	0.5	0	0.4	0.1	0.1	292	226
** *LTBR* **	0.2	0.1	198	146	1	0.2	1	1	74	65
** *TGFBR3* **	0.2	0	0.4	0	88	50	182	129	22	29
** *ACVR1B* **	4	3	21	13	7	5	6	4	28	62
** *CXCR4* **	662	876	9	4	901	483	1185	176	43	10
** *CXCR5* **	136	126	0	0.1	15	16	2	0.2	0.3	0
** *CCR2* **	0.6	0.9	4	2	4	3	1	2	0	0
** *CCR5* **	0	0	0	0	3	0.6	3	1	0.3	0.3
** *CCR6* **	0.5	1	0	0	42	18	12	3	0	0.1
** *CCR7* **	83	52	0	0	124	129	145	286	6	8
** *CRLF2* **	0	0	0.4	0.5	0.8	1	1	0	194	427
** *NFAM1* **	0.1	0.2	700	705	0.3	0.2	0.5	0.6	14	5
** *PTAFR* **	0.2	0.3	6	10	0.1	0	0.2	0	53	25

**Table 20 ijms-25-13050-t020:** Other receptors.

Gene	B Cells	B Cells	Monocytes	Monocytes	CD4^+^ T Cells	CD4^+^ T Cells	CD8^+^ T Cells	CD8^+^ T Cells	MCs	MCs
** *P2RX5* ** ** *(ATP)* **	1081	658	11	8	13	15	15	18	0.8	0.2
** *ADIPOR2* **	3	7	3	4	4	4	8	9	48	101
** *ESRRA* **	16	11	450	224	9	6	12	18	9	12
** *ADRB2* **	16	16	23	24	36	27	73	205	561	590
** *CNR2* **	39	60	0.1	0	0	0	3	2	0	0
** *CNRIP1* **	0	0	0	0	0	0	0	0	113	74
** *RXRA* **	14	7	1420	1336	14	11	13	11	181	109
** *NRP2* **	2	0.6	2	3	0.1	0	0.1	0	87	132
** *LRPAP1* **	85	58	813	1307	95	94	91	48	56	59
** *GPR18* **	79	87	0.1	0.3	36	23	48	32	0	0
** *GPR56* **	0	0	0.6	0.1	3	0.9	189	162	32	2
** *GPR141* **	0	0	5	3	0	0	10	2	80	79
** *GPR174* **	76	111	0.3	0	148	151	156	105	0.4	0.1
** *NPTXR* **	0.2	0.1	0	0	14	12	12	45	6	9
** *MAS1L* **	0	0	0	0	0	0	0.2	0	119	26
** *LRP4* **	0	0	0.1	0	0	0	0	0	148	131
** *HRH2* **	0	0	5	4	0.1	0	0	0.4	0.1	0
** *HRH4* **	0	0	0	0	0.6	0.4	0	0	27	7
** *EDNRB* **	0	0	0	0.1	0	0	0	0	101	63
** *DRD2* **	0	0	0	0	0	0	0	0	35	19
** *DRD5* **	0	0	33	18	0.1	0	0.1	0	0	0
** *AMHR2* **	0	0	0	0	0.1	0	0	0	59	26
** *ADORA3* **	0	0	0.5	0.3	0	0	0	0	16	3
** *STAB1* **	0	0.1	159	192	0.2	0	0.3	0.2	22	14
** *GPBAR1* **	0.1	0	16	19	0.1	0.1	0.3	0.1	0	0
** *PTGIR* **	1	0.5	36	55	0.9	4	0.1	0.4	1	2
** *LDLR* **	1	1	279	78	32	22	42	24	263	535
** *LRP1* **	1	1	424	841	4	7	6	2	59	1
** *APOBR* **	1	2	186	171	37	17	42	27	25	9
** *VDR* **	2	0.1	222	128	1	0.9	0.6	2	10	22
** *FZD1* **	0.4	0.9	15	14	0	0	0.2	0	28	24
** *FZD5* **	1	0.5	10	14	0.1	0	0.4	3	46	30
** *SCARF1* **	0.4	0.2	42	27	0.1	0.6	0.4	1	22	21
** *TSPAN4* **	4	0.8	15	19	2	0.5	3	2	86	69
** *MCOLN1* **	10	8	227	133	10	11	8	11	46	32
** *PLXNB2* **	11	2	898	473	0.8	2	2	12	34	45
** *TRPM2* **	12	7	324	243	0.3	0.9	1	4	4	3

**Table 21 ijms-25-13050-t021:** Calcium, sodium, chloride, and potassium channel transporters and regulators.

Gene	B Cells	B Cells	Monocytes	Monocytes	CD4^+^ T Cells	CD4^+^ T Cells	CD8^+^ T Cells	CD8^+^ T Cells	MCs	MCs
** *CACNA2D1* **	0	0	0	0	0	0	0	0	35	43
** *CACNA2D2* **	0.4	1	0.1	0	6	2	25	46	84	59
** *KCTD12* **	2	1	155	269	3	4	2	3	55	7
** *KCNA3* **	71	95	3	2	263	251	238	107	0.9	3
** *KCNH8* **	71	47	0.4	0.1	0.2	0	0	0	0.8	0.3
** *ATP1B1* **	3	9	4	2	8	7	3	3	204	187
** *TTYH3* **	15	13	886	511	5	3	7	4	32	69
** *FXYD6* **	0.5	0.1	60	127	0.1	0	0.5	0	10	1

**Table 22 ijms-25-13050-t022:** Angiogenesis inhibitors and promoters.

Gene	B Cells	B Cells	Monocytes	Monocytes	CD4^+^ T Cells	CD4^+^ T Cells	CD8^+^ T Cells	CD8^+^ T Cells	MCs	MCs
** *VASH1* **	0.5	0.7	47	31	3	4	6	7	6	0.5
** *ENG* **	4	5	39	18	5	6	8	12	113	178

**Table 23 ijms-25-13050-t023:** Sialic acid-binding Ig-like lectins (siglecs).

Gene	B Cells	B Cells	Monocytes	Monocytes	CD4^+^ T Cells	CD4^+^ T Cells	CD8^+^ T Cells	CD8^+^ T Cells	MCs	MCs
** *SIGLEC1* **	0	0	0.4	0.3	0	0	0	0	0	0
** *CD22* ** ** *(Siglec-2)* **	737	655	3	0.4	0.2	0	1	0	146	110
** *CD33* ** ** *(Siglec-3)* **	0.7	0	83	38	0.8	0	1	0	42	28
** *MAG* ** ** *(Siglec-4)* **	0.8	0.3	0.9	0.6	0	0.1	0	0.2	7	3
** *SIGLEC5* **	7	3	47	58	0	0	0.1	0.4	16	10
** *SIGLEC6* **	16	14	0.1	0.1	0	0.1	0	0	484	289
** *SIGLEC7* **	0	0	15	32	0.1	0	3	0.7	6	5
** *SIGLEC8* **	0	0	0	0	0	0	0	0	129	33
** *SIGLEC9* **	0	0	70	70	1	0.2	4	1	19	25
** *SIGLEC10* **	42	18	90	140	0.8	0	0.7	2	2	1
** *SIGLEC11* **	39	20	96	131	0.2	0	0.2	2	2	0.5
** *SIGLEC12* **	0.1	0.6	3	0.6	0	0	0	0	2	2
** *SIGLEC14* **	24	27	34	34	0.3	0.2	0.6	0.7	12	10
** *SIGLEC15* **	0	0	0	0	0	0	0	0	0	0
** *SIGLEC16* **	0.5	2	4	4	1	0.7	2	0.5	0.8	0.8

**Table 24 ijms-25-13050-t024:** S100 proteins.

Gene	B Cells	B Cells	Monocytes	Monocytes	CD4^+^ T Cells	CD4^+^ T Cells	CD8^+^ T Cells	CD8^+^ T Cells	MCs	MCs
** *S100A4* **	49	38	575	1189	623	430	385	147	357	227
** *S100A6* **	40	45	251	535	210	151	168	91	233	200
** *S100A8* **	2	1	807	1134	3	1	8	4	0	0.4
** *S100A9* **	2	3	1959	3562	5	3	9	3	0	0
** *S100A10* **	28	41	1867	212	319	234	250	117	102	248
** *S100Z* **	0.2	0.2	32	40	2	1	0.1	0.5	0.1	0
** *S100A11* **	4	3	74	94	48	20	30	4	322	749
** *S100A12* **	0.5	0.7	266	255	0.4	0.1	2	0.5	0.1	0

**Table 25 ijms-25-13050-t025:** Cell adhesion and other membrane proteins.

Gene	B Cells	B Cells	Monocytes	Monocytes	CD4^+^ T Cells	CD4^+^ T Cells	CD8^+^ T Cells	CD8^+^ T Cells	MCs	MCs
** *ITGB1* **	104	104	107	98	331	203	270	82	177	244
** *ITGB2* **	137	120	1079	1024	409	349	566	996	10	7
** *ITGB3* **	0.2	2	6	2	0	0.4	0.1	1	23	34
** *ITGB4* **	0.6	0	0.1	0.3	0.2	0.5	0	0.1	4	5
** *ITGB5* **	0	0	0	0.1	0.2	0	0.2	0	0.2	0.1
** *ITGB6* **	0	0	0	0	0	0	0	0	0	0
** *ITGB7* **	16	22	3	2	71	56	64	16	0.3	1
** *ITGB8* **	0.1	0	0	0	0.4	0	0	0	0.5	0.3
** *ITGA1* **	0	0	0	0.6	9	5	18	4	1	2
** *ITGA2* **	0	0	0	0.1	0.7	0.2	0	0	0.1	0
** *ITGA3* **	3	5	0	0.2	5	5	4	6	98	75
** *ITGA4* **	257	231	115	71	195	222	246	225	53	25
** *ITGA5* **	3	3	447	163	33	47	36	65	162	358
** *ITGA6* **	0	0	0.1	0.5	106	135	91	31	29	33
** *ITGA7* **	0.5	0.2	0.9	1	0.4	0.1	0.3	0.1	0.2	0.1
** *ITGA8* **	0	0	0	0	0	0	0	0	0.2	0.1
** *ITGA9* **	0.2	0.1	0.5	0.5	0	0.2	0	0.2	148	127
** *ITGAL* **	95	83	358	282	258	320	500	614	2	0.4
** *ITGAM* **	29	20	387	292	5	3	41	21	146	108
** *ITGAV* **	1	0.5	5	5	2	2	4	1	29	148
** *ITGAX* **	5	1	84	142	1	0.2	4	5	86	231
** *ICAM1* **	1	1	2	13	4	0.7	2	0.9	17	15
** *ICAM2* **	34	44	20	13	82	74	66	69	22	20
** *ICAM3* **	42	69	91	278	141	134	87	47	0.8	0.5
** *ICAM4* **	0.1	0.1	2	2	0.3	0	0.4	0	4	5
** *ICAM5* **	0	0	4	3	0	0	0.1	0.4	8	5
** *PXN* **	3	4	73	43	235	164	243	406	180	127
** *SELP* **	2	3	0.9	0.4	1	3	1	1	2	3
** *SELPLG* **	4	2	80	109	192	130	155	87	60	21
** *CELSR1* **	129	63	0.4	0.3	1	0.6	0.9	2	7	7
** *FAT1* **	0.1	0.5	0	0	0	0.1	0	0	34	36
** *EMP1* **	0.4	0.2	15	7	1	0.5	1	2	333	517
** *TJP2* **	2	0.7	13	9	5	4	2	2	60	110
** *MAL* **	0.2	2	0	0	320	398	179	488	6	14
** *GPNMB* **	0	0	1	1	0	0	0	0	543	467
** *TMEM176B* **	0.2	1	149	1605	0	0.2	0.2	2	30	36
** *AMICA1* **	0.1	0	85	188	2	2	3	2	0	0.3
** *SIDT1* **	41	31	1	0.1	29	31	50	39	0.5	0.7
** *AQP3* **	5	6	0.6	0.1	268	242	170	393	8	29
** *ITM2A* **	9	28	0.1	0.3	148	290	94	104	93	135

**Table 26 ijms-25-13050-t026:** Cell signaling proteins.

Gene	B Cells	B Cells	Monocytes	Monocytes	CD4^+^ T Cells	CD4^+^ T Cells	CD8^+^ T Cells	CD8^+^ T Cells	MCs	MCs
** *TYROBP* **	3	4	1164	1169	10	11	170	103	735	543
** *LYN* **	504	428	125	93	3	2	22	7	84	68
** *SYK* **	305	215	315	197	2	0.5	2	0.5	35	16
** *BTK* **	38	44	11	8	0	0.1	0.4	0	155	66
** *BLK* **	41	58	0.4	0.3	0.7	1	2	0.7	0	0
** *BLNK* **	250	291	22	31	38	26	37	13	81	67
** *LCK* **	0.2	0.2	0	0	8	10	12	15	0	0
** *THEMIS* **	0	0	0	0	40	45	72	29	0	0
** *LIME1* **	18	20	6	4	231	272	198	300	2	4
** *ZAP70* **	3	1	0.5	0.4	140	185	191	314	0	0.7
** *LCP2* **	3	6	68	48	259	283	249	114	427	252
** *LAT* **	7	3	108	125	237	328	251	615	290	352
** *JAK3* **	72	67	30	44	148	183	99	149	3	2
** *MATK* **	4	4	3	0.9	23	24	101	162	10	3
** *FYB* **	4	2	427	191	1068	1287	960	1136	2	1
** *RAB3D* **	10	12	303	363	6	11	8	10	64	22
** *RAB34* **	5	5	165	206	1	5	3	2	60	54
** *HCST* **	9	10	72	81	183	136	332	263	38	17
** *HCK* **	58	36	254	177	0.6	0.2	2	2	2	0.1
** *RASGEF1B* **	125	75	8	2	7	5	12	29	434	645
** *RASGRP4* **	0.4	0.2	54	43	0.7	0.6	0.5	0.1	118	48
** *FGD2* **	93	117	108	168	1	0.2	0.4	0.2	0.9	0
** *PRAM1* **	0.8	0.4	708	501	2	1	2	0.7	0.4	2
** *GCSAML* **	0.1	0.1	0.3	0.2	0.2	0.1	0	0	468	663
** *RHOU* **	0	0.1	200	141	3	2	3	1	13	13
** *RALGPS2* **	148	223	0.6	0.2	5	5	6	0.6	3	3
** *ICOSLG* **	533	372	216	160	13	7	9	12	26	32
** *TBC1D8* **	7	1	333	165	0.4	1	0.6	0.9	7	18
** *TBC1D9* **	464	283	75	78	1	0.4	1	0.1	16	27
** *BANK1* **	332	516	1	0.6	0.4	0	0.7	0.1	0	0.1
** *TCL1A* **	182	342	0.1	0.1	0.2	0	0.6	0	0	0.1
** *PLEKHG1* **	172	120	0.3	0	2	4	1	2	11	14
** *PLCG2* **	767	461	682	591	1	0.1	20	8	192	89
** *RHOH* **	574	642	0.9	0.3	297	388	325	531	351	166
** *TPD52* **	112	153	0.6	0.3	39	29	33	8	3	2
** *PDE3B* **	3	3	4	0.3	87	96	155	173	0.5	0.3
** *PDE4D* **	98	78	5	3	0.8	4	1	2	12	30
** *SNX22* **	82	210	0	0.3	0	0	0.2	0	0.4	0.6
** *RIN1* **	0.2	0.2	27	22	2	3	4	3	30	92
** *RIN2* **	1	0.4	132	132	0.1	0.2	0.3	0.2	54	45
** *ITK* **	0.1	0	0	0.1	208	241	203	193	78	395
** *YES1* **	0.2	0.2	0.3	0.1	27	36	71	44	84	107
** *GNAI1* **	0	0	0	0.1	1	0.1	0.3	0	98	28
** *GNAQ* **	0.4	0.7	154	125	80	118	57	29	127	108
** *PPM1H* **	0	0	1	4	0	0	0.1	0	118	37
** *CALB2* **	0	0	0	0	0	0	0	0	98	295
** *PDE3A* **	0	0	0.1	0	0	0	0.1	0	44	17
** *PDE4B* **	826	508	43	31	201	122	179	125	7	10
** *TIE1* **	0	0	0	0.1	0	0	0.2	0.4	27	28
** *RUSC2* **	0	0.2	45	26	0.6	0.4	0.2	0.2	40	48
** *RAB32* **	0	0	22	11	0	0	0.5	0	40	38
** *RGS13* **	0.2	0.1	0.3	0	0.2	0	0	0	281	181
** *NEDD9* **	0	0.4	0.8	5	8	7	6	2	158	167
** *RAB27B* **	0.1	0.2	2	1	2	2	3	0.4	277	87
** *RAB40C* **	11	10	125	129	10	10	7	11	12	11
** *PREX1* **	40	40	65	128	108	55	101	12	13	6
** *GMPR* **	0	0.3	10	3	0.3	0.5	0.5	0.7	88	52
** *ARHGAP18* **	7	5	10	10	5	7	9	7	299	345
** *ARHGAP22* **	5	4	73	51	0	0	0	0	0.1	0.4
** *ARHGEF10L* **	6	3	376	247	0.5	0	0.6	0.2	14	1
** *ARHGEF40* **	0	0.1	33	36	0.2	0	0.9	0	104	91
** *PIK3R3* **	0	0	0	0	0.6	0.6	3	0.9	43	40
** *EVPL* **	0	0	0	0	0.3	0.1	3	4	327	168
** *GIMAP7* **	0	0	4	13	52	83	77	11	12	6
** *AGAP1* **	0.1	0	0	0	9	7	15	22	226	200
** *ITK* **	0.1	0	0	0.1	208	241	203	193	78	395
** *DAPK1* **	0.5	0.4	113	64	0.2	0.5	0.4	0.9	75	121
** *RILP* **	0.5	0.7	119	116	0.9	1	0.5	0.4	12	8
** *SIRPB1* **	0.4	0.7	76	76	2	3	2	3	0	0
** *TIAM1* **	0.8	0.2	149	183	19	18	10	13	123	98
** *FHL3* **	1	2	85	36	3	3	10	45	42	41
** *LMTK3* **	1	1	0	0	18	19	24	47	0	0.1
** *CASS4* **	0.8	0.1	2	3	60	53	59	58	19	4
** *JDP2* **	2	1	121	116	0.2	0.2	0.1	0	6	5
** *NDRG2* **	2	2	3	0.4	15	12	14	22	191	158
** *FSCN1* **	2	1	10	8	3	4	5	7	211	230
** *PLCB3* **	3	3	355	117	3	3	2	8	21	24
** *DUSP3* **	11	6	59	40	7	9	6	22	132	143
** *DUSP4* **	6	2	0	0	32	18	13	21	146	488
** *DUSP6* **	24	17	28	33	3	5	4	6	728	951
** *DUSP7* **	8	7	310	296	40	49	39	79	254	137
** *DUSP14* **	4	3	0.3	0.2	17	13	16	16	130	433
** *DUSP23* **	6	9	190	287	17	20	29	47	52	36
** *TBC1D9* **	464	283	75	78	1	0.4	0.9	0.1	16	27
** *TNIK* **	4	3	3	0.5	54	100	65	85	261	278
** *AIF1* **	5	5	1515	1965	33	67	97	131	6	3
** *NREP* **	7	7	13	13	0.7	2	2	2	60	45
** *LPPR2* **	8	5	146	173	4	5	6	3	20	18
** *LRRK2* **	68	40	57	89	0.2	0	0.3	0.2	2	2
** *SDCBP* **	8	12	5	15	122	40	81	12	278	240
** *FGD6* **	11	14	222	157	0.4	1	0.4	1	10	11
** *ALS2* **	12	9	10	13	4	8	9	8	145	80
** *ADAP1* **	13	4	322	148	11	6	34	19	27	32
** *SIT1* **	13	16	0	0	39	31	39	85	0	0
** *PITPNC1* **	13	20	15	9	159	153	207	205	30	38
** *PSTPIP1* **	34	23	802	496	42	41	39	52	0.3	0.2
** *PPM1F* **	13	20	390	263	19	19	19	43	16	16
** *ARAP1* **	18	19	298	805	19	17	13	2	55	81
** *DGKG* **	29	9	107	89	0	0.1	0.1	0	5	7
** *CYFIP1* **	13	7	598	490	8	10	3	6	89	87
** *PLEK* **	26	14	545	543	3	1	33	8	14	7
** *ABI3* **	20	14	351	188	10	12	40	49	7	6
** *ARAP1* **	18	19	298	805	19	17	13	2	55	81
** *RASA4* **	17	5	369	161	3	5	2	7	23	39
** *CAMK4* **	17	7	0.5	0.3	377	575	294	144	0.3	0.9
** *DOK2* **	14	11	1558	1532	140	172	123	134	60	53

**Table 27 ijms-25-13050-t027:** Apoptosis-related molecules and chaperons.

	B Cells	B Cells	Monocyte	sMonocytes	CD4^+^ T Cells	CD4^+^ T Cells	CD8^+^ T Cells	CD8^+^ T Cells	MCs	MCs
** *FAIM2* **	0	0	0	0	0.2	0	0.1	0	50	48
** *BCL2* **	304	218	1	3	332	306	251	432	75	66
** *MCL1* **	232	259	154	306	564	473	504	225	2704	3365
** *CARD11* **	293	232	2	0.4	239	198	239	220	0.9	0.5
** *CLU* **	0.2	1	16	4	9	3	9	6	958	799

**Table 28 ijms-25-13050-t028:** Matrix proteins.

	B Cells	B Cells	Monocytes	Monocytes	CD4^+^ T Cells	CD4^+^ T Cells	CD8^+^ T Cells	CD8^+^ T Cells	MCs	MCs
** *COL4A3* **	54	31	0.1	0	0	0	0.3	0	0.3	0.1
** *COL19A1* **	447	403	2	0.7	0	0	0.3	0	0.1	0.5
** *COL6A2* **	1	0.2	0.1	0	39	23	119	190	242	15
** *COL13A1* **	0	0	0	0	0	0	0.4	0.6	72	34
** *FERMT2* **	0	0	0	0	0.8	0.9	0.5	0	92	103
** *EMILIN2* **	39	6	317	117	0.9	0.5	0.6	1	700	750
** *LAMA5* **	37	27	1	0	0	0	0	0	49	70

**Table 29 ijms-25-13050-t029:** Solute carriers.

Gene	B Cells	B Cells	Monocytes	Monocytes	CD4^+^ T Cells	CD4^+^ T Cells	CD8^+^ T Cells	CD8^+^ T Cells	MCs	MCs
** *SLC1A5* **	3	5	32	42	4	6	5	13	344	376
** *SLC2A9* **	0.2	0.3	51	118	0	0.4	0.6	0.4	0.7	0.8
** *SLC6A8* **	0.5	0.1	1	0.2	0.7	0.1	0.3	0.1	30	72
** *SLC7A7* **	32	21	102	95	2	0.2	1	1	2	2
** *SLC8A3* **	0.1	0.1	0	0	0	0	0	0	47	10
** *SLC9A1* **	17	12	24	16	15	16	17	36	210	150
** *SLC9A7* **	258	244	16	8	6	9	5	6	0.8	0.5
** *SLC9A9* **	19	24	6	3	27	35	23	8	4	5
** *SLC12A4* **	5	4	151	185	4	4	6	3	14	15
** *SLC15A3* **	22	23	154	173	1	1	3	2	6	6
** *SLC16A3* **	4	3	24	19	6	5	5	10	52	181
** *SLC18A2* **	0	0	0.4	0	0.6	0.7	1	2	698	702
** *SLC18B1* **	40	47	7	2	11	21	17	16	1	2
** *SLC25A44* **	6	6	11	5	10	8	9	18	467	189
** *SLC29A1* **	1	0.8	9	14	1	1	1	0.7	102	70
** *SLC30A1* **	9	9	98	78	9	9	20	24	57	50
** *SLC38A1* **	130	74	0	0.3	156	123	186	91	4	5
** *SLC39A11* **	6	8	68	260	9	11	9	2	26	7
** *SLC40A1* **	0	0.2	6	4	35	33	7	3	68	58
** *SLC43A3* **	6	5	43	46	9	10	14	15	352	387
** *SLC44A1* **	6	10	5	3	15	16	12	13	118	203

**Table 30 ijms-25-13050-t030:** Cell cycle and immediate-early response proteins.

	B Cells	B Cells	Monocytes	Monocytes	CD4^+^ T Cells	CD4^+^ T Cells	CD8^+^ T Cells	CD8^+^ T Cells	MCs	MCs
** *CDK14* **	144	113	4	3	0.4	0.6	0.4	0.1	2	7
** *G0S2* **	1	1	850	2437	2	2	2	2	252	45
** *CDK2* **	15	17	9	10	8	12	15	8	28	32
** *CDK4* **	8	10	13	18	9	14	13	27	152	129
** *CDK6* **	28	44	4	3	41	64	47	54	48	33
** *CCNA1* **	0	0	0	0	0	0	0	0	1	4
** *CCNA2* **	0.4	0.2	0.3	0	0.1	0.4	0.4	0.2	1	0.4
** *CCND1* **	4	2	0,3	0	0	0.6	0.1	0.1	131	178
** *CCND2* **	18	25	57	40	272	245	223	166	108	128
** *CCND3* **	82	111	163	139	210	293	270	833	137	129
** *CCNE1* **	0.6	0.7	0.5	0.1	0.7	3	2	0.5	6	4
** *CCNE2* **	0	0.2	0.1	0	0.1	0	1	0.1	6	5
** *IER3* **	2	0.4	904	510	3	12	5	79	1186	1693
** *EGR3* **	2	3	0	0.7	0.9	7	0.5	43	494	194

**Table 31 ijms-25-13050-t031:** Nuclear proteins and splicing factors.

Gene	B Cells	B Cells	Monocytes	Monocytes	CD4^+^ T Cells	CD4^+^ T Cells	CD8^+^ T Cells	CD8^+^ T Cells	MCs	MCs
** *PHLDA1* **	0	0.2	2	2	9	11	10	36	154	220
** *AHNAK2* **	0.2	0.1	0	0	0	0	0	0	139	75
** *SLFN5* **	6	3	23	13	143	160	175	192	208	65
** *LMNA* **	61	15	840	227	77	56	63	39	4302	6983
** *BCL7A* **	43	87	0.9	1	2	3	3	2	7	2
** *SF3B4* **	3	2	110	102	10	14	16	46	6	11

**Table 32 ijms-25-13050-t032:** Cytoskeletal and related proteins.

	B Cells	B Cells	Monocytes	Monocytes	CD4^+^ T Cells	CD4^+^ T Cells	CD8^+^ T Cells	CD8^+^ T Cells	MCs	MCs
** *CTTNBP2* **	0	0	2	0	0.1	0	0	0	138	116
** *TNS1* **	0.2	0	34	10	0	0.1	0	0	848	683
** *MYO1F* **	36	23	2491	868	74	49	117	128	34	12
** *MYO10* **	0.2	0.1	2	0.3	0	0.1	0	0	85	110
** *TPPP3* **	0.3	0	73	48	0	0.1	0.6	0	95	179
** *LSP1* **	1209	1200	4812	5211	808	917	921	1459	12	12
** *DBN1* **	1	0.6	30	15	3	2	91	120	66	65
** *PDLIM7* **	9	13	216	95	4	5	8	23	124	179
** *TNNI2* **	10	7	439	171	0	0.2	0	0.4	11	4
** *ADD2* **	30	57	0.1	0.1	0.6	6	3	6	0.4	0.6
** *TUBB6* **	24	23	46	30	0.2	0.1	0.1	0	99	145
** *GSN* **	15	11	135	180	2	2	5	4	205	198

**Table 33 ijms-25-13050-t033:** Vesicle and protein transport.

Gene	B Cells	B Cells	Monocyte	sMonocytes	CD4^+^ T Cells	CD4^+^ T Cells	CD8^+^ T Cells	CD8^+^ T Cells	MCs	MCs
** *NSG1* **	0	0	0	0	60	52	113	83	1	1
** *STX3* **	7	4	11	20	2	2	3	0.7	149	47
** *DYNLL1* **	15	21	31	14	30	32	34	46	457	756
** *STX11* **	11	10	262	204	31	26	36	35	196	327
** *SNX21* **	2	0.9	90	88	0.6	2	0.9	1	13	17
** *SNX22* **	82	210	0	0.3	0	0	0.2	0	0.4	0.6
** *VPS37C* **	13	8	177	167	3	5	2	3	29	34
** *AP5B1* **	28	36	588	457	15	17	15	31	14	20
** *TSPO* **	33	52	955	967	70	93	54	96	69	70
** *EHBP1L1* **	231	165	5041	5040	120	106	107	72	57	69

**Table 34 ijms-25-13050-t034:** Endogenous retrovirus and oncogenes.

Gene	B Cells	B Cells	Monocytes	Monocytes	CD4^+^ T Cells	CD4^+^ T Cells	CD8^+^ T Cells	CD8^+^ T Cells	MCs	MCs
** *ERVFRD-1* **	0	0	0.1	0	1	1	0	0.1	174	85
** *DLC1* **	0	0	0	0.3	0	0	0.3	0	690	419
** *FES* **	4	9	154	171	0.3	0.2	0.5	3	60	56

## Data Availability

All relevant data is available in the article or in the Appendix A.

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
