# Peer review of "Characterization of Freshly Isolated Human Peripheral Blood B Cells, Monocytes, CD4+ and CD8+ T Cells, and Skin Mast Cells by Quantitative Transcriptomics"

_ijms, 2024, doi:10.3390/ijms252313050_

Round 1

Reviewer 1 Report

Comments and Suggestions for Authors

The manuscript involves the analysis of quantitative transcriptomics of lymphocytes T, B, monocytes, and mast cells. The rationale is adequate. The methodology is complex but understandable. The manuscript contains many details expressed in tables that do not have legends, so the reader gets confused about several details. Some of the tables could be part of supplemental material. The assessment of different specific proteins may be highlighted based on the various subpopulations. A discussion of this possibility will enhance the manuscript. On the other hand, the conclusions are very long and should be shortened. It will also be interesting to add the limitations of the technique in the last part.

Author Response

Response to Reviewer 1

We have avoided adding a long text as table legend as all data is detailed described in the results section preceding the table including the actual reads in the majority of cases. Adding a Table legend should in our mind only be a repetition of the results section and make the manuscript even longer. I understand the comment but think it would only make the manuscript longer without adding additional clarification.

We can put some of the tables as supplementary but it would make it more difficult for the reader if just that table is the main interest of that particular reader to jump between main text and supplementary. We have avoided putting tables in the supplementary as we do not know what table is the main interest of a particular reader. Maybe we remove his or her table of major interest as you cannot by forehand know the particular interest of a specific reader.

The conclusion functions as a summary discussion and is in that sense very short as we also say as we only summarize a very minor part of the findings of this very large manuscript. We would have liked to make it even longer but due to the length of the manuscript we have tried to limit it to a minimum.

I fully agree that we can add a comment on the limitation of the methodology. This has been added to the conclusion section and marked in red for more easy location of the modified part.

We have also updated the summary figure as there were a few spelling mistakes.

Reviewer 2 Report

Comments and Suggestions for Authors

Akula et al. present an interesting and well-structured manuscript. The authors present innovative and well-focused results. However, the authors should make minor changes to improve different aspects:

-The title is too long and unspecific. The authors should be more focused and specific in nature to improve the reader's attention.

-The authors should include more keywords.

-The authors should include a more integrative and mechanistic graphic summary of the manuscript.

-The introduction to the state of the art is correct, but they should include more recent bibliography.

-The methodology should be more specific. The section should include more recent and specific references.

-The authors should justify the statistical methodology used.

-The authors should calculate the sample size.

-The authors should justify the statistical potential.

-The results are interesting, but they should be more self-explanatory.

-The authors should gradually focus each section.

-The discussion is correct. But it should be more mechanistic, ending with the transfer and justification of the limitations.

Author Response

Response to Reviewer 2.

Concerning the Title. We have actually reduced the title to a minimum to make it understandable. If we make it more specific we need to make it even longer so we are trying to find a balance between length and specificity and this is the best title we came up with in between 10 authors with multiple suggestions.  In the first version of the manuscript it was almost twice as long to give a reader the key findings to attract a reader to actually open the article, so sorry to say I have difficult to find a way to shorten it even more without loosing the possibility to attract readers, which I think is very important for the journal to keep the impact factor high.

We have included a number of more keywords if this is acceptable by IJMS? They are marked in red.

We have avoided to make the introduction too long and add many references as there are a very large number of both recent and older references in the entire manuscript. We have tried to add the references that actually first time comes with the statement we refer to in the text as this was the original finding and what we refer too. Then we have added more recent references to actual functions of some of the molecules when such data is available.

Figure 1 acts like a graphical abstract as it is the method used to isolate the cells FACS- showing the different parameters used to isolate and also the purity of the different cell fractions. As described in the text a very similar strategy has also been used for the fifth cell population the mast cells and there we have several references to articles that in more detail describes the actual steps.

No statistical method has been used as we only have two samples for each cell population and you can not do statistics on only two samples. However, we have in previous papers used more samples and as shown in the supplementary material where we have four samples of in vitro cultured mast cells the difference between samples are minor.

We done our best to make the results self-explanatory by giving actual reads and also to explain the function of each gene, and to give a reference or two for readers to be able to go back and look at the actual molecule and its function from a reference that often for the first time describes the function of that particular protein encoded by that gene.

In the conclusion we have added a section discussing the limitation of this methodology in relation to other methods to obtain quantitative information concerning expression levels of all human 21000 genes.

We have also updated the summary figure as there were a few spelling mistakes.